# A Theory for Conditional Generative Modeling on Multiple Data Sources

**Rongzhen Wang** [1 2 3]  **Yan Zhang** [4]  **Chenyu Zheng** [1 2 3]  **Chongxuan Li** [1 2 3 †]  **Guoqiang Wu** [4 †]

## Abstract

The success of large generative models has driven a paradigm shift, leveraging massive multi-source data to enhance model capabilities. However, the interaction among these sources remains theoretically underexplored. This paper takes the first step toward a rigorous analysis of multi-source training in conditional generative modeling, where each condition represents a distinct data source. Specifically, we establish a general distribution estimation error bound in average total variation distance for conditional maximum likelihood estimation based on the bracketing number. Our result shows that when source distributions share certain similarities and the model is expressive enough, multi-source training guarantees a sharper bound than single-source training. We further instantiate the general theory on conditional Gaussian estimation and deep generative models including autoregressive and flexible energy-based models, by characterizing their bracketing numbers. The results highlight that the number of sources and similarity among source distributions improve the advantage of multi-source training. Simulations and real-world experiments validate our theory. Code is available at: `https://github.com/ML-GSAI/Multi-Source-GM`.

## 1. Introduction

Large generative models have achieved remarkable success in generating realistic and complex outputs across natural language (Brown et al., 2020; Touvron et al., 2023) and computer vision (Rombach et al., 2022; OpenAI, 2024). A key factor behind their strong performance is the diverse and rich training data. For instance, large language models are trained on *heterogeneous* datasets comprising web content, books, and code (Brown et al., 2020; Hu et al., 2024b), while image generation models benefit from vast datasets spanning various categories and aesthetic qualities (Peebles & Xie, 2023; Chen et al., 2024; Esser et al., 2024). Empirical evidence suggests that, under certain conditions, training on *multiple data sources* can enhance performance across all sources (Pires et al., 2019; Chen et al., 2024; Allen-Zhu & Li, 2024a). Consequently, data mixture strategies have become an essential research topic (Nguyen et al., 2022; Chidambaram et al., 2022; Hu et al., 2024b).

However, the theoretical underpinnings of this multi-source training paradigm remain poorly understood. This raises a fundamental question: *is it more effective to train separate models on individual data sources, or to train a single model using data from multiple sources?* In this paper, we take the first step toward a rigorous analysis of multi-source training, focusing on its impact on conditional generative models, where each condition represents a distinct data source.

Our first contribution is establishing a general upper bound on distribution estimation error for conditional generative modeling via maximum likelihood estimation (MLE) in Section 3. Specifically, we measure the error using average total variation (TV) distance between the true and estimated conditional distributions across all sources, which scales as $\tilde{\mathcal{O}}(\sqrt{\log \mathcal{N}_{\mathcal{P}_{X|Y}}/n})$, where $n$ is the training set size and $\mathcal{N}_{\mathcal{P}_{X|Y}}$ is the bracketing number of the conditional distribution space $\mathcal{P}_{X|Y}$. Further, when source distributions exhibit parametric similarity, multi-source training effectively reduces the complexity of the distribution space, leading to a provably sharper bound than single-source training.

Technically, our analysis extends classical MLE estimation error bounds from empirical process theory (Wong & Shen, 1995; Geer, 2000; Ge et al., 2024) to the conditional setting by adapting the complexity of the distribution space and measuring the estimation error in terms of average TV distance. Further discussions are provided in Section 6.

As the second contribution, we instantiate our general theory in three specific cases: (1) parametric estimation of conditional Gaussian distributions, a simple example clearly

[1] Gaoling School of Artificial Intelligence, Renmin University of China, Beijing, China [2] Beijing Key Laboratory of Research on Large Models and Intelligent Governance [3] Engineering Research Center of Next-Generation Intelligent Search and Recommendation, MOE [4] School of Software, Shandong University, Shandong, China. [†] Correspondence to: Chongxuan Li <chongxuanli@ruc.edu.cn>, Guoqiang Wu <guoqiangwu@sdu.edu.cn>.

*Proceedings of the 42st International Conference on Machine Learning*, Vancouver, Canada. PMLR 267, 2025. Copyright 2025 by the author(s).

illustrating how source distribution properties influence the benefits of multi-source training, (2) autoregressive models (ARMs), the foundation of large language models (Brown et al., 2020; Touvron et al., 2023; Liu et al., 2024; Bai et al., 2023; Zheng et al., 2024), and (3) energy-based models (EBMs), a general class of generative models for continuous data (LeCun et al., 2006; Du & Mordatch, 2019; Song & Ermon, 2019; Zhao et al., 2022). For each model, we derive explicit estimation error bounds for both multi-source and single-source training by measuring the bracketing number of the corresponding conditional distribution space. Based on the theoretical results in these instantiations, we observe a common pattern: across all cases, the ratio of multi-source to single-source estimation error bounds takes the form $\sqrt{1 - (K-1/K)\beta_{\mathrm{sim}}}$, where $K$ is the number of sources and $\beta_{\mathrm{sim}} \in [0, 1]$ is an inductively derived quantity that can be interpreted as similarity among source distributions, with model-specific definitions detailed in Section 4. This ratio decreases with both $K$ and $\beta_{\mathrm{sim}}$, indicating that the number of sources and their similarity improve the benefits of multi-source training.

A core technical contribution is establishing novel bracketing number bounds for ARMs and EBMs. This is challenging since on the one hand, the bracketing number provides a refined measure of function spaces, requiring carefully designed one-sided bounds over the entire domain. On the other hand, the conditional distribution space of deep generative models is inherently complex, involving both neural network architectures and specific probabilistic characteristics for different generative modeling methods. Please refer to Appendixes C and D for detailed proofs and discussions.

Finally, we validate our theoretical findings through simulations and real-world experiments in Section 5. In simulations, we perform conditional Gaussian estimation, where the MLE solutions can be analytically derived, enabling a rigorous assessment of the tightness of our bounds. The close match between the empirical and theoretical error orders supports the validity of our results. Beyond simulations, we train class-condition diffusion models (Karras et al., 2024) on ILSVRC2012 (Russakovsky et al., 2015) where its semantic hierarchy (Deng et al., 2010) provides a natural way to define inter-source distribution similarity. Empirical results confirm that multi-source training outperforms single-source training by achieving lower FID scores, consistent with our theoretical guarantee in Section 3, and this advantage depends on both the number of sources and their similarity, supporting our insights in Section 4.

## 2. Problem formulation

**Elementary notations.** Scalars, vectors, and matrices are denoted by lowercase letters (e.g., $a$), lowercase boldface letters (e.g., $\boldsymbol{a}$), and uppercase boldface letters (e.g., $\boldsymbol{A}$). We

use $\boldsymbol{a}[m]$ to denote the $m$-th entry of vector $\boldsymbol{a}$, and $\boldsymbol{A}[m,:]$, $\boldsymbol{A}[:,n]$, and $\boldsymbol{A}[m,n]$ to denote the $m$-th row, the $n$-th column, and the entry at the $m$-th row and the $n$-th column of $\boldsymbol{A}$. $(\boldsymbol{a}, \boldsymbol{b})$ denotes the concatenation of $\boldsymbol{a}$ and $\boldsymbol{b}$ as a column vector. We denote $[n] := \{1, \ldots, n\}$ for any $n \in \mathbb{N}$ and $a \vee b$ as $\max\{a, b\}$. For any measurable scalar function $f(\boldsymbol{x})$ on domain $\mathcal{X}$ and real number $1 \leq \mathsf{p} \leq \infty$, its $L^{\mathsf{p}}(\mathcal{X})$-norm is defined as $\|f(\boldsymbol{x})\|_{L^{\mathsf{p}}(\mathcal{X})} := (\int_{\mathcal{X}} |f(\boldsymbol{x})|^{\mathsf{p}} d\boldsymbol{x})^{\frac{1}{\mathsf{p}}}$. When $\mathsf{p} = \infty$, $\|f(\boldsymbol{x})\|_{L^{\infty}(\mathcal{X})} = \sup_{\boldsymbol{x} \in \mathcal{X}} |f(\boldsymbol{x})|$. $\mathbb{I}(\cdot)$ denotes the indicator function. Notation $a_n = \tilde{\mathcal{O}}(b_n)$ indicates $a_n$ is asymptotically bounded above by $b_n$ up to logarithmic factors.

### 2.1. Data from multiple sources

Let $X$ denote the random variable for data (e.g., a natural image) in a data space $\mathcal{X}$, and $Y$ denote the random variable for the source label in a label space $\mathcal{Y}$. Suppose there are $K$ data sources (e.g., $K$ categories of images), each corresponding to an unknown conditional distribution $p_{X|k}^*$ for $k \in [K]$. We assume that $p_{X|k}^*$ is parameterized by a source-specific feature $\phi_k^*$ in a parameter space $\Phi$ and a shared feature $\psi^*$ in a parameter space $\Psi$, such that $p_{X|k}^*(x|k) = p_{\phi_k^*, \psi^*}(x|k)$. The conditional distribution of $X$ given $Y = y$ is consequently expressed as

$$p_{X|Y}^*(\boldsymbol{x}|y) = \prod_{k=1}^{K} \Big( p_{\phi_k^*, \psi^*}(\boldsymbol{x}|k) \Big)^{\mathbb{I}(y=k)}.$$

This compact representation provides convenience for subsequent discussions.

We further assume the distribution of $Y$ is known since the proportion of data from different sources is often manually designed in practice (Deng et al., 2009; Krizhevsky et al., 2009; Brown et al., 2020; Chen et al., 2024). The joint distribution of $X$ and $Y$ is then given by $p_{X,Y}^*(\boldsymbol{x}, y) = p_{X|Y}^*(\boldsymbol{x}|y) p_Y^*(y)$.

### 2.2. Conditional generative modeling

Consider a dataset $S = \{(\boldsymbol{x}_i, y_i)\}_{i=1}^{n}$ consisting of $n$ independent and identically distributed (i.i.d.) data-label pairs sampled from the joint distribution $p_{X,Y}^*$. In the learning phase, a conditional generative model uses maximum likelihood estimation (MLE) to estimate $p_{X|Y}^*$ based on the dataset $S$, where the conditional likelihood is defined as

$$\mathcal{L}_S(p_{X|Y}) := \prod_{i=1}^{n} p_{X|Y}(\boldsymbol{x}_i|y_i). \tag{1}$$

**Multi-source training.** Under multi-source training, the conditional distribution space is given by $\mathcal{P}_{X|Y}^{\mathrm{multi}} :=$

$$\Big\{ p_{X|Y}^{\mathrm{multi}}(\boldsymbol{x}|y) = \textstyle\prod_{k=1}^{K} \big( p_{\phi_k, \psi}(\boldsymbol{x}|k) \big)^{\mathbb{I}(y=k)} : \phi_k \in \Phi, \psi \in \Psi \Big\},$$

and the corresponding estimator of $p^*_{X|Y}$ is

$$\hat{p}^{\text{multi}}_{X|Y} = \underset{p^{\text{multi}}_{X|Y} \in \mathcal{P}^{\text{multi}}_{X|Y}}{\arg\max} \mathcal{L}_S(p^{\text{multi}}_{X|Y}). \qquad (2)$$

Here, we adopt the realizable assumption that true parameters satisfy $\phi^*_k \in \Phi$ and $\psi^* \in \Psi$ following (Ge et al., 2024), which allows the estimation error analysis to focus on the generalization property of the distribution space.

**Single-source training.** Under single-source training, we train $K$ conditional generative models for each source using data exclusively from the corresponding source. For any particular source $k$, denoting $S_k := \{(\boldsymbol{x}_i, y_i) \in S | y_i = k\} = \{\boldsymbol{x}^k_j, k\}^{n_k}_{j=1}$, the corresponding generative model estimate $p^*_{X|k}$ by maximizing the conditional likelihood on $S_k$ as

$$\hat{p}^{\text{single}}_{X|k} = \underset{p^{\text{single}}_{X|k} \in \mathcal{P}^{\text{single}}_{X|k}}{\arg\max} \mathcal{L}_{S_k}(p^{\text{single}}_{X|k}),$$

where $\mathcal{L}_{S_k}(p_{X|k}) := \prod^{n_k}_{j=1} p_{X|k}(\boldsymbol{x}^k_j|k)$ and $\mathcal{P}^{\text{single}}_{X|k} := \{p_{\phi_k,\psi_k}(\boldsymbol{x}|k) : \phi_k \in \Phi, \psi_k \in \Psi\}$.

Separately maximizing these $K$ objectives is equivalent to finding the maximizer of $L_S$ in conditional distribution space $\mathcal{P}^{\text{single}}_{X|Y} :=$

$$\left\{ p^{\text{single}}_{X|Y}(\boldsymbol{x}|y) = \prod^K_{k=1} \left( p_{\phi_k,\psi_k}(\boldsymbol{x}|k) \right)^{\mathbb{I}(y=k)} : \phi_k \in \Phi, \psi_k \in \Psi \right\}.$$

Therefore, the estimator of $p^*_{X|Y}$ under single-source training is

$$\hat{p}^{\text{single}}_{X|Y} = \underset{p^{\text{single}}_{X|Y} \in \mathcal{P}^{\text{single}}_{X|Y}}{\arg\max} \mathcal{L}_S(p^{\text{single}}_{X|Y}). \qquad (3)$$

The introduced multi-source setting abstracts practical scenarios where different sources share certain underlying data structures (via $\psi$) while retaining unique characteristics (via $\phi_k$). At the same time, the single-source setting provides a controlled comparison to rigorously assess whether incorporating other sources improves the model's learning.

**Evaluation for conditional distribution estimation.** We measure the accuracy of conditional distribution estimation by average TV distance between the estimated and true conditional distributions, referred to as *average TV error*:

$$\mathcal{R}_{\overline{\text{TV}}}(\hat{p}_{X|Y}) := \mathbb{E}_Y \left[ \text{TV}(\hat{p}_{X|Y}, p^*_{X|Y}) \right], \qquad (4)$$

where $\text{TV}(\hat{p}_{X|Y}, p^*_{X|Y}) = \frac{1}{2} \int_{\mathcal{X}} |\hat{p}_{X|Y}(\boldsymbol{x}|y) - p^*_{X|Y}(\boldsymbol{x}|y)| d\boldsymbol{x}$ is the total variation distance between $\hat{p}_{X|Y}$ and $p^*_{X|Y}$.

In the following sections, we investigate the effectiveness of multi-source training by measuring and comparing $\mathcal{R}_{\overline{\text{TV}}}(\hat{p}^{\text{multi}}_{X|Y})$ and $\mathcal{R}_{\overline{\text{TV}}}(\hat{p}^{\text{single}}_{X|Y})$.

# 3. Provable advantage of multi-source training

In this section, we establish a general upper bound on the average TV error for conditional MLE and provide a statistical guarantee for the benefits of multi-source training. Our analysis extends the classical MLE guarantees (Geer, 2000; Ge et al., 2024), which leverage the bracketing number and the uniform law of large numbers.

## 3.1. Complexity of the conditional distribution space

We begin by introducing an extended notion of the bracketing number as follows.

**Definition 3.1** ($\epsilon$-upper bracketing number for conditional distribution space)**.** Let $\epsilon$ be a real number that $\epsilon > 0$ and $\mathsf{p}$ be an integer that $1 \leq \mathsf{p} \leq \infty$. An $\epsilon$-upper bracket of a conditional distribution space $\mathcal{P}_{X|Y}$ with respect to $L^{\mathsf{p}}(\mathcal{X})$ is a finite function set $\mathcal{B}$ such that for any $p_{X|Y} \in \mathcal{P}_{X|Y}$, there exists some $p' \in \mathcal{B}$ such that given any $y \in \mathcal{Y}$, it holds

$$\forall \boldsymbol{x} \in \mathcal{X} : p'(\boldsymbol{x}, y) \geq p_{X|Y}(\boldsymbol{x}|y), \text{ and}$$
$$\|p'(\cdot, y) - p_{X|Y}(\cdot|y)\|_{L^{\mathsf{p}}(\mathcal{X})} \leq \epsilon.$$

The $\epsilon$-upper bracketing number $\mathcal{N}_{[]}\left(\epsilon; \mathcal{P}_{X|Y}, L^{\mathsf{p}}(\mathcal{X})\right)$ is the cardinality of the smallest $\epsilon$-upper bracket.

This notion quantifies the minimal set of functions needed to upper bound every conditional distribution within a small margin, reducing error analysis from an infinite to a finite function class. Unlike traditional bracketing numbers for unconditional distributions $p_X$ using two-sided brackets (Wellner, 2002), this extension employs one-sided upper brackets (Ge et al., 2024) and requires uniform coverage across $y$ for all conditional distributions tailored for our setting. We provide a concrete example and corresponding visualization in Appendix G to make this notion more accessible.

## 3.2. Guarantee for conditional MLE

We now present a general error bound which applies to both training strategies.

**Theorem 3.2** (Average TV error bound for conditional MLE, proof in Appendix A.1)**.** *Given a dataset $S$ of size $n$ that i.i.d. sampled from $p^*_{X,Y}$, let $\hat{p}_{X|Y}$ be the maximizer of $L_S(p_{X|Y})$ defined in Equation (1) in conditional distribution space $\mathcal{P}_{X|Y}$. Suppose the real conditional distribution $p^*_{X|Y}$ is contained in $\mathcal{P}_{X|Y}$. Then, for any $0 < \delta \leq 1/2$, it holds with probability at least $1 - \delta$ that*

$$\mathcal{R}_{\overline{\text{TV}}}(\hat{p}_{X|Y}) \leq 3\sqrt{\frac{1}{n}\left(\log \mathcal{N}_{[]}\left(\frac{1}{n}; \mathcal{P}_{X|Y}, L^1(\mathcal{X})\right) + \log\frac{1}{\delta}\right)}.$$

As formulated in Equation (2) and Equation (3), multi-source and single-source training apply conditional MLE on

$S$ within different conditional distribution spaces. The following proposition shows that multi-source training reduces the bracketing number of its distribution space through source similarity.

**Proposition 3.3** (Multi-source training reducing complexity, proof in Appendix A.2.)**.** *Let $\mathcal{P}_{X|Y}^{\mathrm{multi}}$ and $\mathcal{P}_{X|Y}^{\mathrm{single}}$ be as defined in Section 2. Then, for any $\epsilon > 0$ and $1 \leq \mathsf{p} \leq \infty$, we have*

$$\mathcal{N}_{[]}\Big(\epsilon; \mathcal{P}_{X|Y}^{\mathrm{multi}}, L^{\mathsf{p}}(\mathcal{X})\Big) \leq \mathcal{N}_{[]}\Big(\epsilon; \mathcal{P}_{X|Y}^{\mathrm{single}}, L^{\mathsf{p}}(\mathcal{X})\Big).$$

Combining Theorem 3.2 and Proposition 3.3, we conclude that when source distributions have parametric similarity and the model satisfies the realizable assumption, multi-source training can enjoy a sharper estimation guarantee than single-source training. Simulations and real-world experiments in Section 5 support our result.

## 4. Instantiations

We now apply our general analysis to conditional Gaussian estimation and two deep generative models to obtain concrete error bounds.

### 4.1. Parametric estimation on Gaussian distributions

As employed in extensive work (Montanari & Saeed, 2022; Wang & Thrampoulidis, 2022; He et al., 2022; Zheng et al., 2023b; Dandi et al., 2024; Zheng et al., 2023a), Gaussian models provide a simple yet insightful case for illustrating the benefits of multi-source training and enable analytically tractable simulations under our theoretical assumptions.

**Parametric distribution family.** Suppose each of the $K$ conditional distributions is a $d$-dimensional standard Gaussian distribution, i.e.,

$$\forall k \in [K], \quad X|k \sim \mathcal{N}(\boldsymbol{\mu}_k^*, \boldsymbol{I}_d) = (2\pi)^{-\frac{d}{2}} e^{-\frac{1}{2}\|\boldsymbol{x}-\boldsymbol{\mu}_k^*\|_2^2},$$

with a mean vector $\boldsymbol{\mu}_k^*$ and an identity covariance matrix $\boldsymbol{I}_d \in \mathbb{R}^{d \times d}$. We assume each mean vector has two parts: the first $d_1$ entries $\boldsymbol{\mu}_k^*[1:d_1]$ represent the source-specific feature which is potentially different for each source, and the remaining entries $\boldsymbol{\mu}_k^*[d_1 + 1:d]$ represent the shared feature which is identical across all sources. Corresponding to the general formulation in Section 2, we denote

$$\phi_k := \boldsymbol{\mu}_k^*[1:d_1], \psi := \boldsymbol{\mu}_1^*[d_1+1:d] = \cdots = \boldsymbol{\mu}_K^*[d_1+1:d],$$

and the conditional distribution is parameterized as

$$p_{\phi_k, \psi}(\boldsymbol{x}|k) = (2\pi)^{-\frac{d}{2}} e^{-\frac{1}{2}\|\boldsymbol{x}-(\phi_k,\psi)\|_2^2}. \qquad (5)$$

**Statistical guarantee of the average TV error.** In this formulation, the conditional MLE in $\mathcal{P}_{X|Y}^{\mathrm{multi}}$ under multi-source training leads to the following result.

**Theorem 4.1** (Average TV error bound for conditional Gaussian estimation under multi-source training, proof in Appendix B.2)**.** *Let $\hat{p}_{X|Y}^{\mathrm{multi}}$ be the likelihood maximizer defined in Equation* (2) *given $\mathcal{P}_{X|Y}^{\mathrm{multi}}$ with conditional distributions as in Equation* (5)*. Suppose $\Phi = [-B, B]^{d_1}$, $\Psi = [-B, B]^{d-d_1}$ with constant $B > 0$, and $\phi_k^* \in \Phi$, $\psi^* \in \Psi$. Then, for any $0 < \delta \leq 1/2$, it holds with probability at least $1 - \delta$ that*

$$\mathcal{R}_{\overline{\mathrm{TV}}}(\hat{p}_{X|Y}^{\mathrm{multi}}) = \tilde{\mathcal{O}}\Bigg(\sqrt{\frac{(K-1)d_1 + d}{n}}\Bigg).$$

In contrast, single-source training results in an error of $\mathcal{R}_{\overline{\mathrm{TV}}}(\hat{p}_{X|Y}^{\mathrm{single}}) = \tilde{\mathcal{O}}\Big(\sqrt{Kd/n}\Big)$, with a formal result provided in Theorem B.2. The advantage of multi-source learning can be quantified by the ratio of error bounds: $\sqrt{\frac{(K-1)d_1+d}{Kd}} = \sqrt{1 - \frac{K-1}{K}\frac{d-d_1}{d}}$.

Letting $\beta_{\mathrm{sim}} := \frac{d-d_1}{d}$, where $\frac{d-d_1}{d}$ represents the proportion of the shared mean dimensions relative to the total dimensionality, this quantity $\beta_{\mathrm{sim}}$ can thus be interpreted as the similarity among source distributions. As we will see in subsequent instantiations, this general form of ratio $\sqrt{1 - \frac{K-1}{K}\beta_{\mathrm{sim}}}$ applies across Section 4.2 and Section 4.3, with $\beta_{\mathrm{sim}}$ instantiated in a case-specific manner. Further discussion on the notion of $\beta_{\mathrm{sim}}$ and the measure of distribution similarity in practice can be found in Appendix F.

Notably, this ratio decreases with both the number of sources $K$ and source similarity $\beta_{\mathrm{sim}}$. As $K$ increases from 1 to $\infty$, the ratio decreases from 1 to $\sqrt{1 - \beta_{\mathrm{sim}}}$, and as $\beta_{\mathrm{sim}}$ increases from 0 (completely dissimilar distributions) to 1 (completely identical distributions), it decreases from 1 to $\sqrt{1/K}$, reflecting a transition from no asymptotic gain to a constant improvement. This highlights that the number of sources and distribution similarity enhance the benefits of multi-source training. Empirical results in Section 5.2 confirm this trend.

### 4.2. Conditional ARMs on discrete distributions

For deep generative models, our formulations are based on multilayer perceptrons (MLPs), a fundamental network component, with potential extensions to Transformers and convolution networks with existing literature (Lin & Zhang, 2019; Ledent et al., 2021; Shen et al., 2021; Hu et al., 2024a; Trauger & Tewari, 2024; Jiao et al., 2024). We formally define MLPs mainly following notations in Oko et al. (2023)

**Definition 4.2** (Class of MLPs)**.** A class of MLPs $\mathcal{F}(L, W, S, B)$ with depth $L$, width $W$, sparsity $S$, norm $B$, and element-wise ReLU activation that $\mathrm{ReLU}(x) = 0 \vee x$ is defined as $\mathcal{F}(L, W, S, B) := \{\boldsymbol{f}(\boldsymbol{x}) = (\boldsymbol{A}^{(L)}\mathrm{ReLU}(\cdot) + \boldsymbol{b}^{(L)}) \circ \cdots \circ (\boldsymbol{A}^{(1)}\boldsymbol{x} + \boldsymbol{b}^{(1)}) : \{(\boldsymbol{A}^{(l)}, \boldsymbol{b}^{(l)})\}_{l=1}^{L} \in$

$\mathcal{W}(L, W, S, B)\}$, where parameter space $\mathcal{W}(L, W, S, B)$ is defined by $\mathcal{W}(L, W, S, B) := \{\{(\boldsymbol{A}^{(l)}, \boldsymbol{b}^{(l)})\}_{l=1}^{L} : \boldsymbol{A}^{(l)} \in \mathbb{R}^{W_l \times W_{l-1}}, \boldsymbol{b}^{(l)} \in \mathbb{R}^{W_l}, \max_l W_l \leq W, \sum_{l=1}^{L}(\|\boldsymbol{A}^{(l)}\|_0 + \|\boldsymbol{b}^{(l)}\|_0) \leq S, \max_l \|\boldsymbol{A}^{(l)}\|_\infty \vee \|\boldsymbol{b}^{(l)}\|_\infty \leq B\}$.

We now present the formulation for ARMs, which can be viewed as an extension of Uria et al. (2016).

**Probabilistic modeling with autoregression.** Consider a common data scenario for the natural language where $X$ represents a $D$-length text in $[M]^D$. Each dimension of $X$ is an integer token following an $M$-categorical distribution with $M$ being the vocabulary size. Adopting the autoregressive approach of probabilistic modeling, conditional distribution $p_{X|Y}(\boldsymbol{x}|y)$ is factorized using the chain rule as:

$$p(\boldsymbol{x}|y) = p(x_1|y) \cdots p(x_D|\boldsymbol{x}_{<D}, y)$$
$$= p(x_1; \boldsymbol{\rho}(y)) \cdots p(x_D; \boldsymbol{\rho}(\boldsymbol{x}_{<D}, y)).$$

We omit the subscripts for notation simplicity. Here, for any $d \in [D]$, $\boldsymbol{\rho}(x_{<d}, y)$ is the distribution parameter for $X_d$ given $X_{<d}, Y = \boldsymbol{x}_{<d}, y$ that

$$p(x_d = m | \boldsymbol{x}_{<d}, y) = \boldsymbol{\rho}(\boldsymbol{x}_{<d}, y)[m],$$

satisfying $\boldsymbol{\rho}(\boldsymbol{x}_{<d}, y) \in \mathbb{R}_+^M$ and $\sum_{m=1}^{M} \boldsymbol{\rho}(\boldsymbol{x}_{<d}, y)[m] = 1$.

**Distribution estimation via neural network.** Aligning with common practices, we suppose the distribution parameter vector is estimated with $\boldsymbol{\rho}_\theta(x_{<d}, y)$ using a shared neural network parameterized by $\theta$ across all dimensions. The network comprises an embedding layer, an encoding layer, an MLP block, and a softmax output layer.

Specifically, we first look up $\boldsymbol{x}$ and $y$ in two embedding matrices $\boldsymbol{V}_X \in [0, 1]^{M \times d_e}$ and $\boldsymbol{V}_Y \in [0, 1]^{K \times d_e}$, then stack the embeddings to get

$$\boldsymbol{E}_{\boldsymbol{V}_Y, \boldsymbol{V}_X}(\boldsymbol{x}, y) = \begin{bmatrix} \boldsymbol{V}_Y[y, :] \\ \boldsymbol{V}_X[x_1, :] \\ \vdots \\ \boldsymbol{V}_X[x_{D-1}, :] \end{bmatrix} \in [0, 1]^{D \times d_e},$$

where the last dimension of $x$ is excluded since it is not used when estimating the distribution.

Subsequently, we encode each embedding by a linear transformation with parameters $\boldsymbol{A}_0 \in \mathbb{R}^{D \times d_e}$, $\boldsymbol{b}_0 \in \mathbb{R}^D$ and normalize the output with an element-wise sigmoid function $\sigma(x) = \frac{1}{1+e^{-x}}$ as

$$\boldsymbol{v}_{\boldsymbol{A}_0, \boldsymbol{b}_0}(\boldsymbol{E}_{\boldsymbol{V}_Y, \boldsymbol{V}_X}(\boldsymbol{x}, y))$$
$$= \begin{bmatrix} \sigma(\boldsymbol{A}_0[1, :]\boldsymbol{V}_Y[y, :]^\top + \boldsymbol{b}_0[1]) \\ \vdots \\ \sigma(\boldsymbol{A}_0[D, :]\boldsymbol{V}_X[x_{D-1}, :]^\top + \boldsymbol{b}_0[D]) \end{bmatrix} \in [0, 1]^D.$$

To ensure no components related to $\boldsymbol{x}_{\geq d}$ is seen when estimating the conditional probability for $x_d$, we mask $\boldsymbol{v}_{\boldsymbol{A}_0, \boldsymbol{b}_0}$ using a $(D - d)$-dimensional zero vector $\boldsymbol{0}_{D-d}$ as

$$\boldsymbol{v}_{\boldsymbol{A}_0, \boldsymbol{b}_0}^{\backslash \boldsymbol{0}_{D-d}} := \begin{bmatrix} \boldsymbol{v}_{\boldsymbol{A}_0, \boldsymbol{b}_0}[1:d]^\top & \boldsymbol{0}_{D-d}^\top \end{bmatrix}^\top.$$

Then we calculate the distribution parameter vector by an MLP $\boldsymbol{f}_\omega \in \mathcal{F}(L, W, S, B)$ with $W_0 = D$ and $W_L = M$, followed by a softmax layer as

$$\boldsymbol{\rho}_\theta(\boldsymbol{x}_{<d}, y) = \text{softmax}\left(\boldsymbol{f}_\omega\left(\boldsymbol{v}_{\boldsymbol{A}_0, \boldsymbol{b}_0}^{\backslash \boldsymbol{0}_{D-d}}(\boldsymbol{E}_{\boldsymbol{V}_Y, \boldsymbol{V}_X}(\boldsymbol{x}, y))\right)\right),$$

This leads to conditional distribution as

$$p_\theta(\boldsymbol{x}|y) = p(x_1; \boldsymbol{\rho}_\theta(y)) \cdots p(x_D; \boldsymbol{\rho}_\theta(\boldsymbol{x}_{<D}, y)). \quad (6)$$

When training such an ARM, each row of $\boldsymbol{V}_Y$ is only optimized on data with the corresponding condition, while parameters in $\boldsymbol{V}_X, \boldsymbol{A}_0, \boldsymbol{b}_0$, and $\omega$ are optimized on data with all conditions. That means $\boldsymbol{V}_Y[k, :]$ serves as the source-specific parameter, while other parameters are shared across all sources. Corresponding to the general formulation in Section 2, we denote

$$\phi_k := \boldsymbol{V}_Y[k, :], \text{ and } \psi := \{\boldsymbol{V}_X, \boldsymbol{A}_0, \boldsymbol{b}_0, \omega\}.$$

**Statistical guarantee of the average TV error.** In this formulation, the conditional MLE in $\mathcal{P}_{X|Y}^{\text{multi}}$ under multi-source training leads to the following result.

**Theorem 4.3** (Average TV error bound for ARMs under multi-source training, proof in Appendix C.4). *Let $\hat{p}_{X|Y}^{\text{multi}}$ be the likelihood maximizer defined in Equation* (2) *given $\mathcal{P}_{X|Y}^{\text{multi}}$ with conditional distributions as in Equation* (6). *Suppose $\Phi = [0, 1]^{d_e}$, $\Psi = [0, 1]^{M \times d_e} \times [-B, B]^{D \times d_e} \times [-B, B]^D \times \mathcal{W}(L, W, S, B)$ with constants $L, W, S, B > 0$, and $\phi_k^* \in \Phi$, $\psi^* \in \Psi$. Then, for any $0 < \delta \leq 1/2$, it holds with probability at least $1 - \delta$ that*

$$\mathcal{R}_{\overline{\text{TV}}}(\hat{p}_{X|Y}^{\text{multi}}) = \tilde{\mathcal{O}}\left(\sqrt{\frac{L(S + D + (D + M + K)d_e)}{n}}\right).$$

In contrast, single-source training results in an error of $\mathcal{R}_{\overline{\text{TV}}}(\hat{p}_{X|Y}^{\text{single}}) = \tilde{\mathcal{O}}\left(\sqrt{KL(S+D+(D+M+1)d_e)/n}\right)$ with a formal result provided in Theorem C.8. The advantage of multi-source learning is quantified by the ratio of error bounds: $\sqrt{\frac{L(S+D+(D+M+K)d_e)}{KL(S+D+(D+M+1)d_e)}} = \sqrt{1 - \frac{K-1}{K}\beta_{\text{sim}}}$, where the term $\beta_{\text{sim}} := \frac{S+D+(D+M)d_e}{S+D+(D+M+K)d_e+d_e} \in [0, 1]$ quantifies source distribution similarity based on the proportion of shared parameters. This ratio follows the same pattern discussed in Section 4.1 where the number of sources $K$ and the distribution similarity $\beta_{\text{sim}}$ are two key factors improving the advantage of multi-source training.

## 4.3. Conditional EBMs on continuous distributions

In this section, we study distribution estimation for conditional EBMs, a flexible probabilistic modeling approach on continuous data. Our formulation follows Du & Mordatch (2019) with simplified neural network architecture.

**Probabilistic modeling with energy function.** Consider a common scenario with natural image $X$ flattened and normalized in $[0,1]^D$. The conditional distribution $p_{X|Y}(\boldsymbol{x}|y)$ is factorized with an energy function $u(\boldsymbol{x}|y)$ as:

$$p(\boldsymbol{x}|y) = \frac{e^{-u(\boldsymbol{x}|y)}}{\int_{\mathcal{X}} e^{-u(\boldsymbol{s}|y)} d\boldsymbol{s}}.$$

**Distribution estimation via neural network.** We suppose the energy function is estimated with $u_\theta(\boldsymbol{x}|y)$ using a neural network parameterized by $\theta$, which comprises a condition embedding layer and an energy-estimating MLP.

Specifically, we first look up $y$ in a condition embedding matrix $\boldsymbol{V} \in [0,1]^{K \times d_e}$ and concat the embedding with $\boldsymbol{x}$

$$\boldsymbol{e}_{\boldsymbol{V}}(\boldsymbol{x}, y) = \begin{bmatrix} \boldsymbol{x} \\ \boldsymbol{V}[y,:] \end{bmatrix} \in [0,1]^{D+d_e}.$$

Then we use an MLP $f_\omega \in \mathcal{F}(L, W, S, B)$ with $W_0 = D + d_e$ and $W_L = 1$ to estimate the energy as

$$u_\theta(\boldsymbol{x}|y) = f_\omega\big(\boldsymbol{e}_{\boldsymbol{V}}(\boldsymbol{x}, y)\big),$$

where $\theta := \{\boldsymbol{V}, \omega\}$. This leads to a conditional distribution as

$$p_\theta(\boldsymbol{x}|y) = \frac{e^{-u_\theta(\boldsymbol{x}|y)}}{\int_{\mathcal{X}} e^{-u_\theta(\boldsymbol{s}|y)} d\boldsymbol{s}}. \tag{7}$$

When training such an EBM, each row of $\boldsymbol{V}$ is only optimized on data with the corresponding condition, while $\omega$ is optimized on data with all conditions. That means $\boldsymbol{V}[k,:]$ serves as the source-specific parameter and $\omega$ is shared across all sources. Corresponding to the general formulation in Section 2, we denote

$$\phi_k := \boldsymbol{V}[k,:], \text{ and } \psi := \omega.$$

**Statistical guarantee of the average TV error.** In this formulation, the conditional MLE in $\mathcal{P}_{X|Y}^{\text{multi}}$ under multi-source training leads to the following result.

**Theorem 4.4** (Average TV error bound for EBMs under multi-source training, Proof in Appendix D.3). *Let $\hat{p}_{X|Y}^{\text{multi}}$ be the likelihood maximizer defined in Equation* (2) *given $\mathcal{P}_{X|Y}^{\text{multi}}$ with conditional distributions in Equation* (7). *Suppose $\Phi = [0,1]^{d_e}$ and $\Psi = \mathcal{W}(L, W, S, B)$ with constants*

$L, W, S, B > 0$ *and assume $\phi_k^* \in \Phi$, $\psi^* \in \Psi$. Then, for any $0 < \delta \leq 1/2$, it holds with probability at least $1 - \delta$ that*

$$\mathcal{R}_{\overline{\text{TV}}}(\hat{p}_{X|Y}^{\text{multi}}) = \tilde{\mathcal{O}}\left(\sqrt{\frac{L(S + Kd_e)}{n}}\right).$$

In contrast, single-source training results in an error of $\mathcal{R}_{\overline{\text{TV}}}(\hat{p}_{X|Y}^{\text{single}}) = \tilde{\mathcal{O}}\left(\sqrt{LK(S + d_e)/n}\right)$ with a formal proof provided in Theorem D.4. The advantage of multi-source learning is quantified by the ratio of error bounds: $\sqrt{\frac{L(S+Kd_e)}{LK(S+d_e)}} = \sqrt{1 - \frac{K-1}{K}\beta_{\text{sim}}}$, where $\beta_{\text{sim}} := \frac{S}{S+d_e} \in [0,1]$ quantifies source distribution similarity based on the proportion of shared parameters. Similar to the former two cases, the number of sources $K$ and the distribution similarity $\beta_{\text{sim}}$ improve the advantage of multi-source training.

## 5. Experiments

In this section, simulations and real-world experiments are conducted to verify our theoretical results in Section 3 and 4.

### 5.1. Simulations on conditional Gaussian estimation

In this part, we aim to examine the tightness of the derived upper bound that $\mathcal{R}_{\overline{\text{TV}}}(\hat{p}_{X|Y}^{\text{multi}}) = \tilde{\mathcal{O}}(\sqrt{\frac{(K-1)d_1+d}{n}})$ in Theorem 4.1 and $\mathcal{R}_{\overline{\text{TV}}}(\hat{p}_{X|Y}^{\text{single}}) = \tilde{\mathcal{O}}(\sqrt{\frac{Kd}{n}})$ in Theorem B.2.

The number of sources $K$, sample size $n$, and the similarity factor $\beta_{\text{sim}} \in [0,1]$ are key parameters. In all of our simulations, we fix data dimension $d = 10$ and $p_Y^*(k) = 1/K$ all $k \in [K]$. The dissimilar dimension $d_1 = d - \lfloor \beta_{\text{sim}} d \rfloor$. We set the source-specific feature as $\phi_k = k\boldsymbol{1} \in \mathbb{R}^{d_1}$ and the shared feature as $\psi = \boldsymbol{0} \in \mathbb{R}^{d-d_1}$. Under the setting of Section 4.1, conditional MLE has analytical solutions: under multi-source training, we have

$$\hat{\phi}_k = \sum_{y_i=k} \boldsymbol{x}_i[1:d_1]/n_k, \ \hat{\psi} = \sum_{i=1}^n \boldsymbol{x}_i[d_1+1:d]/n,$$

and under single-source training, we have

$$\hat{\phi}_k = \sum_{y_i=k} \boldsymbol{x}_i[1:d_1]/n_k, \ \hat{\psi}_k = \sum_{y_i=k} \boldsymbol{x}_i[d_1+1:d]/n_k.$$

For evaluation, we randomly sample $n^{\text{test}} = 500$ data points according to the true joint distribution $p_{X,Y}^*$. Empirically, we approximate the true TV distance by using the Monte Carlo method based on the test set, which can be written formally as

$$\mathcal{R}_{\overline{\text{TV}}}(\hat{p}_{X|Y}) \approx \frac{1}{2n^{\text{test}}} \sum_{i=1}^{n^{\text{test}}} \left| \frac{\hat{p}_{X|Y}(\boldsymbol{x}_i|y_i)}{p_{X|Y}^*(\boldsymbol{x}_i|y_i)} - 1 \right| = \mathcal{R}_{\overline{\text{TV}}}^{\text{em}}(\hat{p}_{X|Y}).$$

To eliminate the randomness, we average over 5 random runs for each simulation and report the mean results.

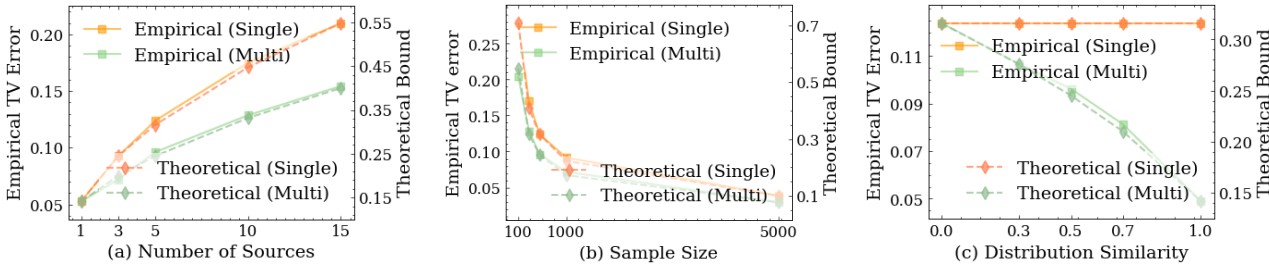

*Figure 1.* Simulation results for conditional Gaussian estimation. Empirical values (solid lines) correspond to the left vertical axis, while theoretical values (dashed lines) correspond to the right. Single-source results are shown in orange, and multi-source results in green. The matched orders of empirical errors and theoretical upper bounds support the validity of results in Section 4.1.

**Order of the average TV error about $K$.** We range the number of sources $K$ in $[1, 3, 5, 10, 15]$ with fixed sample size $n = 500$ and similarity factor $\beta_{\text{sim}} = 0.5$. We display the empirical average TV error for each $K$ in Figure 1(a), with $\mathcal{R}_{\overline{\text{TV}}}^{\text{em}}(\hat{p}_{X|Y}^{\text{multi}})$ colored in green and $\mathcal{R}_{\overline{\text{TV}}}^{\text{em}}(\hat{p}_{X|Y}^{\text{single}})$ colored in orange. Ignoring the influence of constants, it shows a good alignment between empirical errors (in solid lines) and theoretical upper bounds (in dashed lines), both scaling as $\tilde{\mathcal{O}}(\sqrt{K})$.

**Order of the average TV error about $n$.** We range sample size $n$ in $[100, 300, 500, 1000, 5000]$ with fixed number of sources $K = 5$ and similarity factor $\beta_{\text{sim}} = 0.5$. We display the empirical error for each $n$ in Figure 1(b), with $\mathcal{R}_{\overline{\text{TV}}}^{\text{em}}(\hat{p}_{X|Y}^{\text{multi}})$ colored in green and $\mathcal{R}_{\overline{\text{TV}}}^{\text{em}}(\hat{p}_{X|Y}^{\text{single}})$ colored in orange. Ignoring the influence of constants, it shows that the orders of empirical error about $n$ match well with the theoretical upper bounds which scale as $\tilde{\mathcal{O}}(1/\sqrt{n})$.

**Order of the average TV error about $\beta_{\text{sim}}$.** We range similarity factor $\beta_{\text{sim}}$ in $[0, 0.3, 0.5, 0.7, 1]$ with fixed sample size $n = 500$ and number of data sources $K = 5$. We display the empirical average TV error for each $\beta_{\text{sim}}$ in Figure 1(c) to observe how similarity factor $\beta_{\text{sim}}$ impacts the advantage of multi-source training. Concretely, as predicted by the theoretical bounds, the changing of $\beta_{\text{sim}}$ will not influence the performance of single-source training but will decrease the error of multi-source training in the order of $\tilde{\mathcal{O}}(\sqrt{d_1}) = \tilde{\mathcal{O}}(\sqrt{1 - \beta_{\text{sim}}})$. The results show that the theoretical bounds predict the empirical performance well.

To sum up, our simulations verify the validity of our theoretical bounds in Section 4.1. Moreover, in all experiments, $\mathcal{R}_{\overline{\text{TV}}}^{\text{em}}(\hat{p}_{X|Y}^{\text{multi}})$ is consistently smaller than $\mathcal{R}_{\overline{\text{TV}}}^{\text{em}}(\hat{p}_{X|Y}^{\text{single}})$, supporting our results in Section 3

### 5.2. Real-world experiments on diffusion models

In this section, we conduct experiments on diffusion models to validate our theoretical findings in real-world scenarios from two aspects: (1) We empirically compare multi-source and single-source training on conditional diffusion models and evaluate their performance to validate the guaranteed advantage of multi-source training against single-source training proved in Section 3. (2) We investigate the trend of this advantage about key factors—the number of sources and distribution similarity—as discussed in Section 4.

**Experimental settings.** We train class-conditional diffusion models following EDM2 (Karras et al., 2024) at $256 \times 256$ resolution on the selected classes from the ILSVRC2012 training set (Russakovsky et al., 2015), which is a subset of ImageNet (Deng et al., 2009) containing 1.28M natural images from 1000 classes, each annotated with an integer class label from 1 to 1000. In our experiments, we treat each class as a distinct data source. To control similarity among data sources, we manually design two levels of distribution similarity based on the semantic hierarchy of ImageNet (Deng et al., 2010; Bostock., 2018) as shown in Figure 2 in Appendix E.1 along with other experimental details.

For each controlled experiment comparing multi-source and single-source training, we fix $K$ target classes within one similarity level Sim and train the models on a dataset $S$ consisting of $N$ examples per class. Under multi-source training, we train a single conditional diffusion model for all $K$ classes jointly. Under single-source training, we train $K$ separate conditional diffusion models, one for each class. Please refer to Section 2 for the formal formulation of these two strategies. We set each factor with two possible values: the number of classes $K$ in 3 or 10, distribution similarity Sim in 1 or 2, and the sample size per class $N$ in 500 or 1000. This results in a total of 8 sets of experiments comparing multi-source and single-source training.

We evaluate model performance using the average Fréchet Inception Distance (Heusel et al., 2017) (FID, a widely used metric for image generation quality) across all conditions to assess the overall conditional generation performance. Results are displayed in Table 1. Specifically, for multi-source training, we compute the FID for each class and take

*Table 1.* Average FID for single-source and multi-source training. Under different amounts of classes $K$, similarity level Sim, and per-class sample size $N$, multi-source training generally achieves lower average FID than that of single-source training, which is consistent with our theoretical guarantees derived in Section 3.

| $N$ | Sim | $K$ | Avg. FID ↓ (Single) | Avg. FID ↓ (Multi) |
|---|---|---|---|---|
| 500 | 1 | 3 | 30.03 | **29.94** |
| | | 10 | 30.18 | **29.28** |
| | 2 | 3 | 32.69 | **30.69** |
| | | 10 | 30.54 | **28.75** |
| 1000 | 1 | 3 | 28.01 | **26.41** |
| | | 10 | 27.49 | **25.84** |
| | 2 | 3 | 30.58 | **29.35** |
| | | 10 | 29.01 | **27.81** |

the average over all $K$ classes. For single-source training, we compute the FID for each of the $K$ separately trained models on their respective classes and calculate the average. Relative advantage of multi-source training is measured by $\frac{\text{Avg. FID (Single)} - \text{Avg. FID (Multi)}}{\text{Avg. FID (Single)}}$ as displayed in Figure 3.

**Experimental results** In the following, we interpret the results sequentially from the view of our theoretical findings.

From Table 1, we observe that under different amounts of classes $K$, similarity level Sim, and per-class sample size $N$, multi-source training generally achieves lower average FID than that of single-source training, which is consistent with our theoretical guarantees derived in Section 3,

From Figure 3, we observe that for any fixed similarity level Sim and per-class sample size $N$, the relative advantage of multi-sources training with a larger $K$ (the green bars) is larger than that with a smaller $K$ (the nearby orange bars). Additionally, for any fixed $K$ and $N$, the relative advantage of multi-sources training with a larger distribution similarity is larger than that with a smaller distribution similarity (as shown through the dashed lines). These results support our theoretical insights in Section 4 that the number of sources and similarity among source distributions improves the advantage of multi-source training.

## 6. Other related works

**Distribution estimation guarantee for MLE.** Classical approaches investigate distribution estimation for MLE in Hellinger distance based on the bracketing number and the uniform law of large numbers from empirical process theory (Wong & Shen, 1995; Geer, 2000), which yields high-probability bounds of similar order as Theorem 3.2. Ge et al. (2024) extend the analysis to derive TV error bound under

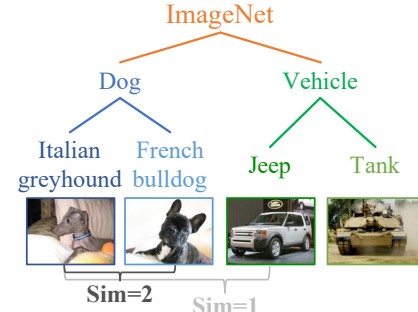

*Figure 2.* Similarity level in ILSVRC2012 training set.

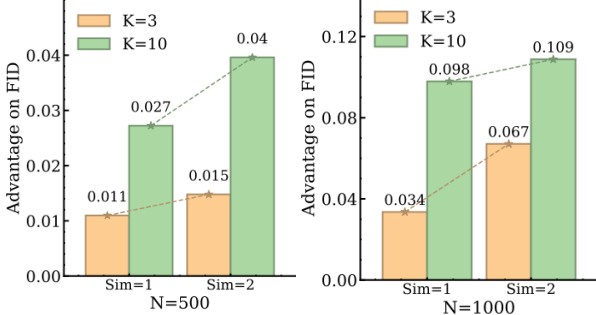

*Figure 3.* Relative advantage of multi-source training. For any fixed similarity level Sim and per-class sample size $N$, a larger $K$ yields a greater FID improvement than a smaller $K$. For any fixed $K$ and $N$, higher distribution similarity leads to greater FID improvement (illustrated by dashed lines). These results support the theoretical findings in Section 4.

the realizable assumption. We further adapt their techniques to conditional generative modeling by introducing the upper bracketing number to quantify the complexity of conditional distribution space in Definition 3.1 and modify the proofs to handle conditional MLE in Appendix A.1.

**Theory on multi-task learning.** Multi-task learning is a well-studied topic in supervised learning (Caruana, 1997; Baxter, 2000). It typically benefits from similarities across tasks, sharing some commonality with multi-source training. However, theoretical analyses in supervised learning often assume a bounded objective (Ben-David & Borbely, 2008; Maurer et al., 2016; Tripuraneni et al., 2020), whereas our MLE analysis imposes no such restriction.

**Advanced theory on generative models.** Among generative models based on (approximate) MLE (LeCun et al., 2006; Uria et al., 2016; Ho et al., 2020), diffusion models have been extensively studied theoretically on its score approximation, sampling behavior, distribution estimation, and scalability (Oko et al., 2023; Chen et al., 2023a;b; Fu et al., 2024; Zheng et al., 2025). This paper focuses on distribution estimation for general conditional generative modeling. Incorporating existing literature could be a promising

direction for future work.

## 7. Conclusion and discussion

This paper provides the first attempt to rigorously analyze the conditional generative modeling on multiple data sources from a distribution estimation perspective. In particular, we establish a general estimation error bound in average TV distance under the realizable assumption based on the bracketing number of the conditional distribution space. When source distributions share parametric similarities, multi-source training has a provable advantage against single-source training by reducing the bracketing number. We further instantiate the general theory on three specific models to obtain concrete error bounds. To achieve this, novel bracketing number bounds for ARMs and EBMs are established. The results show that the number of data sources and the similarity between source distributions enhance the benefits of multi-source training. Simulations and real-world experiments support our theoretical findings.

Our theoretical setting differs from practice in some aspects, e.g., language models have no explicit conditions, and image generation models are commonly conditioned on descriptive text involving multiple conditions. However, our abstraction provides a simplified framework that preserves the core properties of multi-source training and isolates how individual source distributions are learned. Moreover, recent studies suggest adding source labels, such as domain names, at the start of training text for language models can enhance performance (Allen-Zhu & Li, 2024b; Gao et al., 2025), which may become a standard practice in the future.

## Acknowledgments

This work was supported by NSF of China (Nos. 92470118, 62206159); Beijing Nova Program (20220484044); Beijing Natural Science Foundation (L247030); Major Innovation & Planning Interdisciplinary Platform for the "Double-First Class" Initiative, Renmin University of China; the Fundamental Research Funds for the Central Universities, and the Research Funds of Renmin University of China (22XNKJ13); the Natural Science Foundation of Shandong Province (ZR2022QF117), the Fundamental Research Funds of Shandong University. G. Wu was also sponsored by the TaiShan Scholars Program (NO.tsqn202306051).

## Impact Statement

This paper presents work whose goal is to advance the field of Machine Learning. There are many potential societal consequences of our work, none of which we feel must be specifically highlighted here.

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

# A. Proofs for Section 3

## A.1. Proof of Theorem 3.2

*Proof of Theorem 3.2.* This theorem applies to both discrete and continuous random variables, while we use integration notation in the proof for generality. In the following, we first present an elementary inequality (in Equation (10)) which serves as a toolkit for the subsequent derivations. Then we decompose the TV distance and derive its complexity-based upper bound (in Equation (12)) using the former inequality. Finally, after specifying certain constants in this upper bound, a clearer order w.r.t. $n$ is revealed (in Equation (13)).

**Intermediate result induced by union bound.** Let $\epsilon$ be a real number that $\epsilon > 0$ and $\mathsf{p}$ be an integer that $1 \leq \mathsf{p} \leq \infty$. Let $\mathcal{B}$ be an $\epsilon$-upper bracket of $\mathcal{P}_{X|Y}$ w.r.t. $L^1(\mathcal{X})$ such that $|\mathcal{B}| = \mathcal{N}_{[]}\left(\epsilon; \mathcal{P}_{X|Y}, L^1(\mathcal{X})\right)$.

According to the minimum cardinality requirement, we obtain a proposition of $\mathcal{B}$ that: for any $p' \in \mathcal{B}$, $p'(\boldsymbol{x}, y) \geq 0$ on $\mathcal{X} \times \mathcal{Y}$. Let's first consider $\prod_{i=1}^{n} \sqrt{\frac{p'(\boldsymbol{x}_i, y_i)}{p_{X|Y}^*(\boldsymbol{x}_i|y_i)}}$ as a random variable on $S$, where we suppose $p_{X,Y}^*(\boldsymbol{x}_i, y_i) > 0$ since $(\boldsymbol{x}_i, y_i)$ are sampled from $p_{X,Y}^*$ and thus $p_{X|Y}^*(\boldsymbol{x}_i|y_i) \neq 0$. By applying the *Markov inequality*, we have: given any $0 < \delta' < 1$,

$$\Pr_S\left(\prod_{i=1}^{n} \sqrt{\frac{p'(\boldsymbol{x}_i, y_i)}{p_{X|Y}^*(\boldsymbol{x}_i|y_i)}} \geq \frac{1}{\delta'}\mathbb{E}_S\left[\prod_{i=1}^{n} \sqrt{\frac{p'(\boldsymbol{x}_i, y_i)}{p_{X|Y}^*(\boldsymbol{x}_i|y_i)}}\right]\right) \leq \delta'. \tag{8}$$

Applying the *union bound* on all $p' \in \mathcal{B}$, we further have:

$$\Pr_S\left(\forall p' \in \mathcal{B}, \prod_{i=1}^{n} \sqrt{\frac{p'(\boldsymbol{x}_i, y_i)}{p_{X|Y}^*(\boldsymbol{x}_i|y_i)}} < \frac{1}{\delta'}\mathbb{E}_S\left[\prod_{i=1}^{n} \sqrt{\frac{p'(\boldsymbol{x}_i, y_i)}{p_{X|Y}^*(\boldsymbol{x}_i|y_i)}}\right]\right)$$

$$= 1 - \Pr_S\left(\exists p' \in \mathcal{B}, \prod_{i=1}^{n} \sqrt{\frac{p'(\boldsymbol{x}_i, y_i)}{p_{X|Y}^*(\boldsymbol{x}_i|y_i)}} \geq \frac{1}{\delta'}\mathbb{E}_S\left[\prod_{i=1}^{n} \sqrt{\frac{p'(\boldsymbol{x}_i, y_i)}{p_{X|Y}^*(\boldsymbol{x}_i|y_i)}}\right]\right)$$

$$= 1 - \Pr_S\left(\bigcup_{p' \in \mathcal{B}}\left\{\prod_{i=1}^{n} \sqrt{\frac{p'(\boldsymbol{x}_i, y_i)}{p_{X|Y}^*(\boldsymbol{x}_i|y_i)}} \geq \frac{1}{\delta'}\mathbb{E}_S\left[\prod_{i=1}^{n} \sqrt{\frac{p'(\boldsymbol{x}_i, y_i)}{p_{X|Y}^*(\boldsymbol{x}_i|y_i)}}\right]\right\}\right)$$

$$\geq 1 - \sum_{p' \in \mathcal{B}} \Pr_S\left(\prod_{i=1}^{n} \sqrt{\frac{p'(\boldsymbol{x}_i, y_i)}{p_{X|Y}^*(\boldsymbol{x}_i|y_i)}} \geq \frac{1}{\delta'}\mathbb{E}_S\left[\prod_{i=1}^{n} \sqrt{\frac{p'(\boldsymbol{x}_i, y_i)}{p_{X|Y}^*(\boldsymbol{x}_i|y_i)}}\right]\right) \qquad \text{(by \textit{union bound})}$$

$$\geq 1 - \mathcal{N}_{[]}\left(\epsilon; \mathcal{P}_{X|Y}, L^1(\mathcal{X})\right)\delta'. \qquad \text{(by Equation (8))}$$

By denoting that $\delta := \mathcal{N}_{[]}\left(\epsilon; \mathcal{P}_{X|Y}, L^1(\mathcal{X})\right)\delta'$, we have: it holds with probability at least $1 - \delta$ that for all $p' \in \mathcal{B}$,

$$\prod_{i=1}^{n} \sqrt{\frac{p'(\boldsymbol{x}_i, y_i)}{p_{X|Y}^*(\boldsymbol{x}_i|y_i)}} < \frac{\mathcal{N}_{[]}\left(\epsilon; \mathcal{P}_{X|Y}, L^1(\mathcal{X})\right)}{\delta}\mathbb{E}_S\left[\prod_{i=1}^{n} \sqrt{\frac{p'(\boldsymbol{x}_i, y_i)}{p_{X|Y}^*(\boldsymbol{x}_i|y_i)}}\right].$$

Taking logarithms at both sides, we have

$$\frac{1}{2}\sum_{i=1}^{n}\log\frac{p'(\boldsymbol{x}_i,y_i)}{p_{X|Y}^*(\boldsymbol{x}_i|y_i)} \le \log\mathbb{E}_S\left[\prod_{i=1}^{n}\sqrt{\frac{p'(\boldsymbol{x}_i,y_i)}{p_{X|Y}^*(\boldsymbol{x}_i|y_i)}}\right] + \log\frac{\mathcal{N}_{[]}\left(\epsilon;\mathcal{P}_{X|Y},L^1(\mathcal{X})\right)}{\delta}$$

$$= \log\prod_{i=1}^{n}\mathbb{E}_{(\boldsymbol{x}_i,y_i)\sim p_{X,Y}^*}\left[\sqrt{\frac{p'(\boldsymbol{x}_i,y_i)}{p_{X|Y}^*(\boldsymbol{x}_i|y_i)}}\right] + \log\frac{\mathcal{N}_{[]}\left(\epsilon;\mathcal{P}_{X|Y},L^1(\mathcal{X})\right)}{\delta}$$

$$(\{x_i\}_{i=1}^n \text{ are i.i.d. sampled from } p_X^*)$$

$$= n\log\mathbb{E}_{X,Y}\left[\sqrt{\frac{p'(\boldsymbol{x},y)}{p_{X|Y}^*(\boldsymbol{x}|y)}}\right] + \log\frac{\mathcal{N}_{[]}\left(\epsilon;\mathcal{P}_{X|Y},L^1(\mathcal{X})\right)}{\delta}$$

$$= n\log\mathbb{E}_Y\left[\mathbb{E}_{X|Y}\left[\sqrt{\frac{p'(\boldsymbol{x},y)}{p_{X|Y}^*(\boldsymbol{x}|y)}}\right]\right] + \log\frac{\mathcal{N}_{[]}\left(\epsilon;\mathcal{P}_{X|Y},L^1(\mathcal{X})\right)}{\delta}$$

$$= n\log\mathbb{E}_Y\left[\int_{\mathcal{X}} p_{X|Y}^*(\boldsymbol{x}|y)\sqrt{\frac{p'(\boldsymbol{x},y)}{p_{X|Y}^*(\boldsymbol{x}|y)}}d\boldsymbol{x}\right] + \log\frac{\mathcal{N}_{[]}\left(\epsilon;\mathcal{P}_{X|Y},L^1(\mathcal{X})\right)}{\delta}$$

$$= n\log\mathbb{E}_Y\left[\int_{\mathcal{X}} \sqrt{p'(\boldsymbol{x},y)p_{X|Y}^*(\boldsymbol{x}|y)}d\boldsymbol{x}\right] + \log\frac{\mathcal{N}_{[]}\left(\epsilon;\mathcal{P}_{X|Y},L^1(\mathcal{X})\right)}{\delta}.$$

As $\log x \le x - 1$ for all $x > 0$, the inequality can be further transformed into

$$\frac{1}{2}\sum_{i=1}^{n}\log\frac{p'(\boldsymbol{x}_i,y_i)}{p_{X|Y}^*(\boldsymbol{x}_i|y_i)} \le n\left(\mathbb{E}_Y\left[\int_{\mathcal{X}}\sqrt{p'(\boldsymbol{x},y)p_{X|Y}^*(\boldsymbol{x}|y)}d\boldsymbol{x}\right] - 1\right) + \log\frac{\mathcal{N}_{[]}\left(\epsilon;\mathcal{P}_{X|Y},L^1(\mathcal{X})\right)}{\delta}. \tag{9}$$

**Elementary inequality for MLE estimators.** Since the real conditional distribution $p_{X|Y}^*$ is in $\mathcal{P}_{X|Y}$, for the likelihood maximizers $\hat{p}_{X|Y} \in \mathcal{P}_{X|Y}$, we have $L_S(\hat{p}_{X|Y}) = \prod_{i=1}^{n}\hat{p}_{X|Y}(\boldsymbol{x}_i|y_i) \ge L_S(p_{X|Y}^*) = \prod_{i=1}^{n}p_{X|Y}^*(\boldsymbol{x}_i|y_i)$, and thus $\frac{1}{2}\sum_{i=1}^{n}\log\frac{\hat{p}_{X|Y}(\boldsymbol{x}_i|y_i)}{p_{X|Y}^*(\boldsymbol{x}_i|y_i)} = \frac{1}{2}\log\frac{\prod_{i=1}^{n}\hat{p}_{X|Y}(\boldsymbol{x}_i|y_i)}{\prod_{i=1}^{n}p_{X|Y}^*(\boldsymbol{x}_i|y_i)} \ge \frac{1}{2}\log 1 = 0$. According to the definition of upper bracketing number, there exists some $\hat{p}' \in \mathcal{B}$ such that given any $y \in \mathcal{Y}$, it holds that: (i) $\forall x \in \mathcal{X}, \hat{p}'(\boldsymbol{x},y) \ge \hat{p}_{X|Y}(\boldsymbol{x}|y)$, and (ii) $\|\hat{p}'(\cdot,y) - \hat{p}_{X|Y}(\cdot|y)\|_{L^1(\mathcal{X})} = \int_{\mathcal{X}}|\hat{p}'(\boldsymbol{x},y) - \hat{p}_{X|Y}(\boldsymbol{x}|y)|d\boldsymbol{x} \le \epsilon$. Applying (i), we have:

$$\frac{1}{2}\sum_{i=1}^{n}\log\frac{\hat{p}'(\boldsymbol{x}_i,y_i)}{p_{X|Y}^*(\boldsymbol{x}_i|y_i)} \ge \frac{1}{2}\sum_{i=1}^{n}\log\frac{\hat{p}_{X|Y}(\boldsymbol{x}_i|y_i)}{p_{X|Y}^*(\boldsymbol{x}_i|y_i)} \ge 0.$$

Combining this with Equation (9) and rearranging the terms, we have: it holds with at least probability $1 - \delta$ that

$$1 - \mathbb{E}_Y\left[\int_{\mathcal{X}}\sqrt{p'(\boldsymbol{x},y)p_{X|Y}^*(\boldsymbol{x}|y)}d\boldsymbol{x}\right] \le \frac{1}{n}\log\frac{\mathcal{N}_{[]}\left(\epsilon;\mathcal{P}_{X|Y},L^1(\mathcal{X})\right)}{\delta}. \tag{10}$$

This serves as an elementary toolkit for deriving the subsequent upper bounds.

**Decomposing the square of the TV distance.** Recalling that $\mathrm{TV}(\hat{p}_{X|Y},p_{X|Y}^*) = \frac{1}{2}\int_{\mathcal{X}}|\hat{p}_{X|Y}(\boldsymbol{x}|y) - p_{X|Y}^*(\boldsymbol{x}|y)|d\boldsymbol{x}$, we will decompose its square and then bound each term sequentially. First, we use the above $\hat{p}'(\boldsymbol{x},y)$ as an intermediate term to decompose the square of $2\mathrm{TV}(\hat{p}_{X|Y},p_{X|Y}^*)$ into parts that can be effectively upper bounded:

$$\left(2\mathrm{TV}(\hat{p}_{X|Y},p_{X|Y}^*)\right)^2 = \left(\int_{\mathcal{X}}|\hat{p}_{X|Y}(\boldsymbol{x}|y) - p_{X|Y}^*(\boldsymbol{x}|y)|d\boldsymbol{x}\right)^2$$

$$= \underbrace{\left(\int_{\mathcal{X}}|\hat{p}_{X|Y}(\boldsymbol{x}|y) - p_{X|Y}^*(\boldsymbol{x}|y)|d\boldsymbol{x}\right)^2 - \left(\int_{\mathcal{X}}|\hat{p}'(\boldsymbol{x},y) - p_{X|Y}^*(\boldsymbol{x}|y)|d\boldsymbol{x}\right)^2}_{(\mathrm{I})} + \underbrace{\left(\int_{\mathcal{X}}|\hat{p}'(\boldsymbol{x},y) - p_{X|Y}^*(\boldsymbol{x}|y)|d\boldsymbol{x}\right)^2}_{(\mathrm{II})}.$$

For (I), we have

$$
\left(\int_{\mathcal{X}}|\hat{p}_{X|Y}(\boldsymbol{x}|y)-p^*_{X|Y}(\boldsymbol{x}|y)|d\boldsymbol{x}\right)^2-\left(\int_{\mathcal{X}}|\hat{p}'(\boldsymbol{x},y)-p^*_{X|Y}(\boldsymbol{x}|y)|d\boldsymbol{x}\right)^2
$$
$$
=\left(\int_{\mathcal{X}}|\hat{p}_{X|Y}(\boldsymbol{x}|y)-p^*_{X|Y}(\boldsymbol{x}|y)|+|\hat{p}'(\boldsymbol{x},y)-p^*_{X|Y}(\boldsymbol{x}|y)|d\boldsymbol{x}\right)\left(\int_{\mathcal{X}}|\hat{p}_{X|Y}(\boldsymbol{x}|y)-p^*_{X|Y}(\boldsymbol{x}|y)|-|\hat{p}'(\boldsymbol{x},y)-p^*_{X|Y}(\boldsymbol{x}|y)|d\boldsymbol{x}\right)
$$
$$
\leq\left(\int_{\mathcal{X}}|\hat{p}_{X|Y}(\boldsymbol{x}|y)|+|p^*_{X|Y}(\boldsymbol{x}|y)|+|\hat{p}'(\boldsymbol{x},y)-\hat{p}_{X|Y}(\boldsymbol{x}|y)|+|\hat{p}_{X|Y}(\boldsymbol{x}|y)|+|p^*_{X|Y}(\boldsymbol{x}|y)|d\boldsymbol{x}\right)\left(\int_{\mathcal{X}}|\hat{p}_{X|Y}(\boldsymbol{x}|y)-\hat{p}'(\boldsymbol{x},y)|d\boldsymbol{x}\right)
$$
$$
\leq(\epsilon+4)\epsilon.
$$

The first inequality holds for the *triangle inequality* $|a+b|\leq|a|+|b|$ and the *reverse triangle inequality* $||a|-|b||\leq|a-b|$. The second inequality holds for the normalization property of conditional distributions ($\int_{\mathcal{X}}|\hat{p}_{X|Y}(\boldsymbol{x}|y)|d\boldsymbol{x}$ and $\int_{\mathcal{X}}|p^*_{X|Y}(\boldsymbol{x}|y)|d\boldsymbol{x}$ equal 1) and the property of the $\epsilon$-upper bracket ($\int_{\mathcal{X}}|\hat{p}'(\boldsymbol{x},y)-\hat{p}_{X|Y}(\boldsymbol{x}|y)|d\boldsymbol{x}\leq\epsilon$).

For (II), we have

$$
\left(\int_{\mathcal{X}}|\hat{p}'(\boldsymbol{x},y)-p^*_{X|Y}(\boldsymbol{x}|y)|d\boldsymbol{x}\right)^2
$$
$$
\leq\left(\int_{\mathcal{X}}\left(\sqrt{\hat{p}'(\boldsymbol{x},y)}+\sqrt{p^*_{X|Y}(\boldsymbol{x}|y)}\right)^2dx\right)\left(\int_{\mathcal{X}}\left(\sqrt{\hat{p}'(\boldsymbol{x},y)}-\sqrt{p^*_{X|Y}(\boldsymbol{x}|y)}\right)^2dx\right)\quad\text{(by \textit{Cauchy–Schwarz inequality})}
$$
$$
\leq\left(\int_{\mathcal{X}}2\left(\hat{p}'(\boldsymbol{x},y)+p^*_{X|Y}(\boldsymbol{x}|y)\right)d\boldsymbol{x}\right)\left(\int_{\mathcal{X}}\hat{p}'(\boldsymbol{x},y)+p^*_{X|Y}(\boldsymbol{x}|y)-2\sqrt{\hat{p}'(\boldsymbol{x},y)p^*_{X|Y}(\boldsymbol{x}|y)}d\boldsymbol{x}\right)
$$
$$
\text{(by }(a+b)^2\leq2(a^2+b^2))
$$
$$
=2\left(\int_{\mathcal{X}}\hat{p}'(\boldsymbol{x},y)-\hat{p}_{X|Y}(\boldsymbol{x}|y)+\hat{p}_{X|Y}(\boldsymbol{x}|y)+p^*_{X|Y}(\boldsymbol{x}|y)d\boldsymbol{x}\right)
$$
$$
\left(\int_{\mathcal{X}}\hat{p}'(\boldsymbol{x},y)-\hat{p}_{X|Y}(\boldsymbol{x}|y)+\hat{p}_{X|Y}(\boldsymbol{x}|y)+p^*_{X|Y}(\boldsymbol{x}|y)-2\sqrt{\hat{p}'(\boldsymbol{x},y)p^*_{X|Y}(\boldsymbol{x}|y)}d\boldsymbol{x}\right)
$$
$$
\leq2(\epsilon+2)\left(\epsilon+2-2\int_{\mathcal{X}}\sqrt{\hat{p}'(\boldsymbol{x},y)p^*_{X|Y}(\boldsymbol{x}|y)}d\boldsymbol{x}\right).
$$
$$
\text{(by }\int_{\mathcal{X}}|\hat{p}_{X|Y}(\boldsymbol{x}|y)|d\boldsymbol{x}=\int_{\mathcal{X}}|p^*_{X|Y}(\boldsymbol{x}|y)|d\boldsymbol{x}=1\text{ and }\int_{\mathcal{X}}|\hat{p}'(\boldsymbol{x},y)-\hat{p}_{X|Y}(\boldsymbol{x}|y)|d\boldsymbol{x}\leq\epsilon)
$$

Putting together (I) and (II), we get:

$$
\mathrm{TV}(\hat{p}_{X|Y},p^*_{X|Y})=\frac{1}{2}\sqrt{\left(\int_{\mathcal{X}}|\hat{p}_{X|Y}(\boldsymbol{x}|y)-p^*_{X|Y}(\boldsymbol{x}|y)|d\boldsymbol{x}\right)^2}
$$
$$
\leq\frac{1}{2}\sqrt{(\epsilon+4)\epsilon+2(\epsilon+2)\left(\epsilon+2-2\int_{\mathcal{X}}\sqrt{\hat{p}'(\boldsymbol{x},y)p^*_{X|Y}(\boldsymbol{x}|y)}d\boldsymbol{x}\right)}. \tag{11}
$$

**Bounding the average TV error.** Based on the above results, we upper bound the average TV error (defined in Equation (4)) of $\hat{p}_{X|Y}$ as follows:

$$
\begin{aligned}
\mathcal{R}_{\overline{\mathrm{TV}}}(\hat{p}_{X|Y}) =& \mathbb{E}_Y\left[\mathrm{TV}(\hat{p}_{X|Y}, p^*_{X|Y})\right] \\
\leq& \frac{1}{2}\mathbb{E}_Y\left[\sqrt{(\epsilon+4)\epsilon + 2(\epsilon+2)\left(\epsilon+2-2\int_{\mathcal{X}}\sqrt{\hat{p}'(\boldsymbol{x},y)p^*_{X|Y}(\boldsymbol{x}|y)}d\boldsymbol{x}\right)}\right] \quad \text{(by Equation (11))} \\
\leq& \frac{1}{2}\sqrt{\mathbb{E}_Y\left[(\epsilon+4)\epsilon + 2(\epsilon+2)\left(\epsilon+2-2\int_{\mathcal{X}}\sqrt{\hat{p}'(\boldsymbol{x},y)p^*_{X|Y}(\boldsymbol{x}|y)}d\boldsymbol{x}\right)\right]} \\
& \hspace{3cm} \text{(by concavity of } f(x) = \sqrt{x} \text{ and \textit{Jensen's inequality})} \\
=& \frac{1}{2}\sqrt{(\epsilon+4)\epsilon + 2(\epsilon+2)\left(\epsilon+2\left(1-\mathbb{E}_Y\left[\int_{\mathcal{X}}\sqrt{\hat{p}'(\boldsymbol{x},y)p^*_{X|Y}(\boldsymbol{x}|y)}d\boldsymbol{x}\right]\right)\right)}. \\
& \hspace{4cm} \text{(by the linearity of expectation)}
\end{aligned}
$$

Recalling the elementary inequality we derived formerly in Equation (10), we have: it holds with at least probability $1-\delta$ that

$$
\mathcal{R}_{\overline{\mathrm{TV}}}(\hat{p}_{X|Y}) \leq \frac{1}{2}\sqrt{(\epsilon+4)\epsilon + 2(\epsilon+2)\left(\epsilon + \frac{2}{n}\log\frac{\mathcal{N}_{[]}\left(\epsilon; \mathcal{P}_{X|Y}, L^1(\mathcal{X})\right)}{\delta}\right)}. \tag{12}
$$

Recalling that $0 \leq \delta \leq \frac{1}{2}$ and for non-empty $\mathcal{P}_{X|Y}$, $\mathcal{N}_{[]}\left(\epsilon; \mathcal{P}_{X|Y}, L^1(\mathcal{X})\right) \geq 1$, we have $\mathcal{N}_{[]}\left(\epsilon; \mathcal{P}_{X|Y}, L^1(\mathcal{X})\right)/\delta \geq 2 \geq e^{\frac{1}{2}}$. Taking $\epsilon = 1/n$ in Equation (12), it then holds with probability at least $1-\delta$ that

$$
\begin{aligned}
\mathcal{R}_{\overline{\mathrm{TV}}}(\hat{p}_{X|Y}) \leq& \frac{1}{2}\sqrt{(\frac{1}{n}+4)\frac{1}{n} + 2(\frac{1}{n}+2)\left(\frac{1}{n} + \frac{2}{n}\log\frac{\mathcal{N}_{[]}\left(\frac{1}{n}; \mathcal{P}_{X|Y}, L^1(\mathcal{X})\right)}{\delta}\right)} \\
\leq& \frac{1}{2}\sqrt{\frac{5}{n} + 6\left(\frac{1}{n} + \frac{2}{n}\log\frac{\mathcal{N}_{[]}\left(\frac{1}{n}; \mathcal{P}_{X|Y}, L^1(\mathcal{X})\right)}{\delta}\right)} \quad \text{(by } \frac{1}{n} \leq 1\text{)} \\
\leq& \frac{1}{2}\sqrt{\frac{10}{n}\log\frac{\mathcal{N}_{[]}\left(\frac{1}{n}; \mathcal{P}_{X|Y}, L^1(\mathcal{X})\right)}{\delta} + 6\left(\frac{4}{n}\log\frac{\mathcal{N}_{[]}\left(\frac{1}{n}; \mathcal{P}_{X|Y}, L^1(\mathcal{X})\right)}{\delta}\right)} \\
& \hspace{4cm} \text{(by } \frac{1}{n} \leq \frac{2}{n}\log\frac{\mathcal{N}_{[]}\left(\frac{1}{n}; \mathcal{P}_{X|Y}, L^1(\mathcal{X})\right)}{\delta}\text{)} \\
=& \frac{1}{2}\sqrt{\frac{34}{n}\log\frac{\mathcal{N}_{[]}\left(\frac{1}{n}; \mathcal{P}_{X|Y}, L^1(\mathcal{X})\right)}{\delta}} \leq 3\sqrt{\frac{1}{n}\log\frac{\mathcal{N}_{[]}\left(\frac{1}{n}; \mathcal{P}_{X|Y}, L^1(\mathcal{X})\right)}{\delta}} \\
=& 3\sqrt{\frac{1}{n}\left(\log\mathcal{N}_{[]}\left(\frac{1}{n}; \mathcal{P}_{X|Y}, L^1(\mathcal{X})\right) + \log\frac{1}{\delta}\right)}. \tag{13}
\end{aligned}
$$

Until now, we have completed the proof of this theorem.

$\square$

### A.2. Proof of Proposition 3.3

*Proof of Proposition 3.3.* As defined in Section 2, it holds that $\mathcal{P}_{X|Y}^{\text{multi}} \subset \mathcal{P}_{X|Y}^{\text{single}}$. Then, for any $p_{X|Y}^{\text{multi}} \in \mathcal{P}_{X|Y}^{\text{multi}}$, there exists some $p_{X|Y}^{\text{single}} \in \mathcal{P}_{X|Y}^{\text{single}}$ such that $p_{X|Y}^{\text{single}} = p_{X|Y}^{\text{multi}}$. Given any $\epsilon > 0$ and $1 \leq \mathsf{p} \leq \infty$, let $\mathcal{B}^{\text{single}}$ be a $\epsilon$-upper bracket w.r.t. $L^{\mathsf{p}}(\mathcal{X})$ for $\mathcal{P}_{X|Y}^{\text{single}}$ such that $|\mathcal{B}^{\text{single}}| = \mathcal{N}_{[]}\left(\epsilon; \mathcal{P}_{X|Y}^{\text{single}}, L^{\mathsf{p}}(\mathcal{X})\right)$. According to the definition of $\epsilon$-upper bracket (as in Definition 3.1), there exists some $p' \in \mathcal{B}^{\text{single}}$ such that given any $y \in \mathcal{Y}$, it holds that: $\forall \boldsymbol{x} \in \mathcal{X}, p'(\boldsymbol{x}, y) \geq p_{X|Y}^{\text{single}}(\boldsymbol{x}|y) = p_{X|Y}^{\text{multi}}(\boldsymbol{x}|y)$, and $\|p'(\cdot, y) - p_{X|Y}^{\text{multi}}(\cdot|y)\|_{L^{\mathsf{p}}(\mathcal{X})} = \|p'(\cdot, y) - p_{X|Y}^{\text{single}}(\cdot|y)\|_{L^{\mathsf{p}}(\mathcal{X})} \leq \epsilon$. Therefore, $\mathcal{B}^{\text{single}}$ is also a $\epsilon$-upper bracket w.r.t. $L^{\mathsf{p}}(\mathcal{X})$ for $\mathcal{P}_{X|Y}^{\text{multi}}$, and thus $\mathcal{N}_{[]}\left(\epsilon; \mathcal{P}_{X|Y}^{\text{multi}}, L^{\mathsf{p}}(\mathcal{X})\right) \leq |\mathcal{B}^{\text{single}}| = \mathcal{N}_{[]}\left(\epsilon; \mathcal{P}_{X|Y}^{\text{single}}, L^{\mathsf{p}}(\mathcal{X})\right)$. $\qquad\square$

## B. Proofs for Section 4.1

### B.1. Bracketing number of conditional Gaussian distribution space

According to Theorem 3.2, to derive the upper bound of average TV error, we need to measure the upper bracketing number for the conditional Gaussian distribution space. This result mainly follows the bracketing number analysis of Gaussian distribution space in Lemma C.5 in (Ge et al., 2024), and slightly modifies it to conditional Gaussian distribution space.

**Theorem B.1** (Bracketing number upper bound for conditional Gaussian distribution space under multi-source training). *Let $B$ be a constant that $0 < B < \infty$, suppose that $\Phi = [-B, B]^{d_1}$, $\Psi = [-B, B]^{d-d_1}$, and conditional distributions in $\mathcal{P}_{X|Y}^{\text{multi}}$ are formulated as in Equation* (5). *Then, given any $0 < \epsilon \leq 1$, the $\epsilon$-upper bracketing number of $\mathcal{P}_{X|Y}^{\text{multi}}$ w.r.t. $L^1(\mathcal{X})$ satisfies*

$$\mathcal{N}_{[]}\left(\epsilon; \mathcal{P}_{X|Y}^{\text{multi}}, L^1(\mathcal{X})\right) \leq \left(\frac{2(1+d)B}{\epsilon} + 1\right)^{(K-1)d_1 + d}.$$

*Proof.* According to the assumptions, the conditional distribution space expressed by the parametric estimation model is

$$\mathcal{P}_{X|Y}^{\text{multi}} := \left\{ p_{X|Y}^{\text{multi}}(\boldsymbol{x}|y) = \prod_{k=1}^{K}\left(p_{\phi_k, \psi}(\boldsymbol{x}|k)\right)^{\mathbb{I}(y=k)} = \prod_{k=1}^{K}\left((2\pi)^{-\frac{d}{2}} e^{-\frac{1}{2}\|\boldsymbol{x} - (\phi_k, \psi)\|_2^2}\right)^{\mathbb{I}(y=k)} : \phi_k \in [-B, B]^{d_1}, \psi \in [-B, B]^{d-d_1} \right\}.$$

For any $p_{X|Y}^{\text{multi}}(\boldsymbol{x}|y) = \prod_{k=1}^{K}\left((2\pi)^{-\frac{d}{2}} e^{-\frac{1}{2}\|\boldsymbol{x} - (\phi_k, \psi)\|_2^2}\right)^{\mathbb{I}(y=k)} \in \mathcal{P}_{X|Y}^{\text{multi}}$, let's first divide the mean vector $(\phi_k, \psi)$ into $\eta$-width grids with a small constant $\eta > 0$ (the value of $\eta$ will be specified later): If $(\phi_k)_i \in [j\eta, (j+1)\eta)$ for some $j \in \mathbb{Z}$, let $(\bar{\phi}_k)_i = j\eta$ and $\bar{\phi}_k := \left((\bar{\phi}_k)_1, \ldots, (\bar{\phi}_k)_{d_1}\right)$. Similarly, if $(\psi)_i \in [j\eta, (j+1)\eta)$ for some $j \in \mathbb{Z}$, let $(\bar{\psi})_i = j\eta$ and $\bar{\psi} := \left((\bar{\psi})_1, \ldots, (\bar{\psi})_{d-d_1}\right)$. In this case, we have $\|(\phi_k, \psi) - (\bar{\phi}_k, \bar{\psi})\|_2^2 \leq d\eta^2$.

Let

$$p'(\boldsymbol{x}, y) = \prod_{k=1}^{K}\left((2\pi)^{-\frac{d}{2}} e^{-\frac{c_1}{2}\|\boldsymbol{x} - (\bar{\phi}_k, \bar{\psi})\|_2^2 + c_2}\right)^{\mathbb{I}(y=k)}.$$

According to the definition of the bracketing, we want to prove that $p'(\boldsymbol{x}, y) \geq p_{X|Y}^{\text{multi}}(\boldsymbol{x}|y)$. By completing the square w.r.t. $\boldsymbol{x}$, we have

$$-\frac{c_1}{2}\|\boldsymbol{x} - (\bar{\phi}_k, \bar{\psi})\|_2^2 + c_2 - \left(-\frac{1}{2}\|\boldsymbol{x} - (\phi_k, \psi)\|_2^2\right)$$

$$= \frac{1}{2}\left((1 - c_1)\left\|\boldsymbol{x} + \frac{c_1(\bar{\phi}_k, \bar{\psi}) - (\phi_k, \psi)}{1 - c_1}\right\|_2^2 - \frac{c_1}{1 - c_1}\|(\bar{\phi}_k, \bar{\psi}) - (\phi_k, \psi)\|_2^2 + 2c_2\right).$$

Further taking $c_1 = 1 - \eta$ and $c_2 = d(1-\eta)\eta/2$, we have

$$(1 - c_1) \left\| \boldsymbol{x} + \frac{c_1 (\bar{\phi}_k, \bar{\psi}) - (\phi_k, \psi)}{1 - c_1} \right\|_2^2 - \frac{c_1}{1 - c_1} \left\| (\bar{\phi}_k, \bar{\psi}) - (\phi_k, \psi) \right\|_2^2 + 2c_2$$

$$= \eta \left\| \boldsymbol{x} + \frac{c_1 (\bar{\phi}_k, \bar{\psi}) - (\phi_k, \psi)}{1 - c_1} \right\|_2^2 - \frac{1 - \eta}{\eta} \left\| (\bar{\phi}_k, \bar{\psi}) - (\phi_k, \psi) \right\|_2^2 + 2c_2 \qquad (c_1 = 1 - \eta)$$

$$\geq -\frac{1 - \eta}{\eta} \left\| (\bar{\phi}_k, \bar{\psi}) - (\phi_k, \psi) \right\|_2^2 + 2c_2 \qquad (\eta > 0)$$

$$\geq -\frac{1 - \eta}{\eta} d\eta^2 + 2c_2 \qquad (\| (\phi_k, \psi) - (\bar{\phi}_k, \bar{\psi}) \|_2^2 \leq d\eta^2)$$

$$= -d(1 - \eta)\eta + d(1 - \eta)\eta = 0.$$

Therefore, it holds that for all $y \in \mathcal{Y}$,

$$\forall \boldsymbol{x} \in \mathcal{X} : p'(\boldsymbol{x}, y) \geq p_{X|Y}^{\mathrm{multi}}(\boldsymbol{x}|y). \tag{14}$$

Moreover, given any $0 < \epsilon \leq 1$, we take $\eta = \frac{\epsilon}{1+d}$, and thus $c_1 = 1 - \frac{\epsilon}{1+d}$ and $c_2 = \frac{1}{2}(1 - \frac{\epsilon}{1+d})\frac{\epsilon}{\frac{1}{d}+1}$. Since $d \in \mathbb{N}$, we have $\eta \leq \frac{1}{2}$ and $c_2 \leq \frac{1}{2}$. Then, $\| p'(\cdot, y) - p_{X|Y}^{\mathrm{multi}}(\cdot|y) \|_{L^1(\mathcal{X})}$ can be bounded as

$$\| p'(\cdot, y) - p_{X|Y}^{\mathrm{multi}}(\cdot|y) \|_{L^1(\mathcal{X})} = \int_{\mathcal{X}} | p'(\boldsymbol{x}, y) - p_{X|Y}^{\mathrm{multi}}(\boldsymbol{x}|y) | d\boldsymbol{x}$$

$$= \int_{\mathcal{X}} p'(\boldsymbol{x}, y) d\boldsymbol{x} - \int_{\mathcal{X}} p_{X|Y}^{\mathrm{multi}}(\boldsymbol{x}|y) d\boldsymbol{x} = \frac{1}{\sqrt{c_1}} e^{c_2} - 1 \qquad (\int_{\mathcal{X}} e^{-\frac{1}{2}\|\boldsymbol{x}\|_2^2} d\boldsymbol{x} = (2\pi)^{\frac{d}{2}})$$

$$\leq \frac{1}{\sqrt{c_1}}(1 + 2c_2) - 1 \qquad (e^x \leq 1 + 2x \text{ for } x \in [0, \tfrac{1}{2}])$$

$$= \frac{1}{\sqrt{1 - \eta}}(1 + d(1 - \eta)\eta) - 1 \qquad (c_1 = 1 - \eta \text{ and } c_2 = d(1 - \eta)\eta/2)$$

$$\leq (1 + \eta)(1 + d(1 - \eta)\eta) - 1 \qquad (\tfrac{1}{\sqrt{1-x}} \leq 1 + x \text{ for } x \in [0, \tfrac{1}{2}])$$

$$= \eta \left( 1 + d(1 - \eta^2) \right) \leq \eta(1 + d) = \epsilon \tag{15}$$

Combining Equation (14) and Equation (15), we know that for any $p_{X|Y}^{\mathrm{multi}}(\boldsymbol{x}|y) \in \mathcal{P}_{X|Y}^{\mathrm{multi}}$ and $0 < \epsilon \leq 1$, there exists some $p'(\boldsymbol{x}, y) \in \mathcal{B}$ such that given any $y \in \mathcal{Y}$, it holds that $\forall \boldsymbol{x} \in \mathcal{X} : p'(\boldsymbol{x}, y) \geq p_{X|Y}(\boldsymbol{x}|y)$, and $\| p'(\cdot, y) - p_{X|Y}(\cdot|y) \|_{L^p(\mathcal{X})} \leq \epsilon$, where

$$\mathcal{B} := \left\{ p'(\boldsymbol{x}, y) = \prod_{k=1}^K \left( (2\pi)^{-\frac{d}{2}} e^{-\frac{c_1}{2}\|\boldsymbol{x} - (\bar{\phi}_k, \bar{\psi})\|_2^2 + c_2} \right)^{\mathbb{I}(y=k)} : (\bar{\phi}_k)_i, (\bar{\psi})_i \in [-B, B] \cap \eta\mathbb{Z} \right\}$$

Recalling the definition of the upper bracketing number in Definition 3.1, we know that $\mathcal{B}$ is an $\epsilon$-upper bracket of $\mathcal{P}_{X|Y}^{\mathrm{multi}}$ w.r.t. $L^1(\mathcal{X})$. Therefore,

$$\mathcal{N}_{[]} \left( \epsilon; \mathcal{P}_{X|Y}^{\mathrm{multi}}, L^1(\mathcal{X}) \right)$$

$$\leq |\mathcal{B}| = \left| \left\{ \{\bar{\phi}_k\}_{k=1}^K, \bar{\psi} : (\bar{\phi}_k)_i, (\bar{\psi})_i \in [-B, B] \cap \eta\mathbb{Z} \right\} \right|$$

$$\leq \left( \frac{2B}{\eta} + 1 \right)^{K d_1 + d - d_1}$$

$$= \left( \frac{2(1+d)B}{\epsilon} + 1 \right)^{(K-1)d_1 + d},$$

which completes the proof. $\qquad\qquad\square$

## B.2. Proof of Theorem 4.1

*Proof of Theorem 4.1.* As $\phi_k^* \in \Phi$, $\psi^* \in \Psi$, and $\hat{p}_{X|Y}^{\text{multi}}$ is the maximizer of likelihood $L_S(p_{X|Y})$ in $\mathcal{P}_{X|Y}^{\text{multi}}$, according to Theorem 3.2, we know that

$$\mathcal{R}_{\overline{\text{TV}}}(\hat{p}_{X|Y}^{\text{multi}}) \leq 3\sqrt{\frac{1}{n}\left(\log \mathcal{N}_{[]}\left(\frac{1}{n}; \mathcal{P}_{X|Y}^{\text{multi}}, L^1(\mathcal{X})\right) + \log\frac{1}{\delta}\right)}.$$

According to Theorem B.1, it holds that

$$\mathcal{N}_{[]}\left(\frac{1}{n}; \mathcal{P}_{X|Y}^{\text{multi}}, L^1(\mathcal{X})\right) \leq \left(2(1+d)Bn+1\right)^{(K-1)d_1+d}.$$

Therefore, we obtain the result that

$$\mathcal{R}_{\overline{\text{TV}}}(\hat{p}_{X|Y}^{\text{multi}}) \leq 3\sqrt{\frac{1}{n}\left(\left((K-1)d_1+d\right)\log\left(2(1+d)Bn+1\right) + \log\frac{1}{\delta}\right)}.$$

Omitting constants about $n, K, d_1, d, B$, and the logarithm term we have $\mathcal{R}_{\overline{\text{TV}}}(\hat{p}_{X|Y}^{\text{multi}}) = \tilde{\mathcal{O}}\left(\sqrt{\frac{(K-1)d_1+d}{n}}\right)$. $\qquad\square$

## B.3. Average TV error bound under single-source training

**Theorem B.2** (Average TV error bound for conditional Gaussian distribution space under single-source training). *Let $\hat{p}_{X|Y}^{\text{single}}$ be the likelihood maximizer defined in Equation* (3) *given $\mathcal{P}_{X|Y}^{\text{single}}$ with conditional distributions as in Equation* (5). *Suppose $\Phi = [-B, B]^{d_1}$, $\Psi = [-B, B]^{d-d_1}$ with constant $B > 0$, and $\phi_k^* \in \Phi$, $\psi^* \in \Psi$. Then, for any $0 < \delta \leq 1/2$, it holds with probability at least $1 - \delta$ that*

$$\mathcal{R}_{\overline{\text{TV}}}(\hat{p}_{X|Y}^{\text{single}}) = \tilde{\mathcal{O}}\left(\sqrt{\frac{Kd}{n}}\right).$$

*Proof.* The proof is very similar to that in the multi-source case. According to the assumptions, the conditional distribution space expressed by the parametric estimation model is

$$\mathcal{P}_{X|Y}^{\text{single}} := \left\{ p_{X|Y}^{\text{single}}(\boldsymbol{x}|y) = \prod_{k=1}^{K}\left(p_{\phi_k,\psi_k}(\boldsymbol{x}|k)\right)^{\mathbb{I}(y=k)} = \prod_{k=1}^{K}\left((2\pi)^{-\frac{d}{2}}e^{-\frac{1}{2}\|\boldsymbol{x}-(\phi_k,\psi_k)\|_2^2}\right)^{\mathbb{I}(y=k)} : \phi_k \in [-B,B]^{d_1}, \psi_k \in [-B,B]^{d-d_1} \right\}.$$

For any $p_{X|Y}^{\text{single}}(\boldsymbol{x}|y) = \prod_{k=1}^{K}\left((2\pi)^{-\frac{d}{2}}e^{-\frac{1}{2}\|\boldsymbol{x}-(\phi_k,\psi_k)\|_2^2}\right)^{\mathbb{I}(y=k)} \in \mathcal{P}_{X|Y}^{\text{single}}$, let's first divide the mean vector $(\phi_k, \psi_k)$ into $\eta$-width grids with a small constant $\eta > 0$ (the value of $\eta$ will be specified later): If $(\phi_k)_i \in [j\eta, (j+1)\eta)$ for some $j \in \mathbb{Z}$, let $(\bar{\phi}_k)_i = j\eta$ and $\bar{\phi}_k := \left((\bar{\phi}_k)_1, \ldots, (\bar{\phi}_k)_{d_1}\right)$. Similarly, if $(\psi_k)_i \in [j\eta, (j+1)\eta)$ for some $j \in \mathbb{Z}$, let $(\bar{\psi}_k)_i = j\eta$ and $\bar{\psi}_k := \left((\bar{\psi}_k)_1, \ldots, (\bar{\psi}_k)_{d-d_1}\right)$. In this case, we have $\|(\phi_k, \psi_k) - (\bar{\phi}_k, \bar{\psi}_k)\|_2^2 \leq d\eta^2$.

Let

$$p'(\boldsymbol{x}, y) = \prod_{k=1}^{K}\left((2\pi)^{-\frac{d}{2}}e^{-\frac{c_1}{2}\|\boldsymbol{x}-(\bar{\phi}_k,\bar{\psi}_k)\|_2^2+c_2}\right)^{\mathbb{I}(y=k)}.$$

We need $p'(\boldsymbol{x}, y) \geq p_{X|Y}^{\text{single}}(\boldsymbol{x}|y)$ by the definition of the bracketing. By completing the square w.r.t. $\boldsymbol{x}$, we have

$$-\frac{c_1}{2}\|\boldsymbol{x}-(\bar{\phi}_k,\bar{\psi}_k)\|_2^2 + c_2 - \left(-\frac{1}{2}\|\boldsymbol{x}-(\phi_k,\psi_k)\|_2^2\right)$$

$$=\frac{1}{2}\left((1-c_1)\left\|\boldsymbol{x}+\frac{c_1(\bar{\phi}_k,\bar{\psi}_k)-(\phi_k,\psi_k)}{1-c_1}\right\|_2^2 - \frac{c_1}{1-c_1}\left\|(\bar{\phi}_k,\bar{\psi}_k)-(\phi_k,\psi_k)\right\|_2^2 + 2c_2\right).$$

Further taking $c_1 = 1 - \eta$ and $c_2 = d(1 - \eta)\eta/2$, we have

$$(1 - c_1)\left\|\boldsymbol{x} + \frac{c_1(\bar{\phi}_k, \bar{\psi}_k) - (\phi_k, \psi_k)}{1 - c_1}\right\|_2^2 - \frac{c_1}{1 - c_1}\|(\bar{\phi}_k, \bar{\psi}_k) - (\phi_k, \psi_k)\|_2^2 + 2c_2$$

$$= \eta\left\|\boldsymbol{x} + \frac{c_1(\bar{\phi}_k, \bar{\psi}_k) - (\phi_k, \psi_k)}{1 - c_1}\right\|_2^2 - \frac{1 - \eta}{\eta}\|(\bar{\phi}_k, \bar{\psi}_k) - (\phi_k, \psi_k)\|_2^2 + 2c_2 \qquad (c_1 = 1 - \eta)$$

$$\geq -\frac{1 - \eta}{\eta}\|(\bar{\phi}_k, \bar{\psi}_k) - (\phi_k, \psi_k)\|_2^2 + 2c_2 \qquad (\eta > 0)$$

$$\geq -\frac{1 - \eta}{\eta}d\eta^2 + 2c_2 \qquad (\|(\phi_k, \psi) - (\bar{\phi}_k, \bar{\psi}_k)\|_2^2 \leq d\eta^2)$$

$$= -d(1 - \eta)\eta + d(1 - \eta)\eta = 0.$$

Therefore, it holds that for all $y \in \mathcal{Y}$,

$$\forall \boldsymbol{x} \in \mathcal{X} : p'(\boldsymbol{x}, y) \geq p_{X|Y}^{\text{single}}(\boldsymbol{x}|y). \qquad (16)$$

Moreover, given any $0 < \epsilon \leq 1$, we take $\eta = \frac{\epsilon}{1+d}$, and thus $c_1 = 1 - \frac{\epsilon}{1+d}$ and $c_2 = \frac{1}{2}(1 - \frac{\epsilon}{1+d})\frac{\epsilon}{\frac{1}{d}+1}$. Since $d \in \mathbb{N}$, we have $\eta \leq \frac{1}{2}$ and $c_2 \leq \frac{1}{2}$. Then, $\|p'(\cdot, y) - p_{X|Y}^{\text{single}}(\cdot|y)\|_{L^1(\mathcal{X})}$ can be bounded as

$$\|p'(\cdot, y) - p_{X|Y}^{\text{single}}(\cdot|y)\|_{L^1(\mathcal{X})} = \int_{\mathcal{X}} |p'(\boldsymbol{x}, y) - p_{X|Y}^{\text{single}}(\boldsymbol{x}|y)| d\boldsymbol{x}$$

$$= \int_{\mathcal{X}} p'(\boldsymbol{x}, y) d\boldsymbol{x} - \int_{\mathcal{X}} p_{X|Y}^{\text{single}}(\boldsymbol{x}|y) d\boldsymbol{x} = \frac{1}{\sqrt{c_1}} e^{c_2} - 1 \qquad (\int_{\mathcal{X}} e^{-\frac{1}{2}\|\boldsymbol{x}\|_2^2} d\boldsymbol{x} = (2\pi)^{\frac{d}{2}})$$

$$\leq \frac{1}{\sqrt{c_1}}(1 + 2c_2) - 1 \qquad (e^x \leq 1 + 2x \text{ for } x \in [0, \frac{1}{2}])$$

$$= \frac{1}{\sqrt{1 - \eta}}(1 + d(1 - \eta)\eta) - 1 \qquad (c_1 = 1 - \eta \text{ and } c_2 = d(1 - \eta)\eta/2)$$

$$\leq (1 + \eta)(1 + d(1 - \eta)\eta) - 1 \qquad (\frac{1}{\sqrt{1-x}} \leq 1 + x \text{ for } x \in [0, \frac{1}{2}])$$

$$= \eta\Big(1 + d(1 - \eta^2)\Big) \leq \eta(1 + d) = \epsilon \qquad (17)$$

Combining Equation (16) and Equation (17), we know that for any $p_{X|Y}^{\text{single}}(\boldsymbol{x}|y) \in \mathcal{P}_{X|Y}^{\text{single}}$ and $0 < \epsilon \leq 1$, there exists some $p'(\boldsymbol{x}, y) \in \mathcal{B}$ such that given any $y \in \mathcal{Y}$, it holds that $\forall \boldsymbol{x} \in \mathcal{X} : p'(\boldsymbol{x}, y) \geq p_{X|Y}(\boldsymbol{x}|y)$, and $\|p'(\cdot, y) - p_{X|Y}(\cdot|y)\|_{L^p(\mathcal{X})} \leq \epsilon$, where

$$\mathcal{B} := \left\{ p'(\boldsymbol{x}, y) = \prod_{k=1}^{K} \left( (2\pi)^{-\frac{d}{2}} e^{-\frac{c_1}{2}\|\boldsymbol{x} - (\bar{\phi}_k, \bar{\psi}_k)\|_2^2 + c_2} \right)^{\mathbb{I}(y=k)} : (\bar{\phi}_k)_i, (\bar{\psi}_k)_i \in [-B, B] \cap \eta\mathbb{Z} \right\}$$

Recalling the definition of the upper bracketing number in Definition 3.1, we know that $\mathcal{B}$ is an $\epsilon$-upper bracket of $\mathcal{P}_{X|Y}^{\text{single}}$ w.r.t. $L^1(\mathcal{X})$. Therefore,

$$\mathcal{N}_{[]}\Big(\epsilon; \mathcal{P}_{X|Y}^{\text{single}}, L^1(\mathcal{X})\Big)$$

$$\leq |\mathcal{B}| = \left| \left\{ \{\bar{\phi}_k\}_{k=1}^{K}, \{\bar{\psi}_k\}_{k=1}^{K} : (\bar{\phi}_k)_i, (\bar{\psi}_k)_i \in [-B, B] \cap \eta\mathbb{Z} \right\} \right|$$

$$\leq \left( \frac{2B}{\eta} + 1 \right)^{Kd_1 + K(d - d_1)}$$

$$= \left( \frac{2(1 + d)B}{\epsilon} + 1 \right)^{Kd}.$$

Besides, according to Theorem 3.2, we know that

$$\mathcal{R}_{\overline{\mathrm{TV}}}(\hat{p}_{X|Y}^{\mathrm{single}}) \leq 3\sqrt{\frac{1}{n}\left(\log\mathcal{N}_{[]}\left(\frac{1}{n};\mathcal{P}_{X|Y}^{\mathrm{single}},L^1(\mathcal{X})\right)+\log\frac{1}{\delta}\right)}$$

$$\leq 3\sqrt{\frac{1}{n}\left(Kd\log\big(2(1+d)Bn+1\big)+\log\frac{1}{\delta}\right)}.$$

Omitting constants about $n, K, d_1, d, B$, and the logarithm term we have $\mathcal{R}_{\overline{\mathrm{TV}}}(\hat{p}_{X|Y}^{\mathrm{multi}}) = \tilde{\mathcal{O}}\left(\sqrt{\frac{Kd}{n}}\right)$. $\qquad\square$

## C. Proofs for Section 4.2

### C.1. Preliminaries for evaluating the bracketing number of neural networks

Our results build on the intrinsic connection between the bracketing number of conditional distribution spaces and the covering number of neural network models. There is a rich body of work on the complexity of ReLU fully connected neural networks, also known as multilayer perceptrons (MLPs) as defined in Definition 4.2 from various perspectives, including Rademacher complexity (Bartlett et al., 2017), VC-dimension (Bartlett et al., 2019), and covering numbers (Suzuki, 2019; Shen, 2024; Ou & Bölcskei, 2024). Specifically, prior results indicate that the logarithm of the covering number of an MLP scales as $\tilde{\mathcal{O}}(LS)$, where $L$ is the depth and $S$ is the sparsity constraint. Furthermore, Ou & Bölcskei (2024) establish a lower bound, showing that for $B \geq 1$ and $W, L \geq 60$, the covering number scales as $\tilde{\Theta}(LS)$. Their proofs share a common idea. To enhance clarity, we include a detailed derivation below following these prior works.

**Definition C.1** ($\epsilon$-covering number). Let $\epsilon$ be a real number that $\epsilon > 0$ and $\mathsf{p}, \mathsf{q}$ be integers that $1 \leq \mathsf{p}, \mathsf{q} \leq \infty$. An $\epsilon$-cover of a function space $\mathcal{F}$ with respect to $\|\cdot\|$ is a finite function set $\mathcal{C} \subset \mathcal{F}$ such that for any $\boldsymbol{f} \in \mathcal{F}$, there exists some $\boldsymbol{f}' \in \mathcal{C}$ such that $\big\|\|\boldsymbol{f}(\boldsymbol{x})-\boldsymbol{f}'(\boldsymbol{x})\|_{\mathsf{q}}\big\|_{L^{\mathsf{p}}(\mathcal{X})} \leq \epsilon$. In particular, when $\mathsf{p} = \mathsf{q} = \infty$, it requires $\big\|\|\boldsymbol{f}(\boldsymbol{x})-\boldsymbol{f}'(\boldsymbol{x})\|_{\infty}\big\|_{L^{\infty}(\mathcal{X})} = \sup_{\boldsymbol{x}\in\mathcal{X}}\|\boldsymbol{f}(\boldsymbol{x})-\boldsymbol{f}'(\boldsymbol{x})\|_{\infty} \leq \epsilon$. The $\epsilon$-covering number $\mathcal{N}\left(\epsilon;\mathcal{F},\|\cdot\|_{\mathsf{q},L^{\mathsf{p}}(\mathcal{X})}\right)$ is the cardinality of the smallest $\epsilon$-cover with respect to $\|\cdot\|_{\mathsf{q},L^{\mathsf{p}}(\mathcal{X})}$.

**Lemma C.2** (Lipschitz property of ReLU and sigmoid, Lemma A.1 in (Bartlett et al., 2017)). *Element-wise ReLU* $\mathrm{ReLU}(\cdot): \mathbb{R}^d \to \mathbb{R}^d$ *and sigmoid* $\sigma(x) = \frac{1}{1+e^{-x}}$ *are 1-Lipschitz according to* $\|\cdot\|_{\mathsf{p}}$ *for any* $\mathsf{p} \geq 1$.

**Lemma C.3** (Covering number of a composite function class). *Suppose we have two classes of functions, $\mathcal{G}$ consisting of functions mapping from $\mathcal{X}_1$ to $\mathcal{X}_2$ and $\mathcal{F}$ consisting of functions mapping from $\mathcal{X}_2$ to $\mathcal{X}_3$. We denote by $\mathcal{F} \circ \mathcal{G}$ all possible composition of functions in $\mathcal{F}$ and $\mathcal{G}$ that $\mathcal{F} \circ \mathcal{G} = \{\boldsymbol{f} \circ \boldsymbol{g}: \boldsymbol{f} \in \mathcal{F}, \boldsymbol{g} \in \mathcal{G}\}$. Assume that any $\boldsymbol{f} \in \mathcal{F}$ is $\kappa_{\mathcal{F}}$-Lipschitz w.r.t. $\|\cdot\|_{\infty}$, i.e., for all $\boldsymbol{x}_2, \boldsymbol{x}_2' \in \mathcal{X}_2$, $\|\boldsymbol{f}(\boldsymbol{x}_2)-\boldsymbol{f}(\boldsymbol{x}_2')\|_{\infty} \leq \kappa_{\mathcal{F}}\|\boldsymbol{x}_2-\boldsymbol{x}_2'\|_{\infty}$. Then, given constants $\epsilon_{\mathcal{F}}, \epsilon_{\mathcal{G}} > 0$ we have*

$$\mathcal{N}\left(\epsilon_{\mathcal{F}}+\kappa_{\mathcal{F}}\epsilon_{\mathcal{G}};\mathcal{F}\circ\mathcal{G},\|\cdot\|_{\infty,L^{\infty}(\mathcal{X}_1)}\right) \leq \mathcal{N}\left(\epsilon_{\mathcal{F}};\mathcal{F},\|\cdot\|_{\infty,L^{\infty}(\mathcal{X}_2)}\right)\mathcal{N}\left(\epsilon_{\mathcal{G}};\mathcal{G},\|\cdot\|_{\infty,L^{\infty}(\mathcal{X}_1)}\right)$$

*Proof.* Let $\mathcal{C}_{\mathcal{F}}$ be an $\epsilon_{\mathcal{F}}$-cover of $\mathcal{F}$ w.r.t. $\|\cdot\|_{\infty,L^{\infty}(\mathcal{X}_2)}$ such that $|\mathcal{C}_{\mathcal{F}}| = \mathcal{N}\left(\epsilon_{\mathcal{F}};\mathcal{F},\|\cdot\|_{\infty,L^{\infty}(\mathcal{X}_2)}\right)$, and $\mathcal{C}_{\mathcal{G}}$ be an $\epsilon_{\mathcal{G}}$-cover of $\mathcal{G}$ w.r.t. $\|\cdot\|_{\infty,L^{\infty}(\mathcal{X}_1)}$ such that $|\mathcal{C}_{\mathcal{G}}| = \mathcal{N}\left(\epsilon_{\mathcal{G}};\mathcal{G},\|\cdot\|_{\infty,L^{\infty}(\mathcal{X}_1)}\right)$. For any $\boldsymbol{f} \circ \boldsymbol{g} \in \mathcal{F} \circ \mathcal{G}$, there exists $\boldsymbol{f}' \in \mathcal{C}_{\mathcal{F}}$ and $\boldsymbol{g}' \in \mathcal{C}_{\mathcal{G}}$ such that

$$\forall \boldsymbol{x}_2 \in \mathcal{X}_2, \|\boldsymbol{f}(\boldsymbol{x}_2)-\boldsymbol{f}'(\boldsymbol{x}_2)\|_{\infty} \leq \epsilon_{\mathcal{F}}, \quad\text{and}\quad \forall \boldsymbol{x}_1 \in \mathcal{X}_1, \|\boldsymbol{g}(\boldsymbol{x}_1)-\boldsymbol{g}'(\boldsymbol{x}_1)\|_{\infty} \leq \epsilon_{\mathcal{G}}.$$

Then, for any $\boldsymbol{x}_1 \in \mathcal{X}_1$, we have

$$\|\boldsymbol{f}\circ\boldsymbol{g}(\boldsymbol{x}_1)-\boldsymbol{f}'\circ\boldsymbol{g}'(\boldsymbol{x}_1)\|_{\infty} \leq \|\boldsymbol{f}\circ\boldsymbol{g}(\boldsymbol{x}_1)-\boldsymbol{f}'\circ\boldsymbol{g}(\boldsymbol{x}_1)\|_{\infty} + \|\boldsymbol{f}'\circ\boldsymbol{g}(\boldsymbol{x}_1)-\boldsymbol{f}'\circ\boldsymbol{g}'(\boldsymbol{x}_1)\|_{\infty}$$
$$\leq \epsilon_{\mathcal{F}} + \kappa_{\mathcal{F}}\|\boldsymbol{g}(\boldsymbol{x}_1)-\boldsymbol{g}'(\boldsymbol{x}_1)\|_{\infty}$$
$$\leq \epsilon_{\mathcal{F}} + \kappa_{\mathcal{F}}\epsilon_{\mathcal{G}}.$$

Therefore, we have $\mathcal{C}_{\mathcal{F}} \circ \mathcal{C}_{\mathcal{G}}$ is an $\epsilon_{\mathcal{F}} + \kappa_{\mathcal{F}}\epsilon_{\mathcal{G}}$-cover of $\mathcal{F} \circ \mathcal{G}$, and thus

$$\mathcal{N}\left(\epsilon_{\mathcal{F}}+\kappa_{\mathcal{F}}\epsilon_{\mathcal{G}};\mathcal{F}\circ\mathcal{G},\|\cdot\|_{\infty,L^{\infty}(\mathcal{X}_1)}\right) \leq |\mathcal{C}_{\mathcal{F}}\circ\mathcal{C}_{\mathcal{G}}| \leq |\mathcal{C}_{\mathcal{F}}||\mathcal{C}_{\mathcal{G}}| = \mathcal{N}\left(\epsilon_{\mathcal{F}};\mathcal{F},\|\cdot\|_{\infty,L^{\infty}(\mathcal{X}_2)}\right)\mathcal{N}\left(\epsilon_{\mathcal{G}};\mathcal{G},\|\cdot\|_{\infty,L^{\infty}(\mathcal{X}_1)}\right).$$

$\square$

**Lemma C.4** (Covering number of an MLP class). *Given any constant $\delta > 0$, the covering number with respect to* $\|\cdot\|_{\infty, L^\infty(\mathcal{X})}$ *with* $\mathcal{X} \subset [0,1]^{W_0}$ *of an MLP class* $\mathcal{F}(L, W, S, B)$ *defined in Definition 4.2 can be bounded by*

$$\mathcal{N}\left(L(B \vee 1)^{L-1}(W+1)^L \delta; \mathcal{F}(L, W, S, B), \|\cdot\|_{\infty, L^\infty(\mathcal{X})}\right) \leq \left(\frac{2B}{\delta} + 1\right)^S.$$

*Proof.* Fix any $\boldsymbol{x} \in [0,1]^{W_0}$. Given any network $\boldsymbol{f} \in \mathcal{F}(L, W, S, B)$ expressed as

$$\boldsymbol{f}(\boldsymbol{x}) = (\boldsymbol{A}^{(L)}\text{ReLU}(\cdot) + \boldsymbol{b}^{(L)}) \circ \cdots \circ (\boldsymbol{A}^{(1)}\boldsymbol{x} + \boldsymbol{b}^{(1)}),$$

let $\boldsymbol{f}_l(\boldsymbol{x}) := (\boldsymbol{A}^{(l)}\text{ReLU}(\cdot) + \boldsymbol{b}^{(l)}) \circ \cdots \circ (\boldsymbol{A}^{(1)}\boldsymbol{x} + \boldsymbol{b}^{(1)})$ for $l = 2, \ldots, L$ and $\boldsymbol{f}_1(\boldsymbol{x}) = \boldsymbol{A}^{(1)}\boldsymbol{x} + \boldsymbol{b}^{(1)}$.

**Sup-norm of the output at each layer.** We first prove the statement that $\|\boldsymbol{f}_l(\boldsymbol{x})\|_\infty \leq (B \vee 1)^l(W+1)^l$ for all $l \in [L]$ by induction. When $l = 1$,

$$\begin{aligned}
\|\boldsymbol{f}_1(\boldsymbol{x})\|_\infty = \|\boldsymbol{A}^{(1)}\boldsymbol{x} + \boldsymbol{b}^{(1)}\|_\infty &\leq \max_i \|\boldsymbol{A}^{(1)}[i,:]\|_1 \|\boldsymbol{x}\|_\infty + \|\boldsymbol{b}^{(1)}\|_\infty \\
&\leq WB + B \qquad\qquad (W_0 \leq W, \|\boldsymbol{A}^{(1)}\|_\infty \leq B, \|\boldsymbol{b}^{(1)}\|_\infty \leq B, \boldsymbol{x} \in \mathcal{X} \subset [0,1]^{W_0}) \\
&= B(W+1) \leq (B \vee 1)^1(W+1)^1,
\end{aligned}$$

which implies the statement is true for $l = 1$. Assume that for some $l = i \geq 1$, $\|\boldsymbol{f}_i(\boldsymbol{x})\|_\infty \leq (B \vee 1)^i(W+1)^i$, then we have

$$\begin{aligned}
\|\boldsymbol{f}_{i+1}(\boldsymbol{x})\|_\infty = \|\boldsymbol{A}^{(i+1)}\text{ReLU}(\boldsymbol{f}_i(\boldsymbol{x})) + \boldsymbol{b}^{(i+1)}\|_\infty &\leq \max_i \|\boldsymbol{A}^{(i+1)}[i,:]\|_1 \|\text{ReLU}(\boldsymbol{f}_i(\boldsymbol{x}))\|_\infty + \|\boldsymbol{b}^{(i+1)}\|_\infty \\
&\leq WB\|\text{ReLU}(\boldsymbol{f}_i(\boldsymbol{x}))\|_\infty + B \qquad\qquad (W_i \leq W, \|\boldsymbol{A}^{(i+1)}\|_\infty \leq B, \|\boldsymbol{b}^{(i+1)}\|_\infty \leq B) \\
&\leq WB\|\boldsymbol{f}_i(\boldsymbol{x})\|_\infty + B \qquad (\text{ReLU}(\cdot) \text{ is 1-Lipschitz continuous for Lemma C.2 and } \text{ReLU}(\boldsymbol{0}) = \boldsymbol{0}) \\
&\leq WB(B \vee 1)^i(W+1)^i + B \\
&\leq \begin{cases} WB^{i+1}(W+1)^i + B^{i+1}(W+1)^i, & B \geq 1, \\ W(W+1)^i + (W+1)^i, & B < 1, \end{cases} \\
&= (B \vee 1)^{i+1}(W+1)^{i+1},
\end{aligned}$$

which implies the statement is true for $l = i + 1$, completing the induction steps. Therefore, it holds that for all $l \in [L]$,

$$\|\boldsymbol{f}_l(\boldsymbol{x})\|_\infty \leq (B \vee 1)^l(W+1)^l. \tag{18}$$

**Parameter-Lipschitzness at each layer.** For any two different neural networks $\boldsymbol{f}, \boldsymbol{f}' \in \mathcal{F}(L, W, S, B)$ expressed by

$$\boldsymbol{f}(x) = (\boldsymbol{A}^{(L)}\text{ReLU}(\cdot) + \boldsymbol{b}^{(L)}) \circ \cdots \circ (\boldsymbol{A}^{(1)}\boldsymbol{x} + \boldsymbol{b}^{(1)}), \boldsymbol{f}'(x) = (\boldsymbol{A}^{(L)'}\text{ReLU}(\cdot) + \boldsymbol{b}^{(L)'}) \circ \cdots \circ (\boldsymbol{A}^{(1)'}\boldsymbol{x} + \boldsymbol{b}^{(1)'}),$$

with parameter distance that $\max_l \|\boldsymbol{A}^{(l)} - \boldsymbol{A}^{(l)'}\|_\infty \vee \|\boldsymbol{b}^{(l)} - \boldsymbol{b}^{(l)'}\|_\infty \leq \delta$, we prove the statement that $\|\boldsymbol{f}_l(\boldsymbol{x}) - \boldsymbol{f}'_l(\boldsymbol{x})\|_\infty \leq l(B \vee 1)^{l-1}(W+1)^l \delta$ for all $l \in [L]$ by induction. When $l = 1$,

$$\begin{aligned}
\|\boldsymbol{f}_1(\boldsymbol{x}) - \boldsymbol{f}'_1(\boldsymbol{x})\|_\infty = \|\boldsymbol{A}^{(1)}\boldsymbol{x} + \boldsymbol{b}^{(1)} - \boldsymbol{A}^{(1)'}\boldsymbol{x} - \boldsymbol{b}^{(1)'}\|_\infty &\\
&\leq \|(\boldsymbol{A}^{(1)} - \boldsymbol{A}^{(1)'})\boldsymbol{x}\|_\infty + \|\boldsymbol{b}^{(1)} - \boldsymbol{b}^{(1)'}\|_\infty \\
&\leq W\delta + \delta \qquad (W_0 \leq W, \|\boldsymbol{A}^{(1)} - \boldsymbol{A}^{(1)'}\|_\infty \leq \delta, \|\boldsymbol{b}^{(1)} - \boldsymbol{b}^{(1)'}\|_\infty \leq \delta, \boldsymbol{x} \in [0,1]^{W_0}) \\
&= (W+1)\delta \\
&\leq (B \vee 1)^0(W+1)^1\delta,
\end{aligned}$$

which implies the statement is true for $l = 1$. Assume that for some $l = i \geq 1$, $\|\boldsymbol{f}_i(\boldsymbol{x}) - \boldsymbol{f}'_i(\boldsymbol{x})\|_\infty \leq i(B \vee 1)^{i-1}(W+1)^i \delta$, then we have

$$
\begin{aligned}
\|\boldsymbol{f}_{i+1}(\boldsymbol{x}) - \boldsymbol{f}'_{i+1}(\boldsymbol{x})\|_\infty &= \|\boldsymbol{A}^{(i+1)}\mathrm{ReLU}(\boldsymbol{f}_i(\boldsymbol{x})) + \boldsymbol{b}^{(i+1)} - \boldsymbol{A}^{(i+1)'}\mathrm{ReLU}(\boldsymbol{f}'_i(x)) - \boldsymbol{b}^{(i+1)'}\|_\infty \\
&\leq \|\big(\boldsymbol{A}^{(i+1)} - \boldsymbol{A}^{(i+1)'}\big)\mathrm{ReLU}(\boldsymbol{f}_i(\boldsymbol{x}))\|_\infty + \|\boldsymbol{A}^{(i+1)'}\big(\mathrm{ReLU}(\boldsymbol{f}_i(\boldsymbol{x})) - \mathrm{ReLU}(\boldsymbol{f}'_i(\boldsymbol{x}))\big)\|_\infty \\
&\quad + \|\boldsymbol{b}^{(i+1)} - \boldsymbol{b}^{(i+1)'}\|_\infty \\
&\leq W\delta\|\mathrm{ReLU}(\boldsymbol{f}_i(\boldsymbol{x}))\|_\infty + WB\big\|\mathrm{ReLU}(\boldsymbol{f}_i(\boldsymbol{x})) - \mathrm{ReLU}(\boldsymbol{f}'_i(\boldsymbol{x}))\big\|_\infty + \delta \\
&\qquad (W_i \leq W, \|\boldsymbol{A}^{(i+1)} - \boldsymbol{A}^{(i+1)'}\|_\infty \leq \delta, \|\boldsymbol{A}^{(i+1)'}\|_\infty \leq B, \|\boldsymbol{b}^{(i+1)} - \boldsymbol{b}^{(i+1)'}\|_\infty \leq \delta) \\
&\leq W\delta\|\boldsymbol{f}_i(\boldsymbol{x})\|_\infty + WB\big\|\boldsymbol{f}_i(\boldsymbol{x}) - \boldsymbol{f}'_i(\boldsymbol{x})\big\|_\infty + \delta \\
&\qquad (\mathrm{ReLU}(\cdot) \text{ is 1-Lipschitz continuous for Lemma C.2 and } \mathrm{ReLU}(\boldsymbol{0}) = \boldsymbol{0}) \\
&\leq W\delta(B \vee 1)^i(W+1)^i + WB\big\|\boldsymbol{f}_i(\boldsymbol{x}) - \boldsymbol{f}'_i(\boldsymbol{x})\big\|_\infty + \delta \qquad\qquad\quad \text{(Equation (18))} \\
&\leq W\delta(B \vee 1)^i(W+1)^i + WBi(B \vee 1)^{i-1}(W+1)^i\delta + \delta \\
&\leq \Big(W(B \vee 1)^i(W+1)^i + iW(B \vee 1)^i(W+1)^i + 1\Big)\delta \\
&= \Big((i+1)W(B \vee 1)^i(W+1)^i + 1\Big)\delta \\
&\leq \begin{cases} \big(W(i+1)B^i(W+1)^i + (i+1)B^i(W+1)^i\big)\delta, & B \geq 1, \\ \big(W(i+1)(W+1)^i + (i+1)(W+1)^i\big)\delta, & B < 1, \end{cases} \\
&= (i+1)(B \vee 1)^i(W+1)^{i+1}\delta
\end{aligned}
$$

which implies the statement is true for $l = i + 1$, completing the induction steps. Therefore, it holds that for all $l \in [L]$,

$$
\|\boldsymbol{f}_l(\boldsymbol{x}) - \boldsymbol{f}'_l(\boldsymbol{x})\|_\infty \leq l(B \vee 1)^{l-1}(W+1)^l \delta. \tag{19}
$$

**Discretization of entry space.** Let $\mathcal{S}_{\mathrm{entry}}(\{(\boldsymbol{A}^{(l)}, \boldsymbol{b}^{(l)})\}_{l=1}^L) \coloneqq \bigcup_{l=1}^L \Big(\mathcal{S}_{\mathrm{entry}}(\boldsymbol{A}^{(l)}) \bigcup \mathcal{S}_{\mathrm{entry}}(\boldsymbol{b}^{(l)})\Big)$ with $\mathcal{S}_{\mathrm{entry}}(\boldsymbol{A})$ denotes the value space of all entries in $\boldsymbol{A}$ and $\mathcal{S}_{\mathrm{entry}}(\boldsymbol{b})$ denotes the value space of all entries in $b$. Now we discretize the value spaces of $\mathcal{F}(L, W, S, B)$ into $\delta$-width grids and get a finite class of neural network as $\mathcal{F}_{\delta\mathbb{Z}}(L, W, S, B) \coloneqq \{\boldsymbol{f} \in \mathcal{F}(L, W, S, B) : \mathcal{S}_{\mathrm{entry}}(\{(\boldsymbol{A}^{(l)}, \boldsymbol{b}^{(l)})\}_{l=1}^L)) = [-B, B] \cap \delta\mathbb{Z}\}$, where $\delta\mathbb{Z} = \{k\delta | k \in \mathbb{Z}\}$ Then, for any $\boldsymbol{f} \in \mathcal{F}(L, W, S, B)$ expressed as $\boldsymbol{f}(x) = (\boldsymbol{A}^{(L)}\mathrm{ReLU}(\cdot) + \boldsymbol{b}^{(L)}) \circ \cdots \circ (\boldsymbol{A}^{(1)}\boldsymbol{x} + \boldsymbol{b}^{(1)})$, there exists $\boldsymbol{f}' \in \mathcal{F}_{\delta\mathbb{Z}}(L, W, S, B)$ expressed as $\boldsymbol{f}'(\boldsymbol{x}) = (\boldsymbol{A}^{(L)'}\mathrm{ReLU}(\cdot) + \boldsymbol{b}^{(L)'}) \circ \cdots \circ (\boldsymbol{A}^{(1)'}\boldsymbol{x} + \boldsymbol{b}^{(1)'})$ such that $\max_l \|\boldsymbol{A}^{(l)} - \boldsymbol{A}^{(l)'}\|_\infty \vee \|\boldsymbol{b}^{(l)} - \boldsymbol{b}^{(l)'}\|_\infty \leq \delta$. According to Equation (19), we have for any $\boldsymbol{x} \in [0, 1]^{W_0}$, $\|\boldsymbol{f}(\boldsymbol{x}) - \boldsymbol{f}'(\boldsymbol{x})\|_\infty \leq L(B \vee 1)^{L-1}(W+1)^L\delta$. Therefore, $\mathcal{F}_{\delta\mathbb{Z}}(L, W, S, B)$ is an $L(B \vee 1)^{L-1}(W+1)^L\delta$-cover of $\mathcal{F}(L, W, S, B)$ with respect to $\|\cdot\|_{\infty, L^\infty(\mathcal{X})}$ and thus we have

$$
\begin{aligned}
&\mathcal{N}\Big(L(B \vee 1)^{L-1}(W+1)^L\delta; \mathcal{F}(L, W, S, B), \|\cdot\|_{\infty, L^\infty(\mathcal{X})}\Big) \\
&\leq |\mathcal{F}_{\delta\mathbb{Z}}(L, W, S, B)| = \left|\left\{(\boldsymbol{A}^{(l)}, \boldsymbol{b}^{(l)})\}_{l=1}^L : \mathcal{S}_{\mathrm{entry}}(\boldsymbol{A}^{(l)}) = \mathcal{S}_{\mathrm{entry}}(\boldsymbol{b}^{(l)}) = [-B, B] \cap \delta\mathbb{Z}\right\}\right| \leq \left(\frac{2B}{\delta} + 1\right)^S,
\end{aligned}
$$

which completes the proof. $\qquad\square$

Here, we further establish the Lipschitz property of MLPs, which is useful in the following proofs for deriving the covering number for MLPs with an embedding layer applied to input data.

**Lemma C.5** (Lipschitz property of MLPs about the input). *For any $\boldsymbol{f} \in \mathcal{F}(L, W, S, B)$ defined in Definition 4.2, $\boldsymbol{f}$ is $B^L W^L$-Lipschitz continuous w.r.t. $\|\cdot\|_\infty$ about $\boldsymbol{x}$ on $\mathcal{X}$, i.e., for any $\boldsymbol{x}, \boldsymbol{x}' \in \mathcal{X} \subset \mathbb{R}^{W_0}$, it holds that*

$$
\|\boldsymbol{f}(\boldsymbol{x}) - \boldsymbol{f}(\boldsymbol{x}')\|_\infty \leq B^L W^L \|\boldsymbol{x} - \boldsymbol{x}'\|_\infty.
$$

*Proof.* Fix any $\boldsymbol{x}, \boldsymbol{x}' \in \mathcal{X}$. Given any $\boldsymbol{f} \in \mathcal{F}(L, W, S, B)$ expressed as $\boldsymbol{f}(x) = (\boldsymbol{A}^{(L)}\mathrm{ReLU}(\cdot) + \boldsymbol{b}^{(L)}) \circ \cdots \circ (\boldsymbol{A}^{(1)}\boldsymbol{x} + \boldsymbol{b}^{(1)})$, let $\boldsymbol{f}_l(x) \coloneqq (\boldsymbol{A}^{(l)}\mathrm{ReLU}(\cdot) + \boldsymbol{b}^{(l)}) \circ \cdots \circ (\boldsymbol{A}^{(1)}\boldsymbol{x} + \boldsymbol{b}^{(1)})$ for $l = 2, \ldots, L$ and $\boldsymbol{f}_1(x) = \boldsymbol{A}^{(1)}\boldsymbol{x} + \boldsymbol{b}^{(1)}$. We prove the

statement that $\|\boldsymbol{f}_l(\boldsymbol{x}) - \boldsymbol{f}_l(\boldsymbol{x}')\|_\infty \leq B^l W^l \|\boldsymbol{x} - \boldsymbol{x}'\|_\infty$ for all $l \in [L]$ by induction. When $l = 1$,

$$
\begin{aligned}
\|\boldsymbol{f}_1(\boldsymbol{x}) - \boldsymbol{f}_1(\boldsymbol{x}')\|_\infty &= \|\boldsymbol{A}^{(1)}\boldsymbol{x} + \boldsymbol{b}^{(1)} - \boldsymbol{A}^{(1)}\boldsymbol{x}' - \boldsymbol{b}^{(1)}\|_\infty = \|\boldsymbol{A}^{(1)}(\boldsymbol{x} - \boldsymbol{x}')\|_\infty \\
&\leq BW\|\boldsymbol{x} - \boldsymbol{x}'\|_\infty \qquad\qquad\qquad (W_0 \leq W, \|\boldsymbol{A}^{(1)}\|_\infty \leq B) \\
&= B^1 W^1 \|\boldsymbol{x} - \boldsymbol{x}'\|_\infty,
\end{aligned}
$$

which implies the statement is true for $l = 1$. Assume that for some $l = i \geq 1$, $\|\boldsymbol{f}_i(\boldsymbol{x}) - \boldsymbol{f}_i(\boldsymbol{x}')\|_\infty \leq B^i W^i \|\boldsymbol{x} - \boldsymbol{x}'\|_\infty$, then we have

$$
\begin{aligned}
\|\boldsymbol{f}_{i+1}(\boldsymbol{x}) - \boldsymbol{f}_{i+1}(\boldsymbol{x}')\|_\infty &= \|\boldsymbol{A}^{(i+1)}\mathrm{ReLU}\big(\boldsymbol{f}_i(\boldsymbol{x})\big) + \boldsymbol{b}^{(i+1)} - \boldsymbol{A}^{(i+1)}\mathrm{ReLU}\big(\boldsymbol{f}_i(\boldsymbol{x}')\big) - \boldsymbol{b}^{(i+1)}\|_\infty \\
&= \|\boldsymbol{A}^{(i+1)}\big(\mathrm{ReLU}\big(\boldsymbol{f}_i(\boldsymbol{x})\big) - \mathrm{ReLU}\big(\boldsymbol{f}_i(\boldsymbol{x}')\big)\big)\|_\infty \\
&\leq WB\|\mathrm{ReLU}\big(\boldsymbol{f}_i(\boldsymbol{x})\big) - \mathrm{ReLU}\big(\boldsymbol{f}_i(\boldsymbol{x}')\big)\|_\infty \qquad (W_i \leq W, \|\boldsymbol{A}^{(i+1)}\|_\infty \leq B) \\
&\leq WB\|\boldsymbol{f}_i(\boldsymbol{x}) - \boldsymbol{f}_i(\boldsymbol{x}')\|_\infty \qquad (\mathrm{ReLU}(\cdot) \text{ is 1-Lipschitz continuous for Lemma C.2}) \\
&\leq WBB^i W^i \|\boldsymbol{x} - \boldsymbol{x}'\|_\infty \\
&= B^{i+1} W^{i+1} \|\boldsymbol{x} - \boldsymbol{x}'\|_\infty,
\end{aligned}
$$

which implies the statement is true for $l = i + 1$, completing the induction steps. Therefore, it holds that for all $l \in [L]$,

$$
\|\boldsymbol{f}_l(\boldsymbol{x}) - \boldsymbol{f}_l(\boldsymbol{x}')\|_\infty \leq B^l W^l \|\boldsymbol{x} - \boldsymbol{x}'\|_\infty.
$$

Thus, when $l = L$, we have $\|\boldsymbol{f}(\boldsymbol{x}) - \boldsymbol{f}(\boldsymbol{x}')\|_\infty = \|\boldsymbol{f}_L(\boldsymbol{x}) - \boldsymbol{f}_L(\boldsymbol{x}')\|_\infty \leq B^L W^L \|\boldsymbol{x} - \boldsymbol{x}'\|_\infty$. $\qquad\square$

### C.2. Covering number of the logit space of ARMs

We first characterize the output function space of the neural network without softmax operation, i.e., the unnormalized distribution parameter space, commonly referred to as logits in ARMS. This result serves as the foundation for deriving the bracketing number of the conditional probability mass function for each dimension. The derivation carefully analyzes the covering number of outputs for the entire network, which consists of the embedding layer, encoding layer, and an MLP. This analysis makes use of the previously established Lemma C.4 and Lemma C.5.

**Lemma C.6** (Covering number of the unnormalized distribution parameter vectors). *Let* $\boldsymbol{H}_\theta(\boldsymbol{x}, y) := \big[\boldsymbol{h}_{\theta,1}(\boldsymbol{x}, y) \cdots \boldsymbol{h}_{\theta,D}(\boldsymbol{x}, y)\big] : \mathcal{X} \times \mathcal{Y} \to \mathbb{R}^{M \times D}$ *with* $\boldsymbol{h}_{\theta,d}(\boldsymbol{x}, y) = \boldsymbol{f}_\omega\Big(\boldsymbol{v}_{\boldsymbol{A}_0, \boldsymbol{b}_0}^{\backslash \boldsymbol{0}_{D-d}}\big(\boldsymbol{E}_{\boldsymbol{V}_Y, \boldsymbol{V}_X}(\boldsymbol{x}, y)\big)\Big) : \mathcal{X} \times \mathcal{Y} \to \mathbb{R}^M$ *as defined in Section 4.2. Let* $\mathcal{H}_\theta = \{\boldsymbol{H}_\theta(\boldsymbol{x}, y) : \omega \in \mathcal{W}(L, W, S, B), \boldsymbol{A}_0 \in [-B, B]^{D \times d_e}, \boldsymbol{b}_0 \in [-B, B]^D, \boldsymbol{V}_X \in [0, 1]^{M \times d_e}, \boldsymbol{V}_Y \in [0, 1]^{K \times d_e}\}$ *with constants* $L, W, S, B > 0$. *Then, given any* $\epsilon > 0$, *we have*

$$
\mathcal{N}\Big(\epsilon; \mathcal{H}_\theta, \|\cdot\|_{\infty, L^\infty(\mathcal{X} \times \mathcal{Y})}\Big) \leq \left(\frac{3(L+3)(B \vee 1)^{L+2}(W+1)^L}{\epsilon}\right)^{S + D + (D + M + K)d_e}
$$

*Proof.* For any $d \in [D]$, $\boldsymbol{h}_{\theta,d}$ can be written as $\boldsymbol{f}_\omega \circ \boldsymbol{v}_{\boldsymbol{A}_0, \boldsymbol{b}_0}^{\backslash \boldsymbol{0}_{D-d}} \circ \boldsymbol{E}_{\boldsymbol{V}_Y, \boldsymbol{V}_X}$. Let the embedding space

$$
\begin{aligned}
\mathcal{G}_\alpha &:= \{\boldsymbol{G}_\theta(\boldsymbol{x}, y) = \Big[\boldsymbol{v}_{\boldsymbol{A}_0, \boldsymbol{b}_0}^{\backslash \boldsymbol{0}_D} \circ \boldsymbol{E}_{\boldsymbol{V}_Y, \boldsymbol{V}_X}(\boldsymbol{x}, y), \cdots, \boldsymbol{v}_{\boldsymbol{A}_0, \boldsymbol{b}_0}^{\backslash \boldsymbol{0}_0} \circ \boldsymbol{E}_{\boldsymbol{V}_Y, \boldsymbol{V}_X}(\boldsymbol{x}, y)\Big] : \alpha = \{\boldsymbol{A}_0, \boldsymbol{b}_0, \boldsymbol{V}_X, \boldsymbol{V}_Y\}, \\
&\quad \boldsymbol{A}_0 \in [-B, B]^{D \times d_e}, \boldsymbol{b}_0 \in [-B, B]^D, \boldsymbol{V}_X \in [0, 1]^{M \times d_e}, \boldsymbol{V}_Y \in [0, 1]^{K \times d_e}\}.
\end{aligned}
$$

where

$$
\boldsymbol{v}_{\boldsymbol{A}_0, \boldsymbol{b}_0} \circ \boldsymbol{E}_{\boldsymbol{V}_Y, \boldsymbol{V}_X}(\boldsymbol{x}, y) = \begin{bmatrix} \sigma\big(\boldsymbol{A}_0[1, :]\boldsymbol{V}_Y[y, :]^\top + \boldsymbol{b}_0[1]\big) \\ \vdots \\ \sigma\big(\boldsymbol{A}_0[D, :]\boldsymbol{V}_X[x_{D-1}, :]^\top + \boldsymbol{b}_0[D]\big) \end{bmatrix} = \sigma\Big(\mathrm{diag}\Big(\boldsymbol{A}_0\boldsymbol{E}_{\boldsymbol{V}_Y, \boldsymbol{V}_X}(\boldsymbol{x}, y)^\top\Big) + \boldsymbol{b}_0\Big) \in [0, 1]^D.
$$

Given any $\delta > 0$, we first evaluate the $\delta$-covering number of $\mathcal{G}_\alpha(\boldsymbol{x}, y)$ w.r.t. $\|\cdot\|_{\infty, L^\infty(\mathcal{X}_1)}$.

**Covering number of the embedding layer.** Let $\mathcal{S}_{\mathrm{entry}}(\boldsymbol{A})$ denote the union of value spaces of all entries in $\boldsymbol{A}$ and $\mathcal{S}_{\mathrm{entry}}(\boldsymbol{a})$ denote the union of value spaces of all entries in $\boldsymbol{a}$. We first discretize the value spaces of $\mathcal{G}_\alpha$ into $\delta$-width grids to get a finite embedding function class:

$$\mathcal{G}_{\alpha,\delta\mathbb{Z}} := \{\boldsymbol{G}_\theta \in \mathcal{G}_\alpha : \mathcal{S}_{\mathrm{entry}}(\boldsymbol{A}_0) = \mathcal{S}_{\mathrm{entry}}(\boldsymbol{b}_0) = [-B, B] \cap \delta\mathbb{Z}, \mathcal{S}_{\mathrm{entry}}(\boldsymbol{V}_Y) = \mathcal{S}_{\mathrm{entry}}(\boldsymbol{V}_X) = [0, 1] \cap \delta\mathbb{Z}\}.$$

Denote by $\|\alpha - \alpha'\|_\infty := \sup\{\|\boldsymbol{A}_0 - \boldsymbol{A}_0'\|_\infty, \|\boldsymbol{b}_0 - \boldsymbol{b}_0'\|_\infty, \|\boldsymbol{V}_Y - \boldsymbol{V}_Y'\|_\infty, \|\boldsymbol{V}_X - \boldsymbol{V}_X'\|_\infty\}$. For any $\boldsymbol{G}_\alpha \in \mathcal{G}_\alpha$ with $\alpha := \{\boldsymbol{A}_0, \boldsymbol{b}_0, \boldsymbol{V}_X, \boldsymbol{V}_Y\}$, there exists $\boldsymbol{G}_{\alpha'} \in \mathcal{G}_{\alpha,\delta\mathbb{Z}}$ with $\alpha' := \{\boldsymbol{A}_0', \boldsymbol{b}_0', \boldsymbol{V}_X', \boldsymbol{V}_Y'\}$ such that $\|\alpha - \alpha'\|_\infty \leq \delta$. Then we have for any $\boldsymbol{x}, y \in \mathcal{X} \times \mathcal{Y}$,

$$\|\boldsymbol{G}_\alpha(\boldsymbol{x}, y) - \boldsymbol{G}_{\alpha'}(\boldsymbol{x}, y)\|_\infty$$

$$= \left\| \left[ \boldsymbol{v}_{\boldsymbol{A}_0, \boldsymbol{b}_0}^{\backslash \boldsymbol{0}_D} \circ \boldsymbol{E}_{\boldsymbol{V}_Y, \boldsymbol{V}_X}(\boldsymbol{x}, y) - \boldsymbol{v}_{\boldsymbol{A}_0', \boldsymbol{b}_0'}^{\backslash \boldsymbol{0}_D} \circ \boldsymbol{E}_{\boldsymbol{V}_Y', \boldsymbol{V}_X'}(\boldsymbol{x}, y), \cdots, \boldsymbol{v}_{\boldsymbol{A}_0, \boldsymbol{b}_0}^{\backslash \boldsymbol{0}_0} \circ \boldsymbol{E}_{\boldsymbol{V}_Y, \boldsymbol{V}_X}(\boldsymbol{x}, y) - \boldsymbol{v}_{\boldsymbol{A}_0', \boldsymbol{b}_0'}^{\backslash \boldsymbol{0}_0} \circ \boldsymbol{E}_{\boldsymbol{V}_Y', \boldsymbol{V}_X'}(\boldsymbol{x}, y) \right] \right\|_\infty$$

$$\leq \left\| \boldsymbol{v}_{\boldsymbol{A}_0, \boldsymbol{b}_0} \circ \boldsymbol{E}_{\boldsymbol{V}_Y, \boldsymbol{V}_X}(\boldsymbol{x}, y) - \boldsymbol{v}_{\boldsymbol{A}_0', \boldsymbol{b}_0'} \circ \boldsymbol{E}_{\boldsymbol{V}_Y', \boldsymbol{V}_X'}(\boldsymbol{x}, y) \right\|_\infty$$

$$= \left\| \sigma\left( \mathrm{diag}\left( \boldsymbol{A}_0 \boldsymbol{E}_{\boldsymbol{V}_Y, \boldsymbol{V}_X}(\boldsymbol{x}, y)^\top \right) + \boldsymbol{b}_0 \right) - \sigma\left( \mathrm{diag}\left( \boldsymbol{A}_0' \boldsymbol{E}_{\boldsymbol{V}_{Y'}, \boldsymbol{V}_{X'}}(\boldsymbol{x}, y)^\top \right) - \boldsymbol{b}_0' \right) \right\|_\infty$$

$$\leq \left\| \mathrm{diag}\left( \boldsymbol{A}_0 \boldsymbol{E}_{\boldsymbol{V}_Y, \boldsymbol{V}_X}(\boldsymbol{x}, y)^\top \right) + \boldsymbol{b}_0 - \mathrm{diag}\left( \boldsymbol{A}_0' \boldsymbol{E}_{\boldsymbol{V}_{Y'}, \boldsymbol{V}_{X'}}(\boldsymbol{x}, y)^\top \right) - \boldsymbol{b}_0' \right\|_\infty$$

$$(\sigma(\cdot) \text{ is 1-Lipschitz continuous for Lemma C.2})$$

$$\leq \sup_{d \in [D]} \left| \boldsymbol{A}_0[d, :] \boldsymbol{E}_{\boldsymbol{V}_Y, \boldsymbol{V}_X}(\boldsymbol{x}, y)[d, :]^\top - \boldsymbol{A}_0'[d, :] \boldsymbol{E}_{\boldsymbol{V}_Y', \boldsymbol{V}_X'}(\boldsymbol{x}, y)[d, :]^\top \right| + \|\boldsymbol{b}_0 - \boldsymbol{b}_0'\|_\infty$$

$$\leq \sup_{d \in [D]} \left| (\boldsymbol{A}_0[d, :] - \boldsymbol{A}_0'[d, :]) \boldsymbol{E}_{\boldsymbol{V}_Y, \boldsymbol{V}_X}(\boldsymbol{x}, y)[d, :]^\top \right| + \left| \boldsymbol{A}_0'[d, :] (\boldsymbol{E}_{\boldsymbol{V}_Y, \boldsymbol{V}_X}(\boldsymbol{x}, y)[d, :]^\top - \boldsymbol{E}_{\boldsymbol{V}_Y', \boldsymbol{V}_X'}(\boldsymbol{x}, y)[d, :]^\top) \right| + \delta$$

$$\leq \sup_{d \in [D]} d_{\mathrm{e}}\delta \left\| \boldsymbol{E}_{\boldsymbol{V}_Y, \boldsymbol{V}_X}(\boldsymbol{x}, y)[d, :] \right\|_\infty + d_{\mathrm{e}}B \left\| \boldsymbol{E}_{\boldsymbol{V}_Y, \boldsymbol{V}_X}(\boldsymbol{x}, y)[d, :] - \boldsymbol{E}_{\boldsymbol{V}_Y', \boldsymbol{V}_X'}(\boldsymbol{x}, y)[d, :] \right\|_\infty + \delta$$

$$(\boldsymbol{A}_0 \in [-B, B]^{D \times d_{\mathrm{e}}}, \|\boldsymbol{A}_0 - \boldsymbol{A}_0'\|_\infty \leq \delta)$$

$$= \begin{cases} d_{\mathrm{e}}\delta \left\| \boldsymbol{V}_Y[y, :] \right\|_\infty + d_{\mathrm{e}}B \left\| \boldsymbol{V}_Y[y, :] - \boldsymbol{V}_Y'[y, :] \right\|_\infty + \delta, & d = 1, \\ \sup_{d \in [D]} d_{\mathrm{e}}\delta \left\| \boldsymbol{V}_X[x_{d-1}, :] \right\|_\infty + d_{\mathrm{e}}B \left\| \boldsymbol{V}_X[x_{d-1}, :] - \boldsymbol{V}_X'[x_{d-1}, :] \right\|_\infty + \delta, & d = 2, \ldots, D, \end{cases}$$

$$\leq d_{\mathrm{e}}\delta + d_{\mathrm{e}}B\delta + \delta \qquad (\|\boldsymbol{V}_Y\|_\infty \leq 1, \|\boldsymbol{V}_X\|_\infty \leq 1, \|\boldsymbol{V}_Y - \boldsymbol{V}_Y'\|_\infty \leq \delta, \|\boldsymbol{V}_X - \boldsymbol{V}_X'\|_\infty \leq \delta)$$

$$= (1 + d_{\mathrm{e}} + d_{\mathrm{e}}B)\delta.$$

Therefore, $\mathcal{G}_{\alpha,\delta\mathbb{Z}}$ is an $(1 + d_{\mathrm{e}} + d_{\mathrm{e}}B)\delta$-cover of $\mathcal{G}_\alpha$ w.r.t. $\|\cdot\|_{\infty, L^\infty(\mathcal{X} \times \mathcal{Y})}$ and thus we have

$$\mathcal{N}\left( (1 + d_{\mathrm{e}} + d_{\mathrm{e}}B)\delta; \mathcal{G}_\alpha, \|\cdot\|_{\infty, L^\infty(\mathcal{X} \times \mathcal{Y})} \right) \leq |\mathcal{G}_{\alpha,\delta\mathbb{Z}}|$$

$$= \left| \{\alpha = (\boldsymbol{A}_0, \boldsymbol{b}_0, \boldsymbol{V}_Y, \boldsymbol{V}_X) : \mathcal{S}_{\mathrm{entry}}(\boldsymbol{A}_0) = \mathcal{S}_{\mathrm{entry}}(\boldsymbol{b}_0) = [-B, B] \cap \delta\mathbb{Z}, \mathcal{S}_{\mathrm{entry}}(\boldsymbol{V}_Y) = \mathcal{S}_{\mathrm{entry}}(\boldsymbol{V}_X) = [0, 1] \cap \delta\mathbb{Z}\} \right|$$

$$\leq \left( \frac{2B}{\delta} + 1 \right)^{Dd_{\mathrm{e}} + D} \left( \frac{1}{\delta} + 1 \right)^{Md_{\mathrm{e}} + Kd_{\mathrm{e}}}. \tag{20}$$

**Composition of an MLP.** Let $\mathcal{C}_{\mathcal{F}}$ be an $\epsilon_{\mathcal{F}} = L(B \vee 1)^{L-1}(W + 1)^L\delta$-cover of $\mathcal{F}\{L, W, S, B\}$ w.r.t. $\|\cdot\|_{\infty, L^\infty([0,1]^D)}$ such that $|\mathcal{C}_{\mathcal{F}}| = \mathcal{N}\left( \epsilon_{\mathcal{F}}; \mathcal{F}(L, W, B, S), \|\cdot\|_{\infty, L^\infty([0,1]^D)} \right)$ and $\mathcal{C}_{\mathcal{G}}$ be an $\epsilon_{\mathcal{G}} = (1 + d_{\mathrm{e}} + d_{\mathrm{e}}B)\delta$-cover of $\mathcal{G}_\alpha$ w.r.t. $\|\cdot\|_{\infty, L^\infty([0,1]^D)}$ such that $|\mathcal{C}_{\mathcal{G}}| = \mathcal{N}\left( \epsilon_{\mathcal{G}}; \mathcal{G}_\alpha, \|\cdot\|_{\infty, L^\infty(\mathcal{X} \times \mathcal{Y})} \right)$. For any $\boldsymbol{H}_\theta := \left[ \boldsymbol{f}_\omega \circ \boldsymbol{G}_\alpha[:, 1] \quad \cdots \quad \boldsymbol{f}_\omega \circ \boldsymbol{G}_\alpha[:, D] \right] \in \mathcal{H}_\theta$, there exists $\boldsymbol{f}_\omega' \in \mathcal{C}_{\mathcal{F}}$ and $\boldsymbol{G}_\alpha' \in \mathcal{C}_{\mathcal{G}}$ such that

$$\forall \boldsymbol{v} \in [0, 1]^D, \|\boldsymbol{f}_\omega(\boldsymbol{v}) - \boldsymbol{f}_\omega'(\boldsymbol{v})\|_\infty \leq \epsilon_{\mathcal{F}}, \text{ and } \forall \boldsymbol{x}, y \in \mathcal{X} \times \mathcal{Y}, \|\boldsymbol{G}_\alpha(\boldsymbol{x}, y) - \boldsymbol{G}_\alpha'(\boldsymbol{x}, y)\|_\infty \leq \epsilon_{\mathcal{G}}.$$

Let $\boldsymbol{H}_\theta' \coloneqq \left[\boldsymbol{f}_\omega' \circ \boldsymbol{G}_\alpha'[:,1] \; \cdots \; \boldsymbol{f}_\omega' \circ \boldsymbol{G}_\alpha'[:,D]\right]$. We have for all $x, y \in \mathcal{X} \times \mathcal{Y}$,

$$
\begin{aligned}
\|\boldsymbol{H}_\theta - \boldsymbol{H}_\theta'\|_\infty &= \left\|\left[\boldsymbol{f}_\omega \circ \boldsymbol{G}_\alpha[:,1] - \boldsymbol{f}_\omega' \circ \boldsymbol{G}_\alpha'[:,1], \; \cdots, \; \boldsymbol{f}_\omega \circ \boldsymbol{G}_\alpha[:,D] - \boldsymbol{f}_\omega' \circ \boldsymbol{G}_\alpha'[:,D]\right]\right\|_\infty \\
&= \sup_d \left\|\boldsymbol{f}_\omega \circ \boldsymbol{G}_\alpha[:,d] - \boldsymbol{f}_\omega' \circ \boldsymbol{G}_\alpha'[:,d]\right\|_\infty \\
&\leq \sup_d \left\{ \left\|\boldsymbol{f}_\omega \circ \boldsymbol{G}_\alpha[:,d] - \boldsymbol{f}_\omega' \circ \boldsymbol{G}_\alpha[:,d]\right\| + \left\|\boldsymbol{f}_\omega' \circ \boldsymbol{G}_\alpha[:,d] - \boldsymbol{f}_\omega' \circ \boldsymbol{G}_\alpha'[:,d]\right\|_\infty \right\} \\
&\leq \sup_d \left\{ \epsilon_\mathcal{F} + B^L W^L \epsilon_\mathcal{G} \right\} \qquad (\boldsymbol{f}_\omega \text{ is } B^L W^L\text{-Lipschitz continuous as in Lemma C.5}) \\
&= \epsilon_\mathcal{F} + B^L W^L \epsilon_\mathcal{G}.
\end{aligned}
$$

Therefore, $\mathcal{C}_\mathcal{H} \coloneqq \{\boldsymbol{H}_\theta' \coloneqq \left[\boldsymbol{f}_\omega' \circ \boldsymbol{G}_\alpha'[:,1] \; \cdots \; \boldsymbol{f}_\omega' \circ \boldsymbol{G}_\alpha'[:,D]\right] : \boldsymbol{f}_\omega' \in \mathcal{C}_\mathcal{F}, \boldsymbol{G}_\alpha' \in \mathcal{C}_\mathcal{G}\}$ is an $\epsilon_\mathcal{F} + B^L W^L \epsilon_\mathcal{G}$-cover of $\mathcal{H}_\theta$ w.r.t. $\|\cdot\|_{\infty, L^\infty(\mathcal{X} \times \mathcal{Y})}$, and thus

$$
\mathcal{N}\left( \left( L(B \vee 1)^{L-1}(W+1)^L + B^L W^L (1 + d_\mathrm{e} + d_\mathrm{e} B)\right)\delta; \mathcal{H}_\theta, \|\cdot\|_{\infty, L^\infty(\mathcal{X} \times \mathcal{Y})}\right)
$$

$$
\leq |\mathcal{C}_\mathcal{H}| \leq |\mathcal{C}_\mathcal{F}||\mathcal{C}_\mathcal{G}|
$$

$$
= \mathcal{N}\left( L(B \vee 1)^{L-1}(W+1)^L \delta; \mathcal{F}(L,W,B,S), \|\cdot\|_{\infty, L^\infty([0,1]^D)}\right) \mathcal{N}\left( (1 + d_\mathrm{e} + d_\mathrm{e} B)\delta; \mathcal{G}_\alpha, \|\cdot\|_{\infty, L^\infty(\mathcal{X} \times \mathcal{Y})}\right)
$$

$$
\leq \left(\frac{2B}{\delta} + 1\right)^S \left(\frac{2B}{\delta} + 1\right)^{D d_\mathrm{e} + D} \left(\frac{1}{\delta} + 1\right)^{M d_\mathrm{e} + K d_\mathrm{e}} \qquad \text{(Lemma C.4 and Equation (20))}
$$

$$
\leq \left(\frac{(2B \vee 1)}{\delta} + 1\right)^{S + D d_\mathrm{e} + D + M d_\mathrm{e} + K d_\mathrm{e}}
$$

$$
\leq \left(\frac{3(B \vee 1)}{\delta}\right)^{S + D + (D + M + K) d_\mathrm{e}}. \qquad \left(\frac{(B \vee 1)}{\delta} \geq 1\right)
$$

Taking $\epsilon = \left( L(B \vee 1)^{L-1}(W+1)^L + B^L W^L(1 + d_\mathrm{e} + d_\mathrm{e} B)\right)\delta$, we have

$$
\begin{aligned}
\mathcal{N}\left(\epsilon; \mathcal{H}_\theta, \|\cdot\|_{\infty, L^\infty(\mathcal{X} \times \mathcal{Y})}\right) &\leq \left(\frac{3(B \vee 1)\left(L(B \vee 1)^{L-1}(W+1)^L + B^L W^L(1 + d_\mathrm{e} + d_\mathrm{e} B)\right)}{\epsilon}\right)^{S + D + (D + M + K) d_\mathrm{e}} \\
&\leq \left(\frac{3(B \vee 1)\left(L(B \vee 1)^{L-1}(W+1)^L d_\mathrm{e}(B \vee 1) + 3 B^L W^L d_\mathrm{e}(B \vee 1)\right)}{\epsilon}\right)^{S + D + (D + M + K) d_\mathrm{e}} \\
&\qquad\qquad\qquad\qquad\qquad\qquad\qquad\qquad (d_\mathrm{e}, (B \vee 1) \geq 1) \\
&\leq \left(\frac{3(B \vee 1)\left(L(B \vee 1)^{L+1}(W+1)^L d_\mathrm{e} + 3(B \vee 1)^{L+1}(W+1)^L d_\mathrm{e}\right)}{\epsilon}\right)^{S + D + (D + M + K) d_\mathrm{e}} \\
&= \left(\frac{3(L+3)(B \vee 1)^{L+2}(W+1)^L}{\epsilon}\right)^{S + D + (D + M + K) d_\mathrm{e}}.
\end{aligned}
$$

$\qquad\qquad\qquad\qquad\qquad\qquad\qquad\qquad\qquad\qquad\qquad\qquad\qquad\qquad\qquad\qquad\qquad\qquad\qquad \square$

## C.3. Bracketing number of conditional probability space on each dimension

**Lemma C.7** (Bracketing number of conditional probability space on each dimension). *Let* $\boldsymbol{P}_\theta(x, y) \coloneqq \left[\boldsymbol{\rho}_\theta(y) \; \cdots \; \boldsymbol{\rho}_\theta(x_{<D}, y)\right] = \mathrm{softmax}\left(\boldsymbol{H}_\theta(x, y)\right) : \mathcal{X} \times \mathcal{Y} \to [0,1]^{M \times D}$ *where* $\mathrm{softmax}(\boldsymbol{H})$ *denotes element-wise soft-max operation that* $\mathrm{softmax}(\boldsymbol{H})[m:d] = \frac{e^{\boldsymbol{H}[m,d]}}{\sum_{i=1}^M e^{\boldsymbol{H}[i,d]}}$. *Given a class of autoregressive conditional distributions that* $\mathcal{P}_{X|Y} = \left\{ p_\theta(x|y) = p(x_1; \boldsymbol{\rho}_\theta(y)) \cdots p(x_D; \boldsymbol{\rho}_\theta(x_{<D}, y)) : \boldsymbol{H}_\theta \in \mathcal{H}_\theta \right\}$ *with* $p_\theta(x|y)$ *as defined in Section 4.2, then for any* $0 < \epsilon \leq 1$, *it holds that*

$$
\mathcal{N}_{[]}\left(\epsilon; \mathcal{P}_{X|Y}, L^1(\mathcal{X})\right) \leq \mathcal{N}\left(\frac{\epsilon}{8ed}; \mathcal{H}_\theta, \|\cdot\|_{\infty, L^\infty(\mathcal{X} \times \mathcal{Y})}\right).
$$

*Proof.* Let $\mathcal{C}_{\mathcal{H}}$ be an $\epsilon_{\mathcal{H}}$-cover of $\mathcal{H}_\theta$ that $|\mathcal{C}_{\mathcal{H}}| = \mathcal{N}\Big(\epsilon_{\mathcal{H}}; \mathcal{H}_\theta, \|\cdot\|_{\infty, L^\infty(\mathcal{X} \times \mathcal{Y})}\Big)$. According to the definition in Section 4.2,

$$
p_\theta(\boldsymbol{x}|y) = p(x_1; \boldsymbol{\rho}_\theta(y)) \cdots p(x_D; \boldsymbol{\rho}_\theta(\boldsymbol{x}_{<D}, y))
$$

$$
= p(x_1; \boldsymbol{P}_\theta[:, 1]) \cdots p(x_D; \boldsymbol{P}_\theta(\boldsymbol{x}, y)[:, D]) = \prod_{d=1}^{D} p(x_d; \boldsymbol{P}_\theta(\boldsymbol{x}, y)[:, d]) = \prod_{d=1}^{D} \boldsymbol{P}_\theta(\boldsymbol{x}, y)[x_d, d]
$$

$$
= \prod_{d=1}^{D} \prod_{m=1}^{M} \big(\mathrm{softmax}(\boldsymbol{H}_\theta(\boldsymbol{x}, y))[x_d, d]\big)^{\mathbb{I}(x_d = m)} = \prod_{d=1}^{D} \prod_{m=1}^{M} \left(\frac{e^{\boldsymbol{H}_\theta(\boldsymbol{x}, y)[m, d]}}{\sum_{i=1}^{M} e^{\boldsymbol{H}_\theta(\boldsymbol{x}, y)[i, d]}}\right)^{\mathbb{I}(x_d = m)}.
$$

Then, for any $p_\theta \in \mathcal{P}_{X|Y}$, there exists $\boldsymbol{H}'_\theta \in \mathcal{C}_{\mathcal{H}}$ such that for all $\boldsymbol{x}, y \in \mathcal{X} \times \mathcal{Y}$, $\|\boldsymbol{H}_\theta(\boldsymbol{x}, y) - \boldsymbol{H}'_\theta(\boldsymbol{x}, y)\| \leq \epsilon_{\mathcal{H}}$ which equals to $\forall m \in [M], d \in [D], \boldsymbol{H}'_\theta(\boldsymbol{x}, y)[m, d] - \epsilon_{\mathcal{H}} \leq \boldsymbol{H}_\theta(\boldsymbol{x}, y)[m, d] \leq \boldsymbol{H}'_\theta(\boldsymbol{x}, y)[m, d] + \epsilon_{\mathcal{H}}$. Let $p'_\theta(\boldsymbol{x}|y) = \prod_{d=1}^{D} \prod_{m=1}^{M} \left(\frac{e^{\boldsymbol{H}'_\theta(\boldsymbol{x}, y)[m, d] + \epsilon_{\mathcal{H}}}}{\sum_{i=1}^{M} e^{\boldsymbol{H}'_\theta(\boldsymbol{x}, y)[i, d] - \epsilon_{\mathcal{H}}}}\right)^{\mathbb{I}(x_d = m)}$ and denote $\boldsymbol{P}'_\theta(\boldsymbol{x}, y)[x_d, d] := \prod_{m=1}^{M} \left(\frac{e^{\boldsymbol{H}'_\theta(\boldsymbol{x}, y)[m, d] + \epsilon_{\mathcal{H}}}}{\sum_{i=1}^{M} e^{\boldsymbol{H}'_\theta(\boldsymbol{x}, y)[i, d] - \epsilon_{\mathcal{H}}}}\right)^{\mathbb{I}(x_d = m)}$. We immediately have: for all $\boldsymbol{x}, y \in \mathcal{X} \times \mathcal{Y}$, $p'_\theta(\boldsymbol{x}|y) \geq p_\theta(\boldsymbol{x}|y)$, since $\forall m \in [M], d \in [D], e^{\boldsymbol{H}'_\theta(\boldsymbol{x}, y)[m, d] + \epsilon_{\mathcal{H}}} \geq e^{\boldsymbol{H}_\theta(\boldsymbol{x}, y)[m, d]}$ and $e^{\boldsymbol{H}'_\theta(\boldsymbol{x}, y)[i, d] - \epsilon_{\mathcal{H}}} \leq e^{\boldsymbol{H}_\theta(\boldsymbol{x}, y)[i, d]}$. Moreover, we have for all $\boldsymbol{x}, y \in \mathcal{X} \times \mathcal{Y}$,

$$
\boldsymbol{P}'_\theta(\boldsymbol{x}, y)[x_d, d] - \boldsymbol{P}_\theta(\boldsymbol{x}, y)[x_d, d]
$$

$$
= \prod_{m=1}^{M} \left(\frac{e^{\boldsymbol{H}'_\theta(\boldsymbol{x}, y)[m, d] + \epsilon_{\mathcal{H}}}}{\sum_{i=1}^{M} e^{\boldsymbol{H}'_\theta(\boldsymbol{x}, y)[i, d] - \epsilon_{\mathcal{H}}}}\right)^{\mathbb{I}(x_d = m)} - \prod_{m=1}^{M} \left(\frac{e^{\boldsymbol{H}_\theta(\boldsymbol{x}, y)[m, d]}}{\sum_{i=1}^{M} e^{\boldsymbol{H}_\theta(\boldsymbol{x}, y)[i, d]}}\right)^{\mathbb{I}(x_d = m)}
$$

$$
= \prod_{m=1}^{M} \left(\frac{e^{\boldsymbol{H}'_\theta(\boldsymbol{x}, y)[m, d] + 2\epsilon_{\mathcal{H}}}}{\sum_{i=1}^{M} e^{\boldsymbol{H}'_\theta(\boldsymbol{x}, y)[i, d]}} - \frac{e^{\boldsymbol{H}_\theta(\boldsymbol{x}, y)[m, d]}}{\sum_{i=1}^{M} e^{\boldsymbol{H}_\theta(\boldsymbol{x}, y)[i, d]}}\right)^{\mathbb{I}(x_d = m)}
$$

$$
= \prod_{m=1}^{M} \left(\frac{\left(e^{\boldsymbol{H}'_\theta(\boldsymbol{x}, y)[m, d] + 2\epsilon_{\mathcal{H}}} - e^{\boldsymbol{H}_\theta(\boldsymbol{x}, y)[m, d]}\right) \sum_{i=1}^{M} e^{\boldsymbol{H}_\theta(\boldsymbol{x}, y)[i, d]} + e^{\boldsymbol{H}_\theta(\boldsymbol{x}, y)[m, d]} \sum_{i=1}^{M} \left(e^{\boldsymbol{H}_\theta(\boldsymbol{x}, y)[i, d]} - e^{\boldsymbol{H}'_\theta(\boldsymbol{x}, y)[i, d]}\right)}{\sum_{i=1}^{M} e^{\boldsymbol{H}'_\theta(\boldsymbol{x}, y)[i, d]} \sum_{i=1}^{M} e^{\boldsymbol{H}_\theta(\boldsymbol{x}, y)[i, d]}}\right)^{\mathbb{I}(x_d = m)}
$$

$$
\leq \prod_{m=1}^{M} \left(\frac{e^{\boldsymbol{H}'_\theta(\boldsymbol{x}, y)[m, d] + 2\epsilon_{\mathcal{H}}} 3\epsilon_{\mathcal{H}} \sum_{i=1}^{M} e^{\boldsymbol{H}_\theta(\boldsymbol{x}, y)[i, d]} + e^{\boldsymbol{H}_\theta(\boldsymbol{x}, y)[m, d]} \sum_{i=1}^{M} e^{\boldsymbol{H}'_\theta(\boldsymbol{x}, y)[i, d] + \epsilon_{\mathcal{H}}} \epsilon_{\mathcal{H}}}{\sum_{i=1}^{M} e^{\boldsymbol{H}'_\theta(\boldsymbol{x}, y)[i, d]} \sum_{i=1}^{M} e^{\boldsymbol{H}_\theta(\boldsymbol{x}, y)[i, d]}}\right)^{\mathbb{I}(x_d = m)}
$$

$$
(|e^a - e^b| \leq e^{a \vee b}|a - b| \text{ and } \|\boldsymbol{H}_\theta(\boldsymbol{x}, y) - \boldsymbol{H}'_\theta(\boldsymbol{x}, y)\| \leq \epsilon_{\mathcal{H}})
$$

$$
\leq \prod_{m=1}^{M} \left(\frac{e^{\boldsymbol{H}_\theta(\boldsymbol{x}, y)[m, d] + 3\epsilon_{\mathcal{H}}} 3\epsilon_{\mathcal{H}} \sum_{i=1}^{M} e^{\boldsymbol{H}'_\theta(\boldsymbol{x}, y)[i, d] + \epsilon_{\mathcal{H}}} + e^{\boldsymbol{H}_\theta(\boldsymbol{x}, y)[m, d]} \sum_{i=1}^{M} e^{\boldsymbol{H}'_\theta(\boldsymbol{x}, y)[i, d] + \epsilon_{\mathcal{H}}} \epsilon_{\mathcal{H}}}{\sum_{i=1}^{M} e^{\boldsymbol{H}'_\theta(\boldsymbol{x}, y)[i, d]} \sum_{i=1}^{M} e^{\boldsymbol{H}_\theta(\boldsymbol{x}, y)[i, d]}}\right)^{\mathbb{I}(x_d = m)}
$$

$$
(\|\boldsymbol{H}_\theta(\boldsymbol{x}, y) - \boldsymbol{H}'_\theta(\boldsymbol{x}, y)\| \leq \epsilon_{\mathcal{H}})
$$

$$
= \prod_{m=1}^{M} \left(\frac{e^{\boldsymbol{H}_\theta(\boldsymbol{x}, y)[m, d] + 3\epsilon_{\mathcal{H}}} 3\epsilon_{\mathcal{H}} e^{\epsilon_{\mathcal{H}}} + e^{\boldsymbol{H}_\theta(\boldsymbol{x}, y)[m, d]} e^{\epsilon_{\mathcal{H}}} \epsilon_{\mathcal{H}}}{\sum_{i=1}^{M} e^{\boldsymbol{H}_\theta(\boldsymbol{x}, y)[i, d]}}\right)^{\mathbb{I}(x_d = m)}
$$

$$
(\|\boldsymbol{H}_\theta(\boldsymbol{x}, y) - \boldsymbol{H}'_\theta(\boldsymbol{x}, y)\| \leq \epsilon_{\mathcal{H}})
$$

$$
= \left(3\epsilon_{\mathcal{H}} e^{4\epsilon_{\mathcal{H}}} + e^{\epsilon_{\mathcal{H}}} \epsilon_{\mathcal{H}}\right) \prod_{m=1}^{M} \left(\frac{e^{\boldsymbol{H}_\theta(\boldsymbol{x}, y)[m, d]}}{\sum_{i=1}^{M} e^{\boldsymbol{H}_\theta(\boldsymbol{x}, y)[i, d]}}\right)^{\mathbb{I}(x_d = m)}
$$

$$
= \left(3\epsilon_{\mathcal{H}} e^{4\epsilon_{\mathcal{H}}} + e^{\epsilon_{\mathcal{H}}} \epsilon_{\mathcal{H}}\right) \boldsymbol{P}_\theta(\boldsymbol{x}, y)[x_d, d] \leq 4\epsilon_{\mathcal{H}} e^{4\epsilon_{\mathcal{H}}} \boldsymbol{P}_\theta(\boldsymbol{x}, y)[x_d, d]. \tag{21}
$$

Given any $y \in \mathcal{Y}$, denoting $a_d := \sum_{\boldsymbol{x}_{\leq d} \in [M]^d} |p'_\theta(\boldsymbol{x}_{\leq d}|y) - p_\theta(\boldsymbol{x}_{\leq d}|y)|$ for $d = 1, \ldots, D$, we have the following recursive

formula for $\{a_d\}_{d\in[D]}$:

$$a_d = \sum_{\boldsymbol{x}_{\leq d}\in[M]^d} \left|p'_\theta(\boldsymbol{x}_{\leq d}|y) - p_\theta(\boldsymbol{x}_{\leq d}|y)\right| = \sum_{\boldsymbol{x}_{\leq d}\in[M]^d} \left|\prod_{j=1}^{d} \boldsymbol{P}'_\theta(\boldsymbol{x},y)[x_j,j] - \prod_{j=1}^{d} \boldsymbol{P}_\theta(\boldsymbol{x},y)[x_j,j]\right|$$

$$= \sum_{\boldsymbol{x}_{\leq d}\in[M]^d} \left|\left(\prod_{j=1}^{d-1} \boldsymbol{P}'_\theta(\boldsymbol{x},y)[x_j,j] - \prod_{j=1}^{d-1} \boldsymbol{P}_\theta(\boldsymbol{x},y)[x_j,j]\right) \boldsymbol{P}'_\theta(\boldsymbol{x},y)[x_d,d]\right.$$

$$+ \sum_{\boldsymbol{x}_{\leq d}\in[M]^d} \left|\prod_{j=1}^{d-1} \boldsymbol{P}_\theta(\boldsymbol{x},y)[x_j,j]\big(\boldsymbol{P}'_\theta(\boldsymbol{x},y)[x_d,d] - \boldsymbol{P}_\theta(\boldsymbol{x},y)[x_d,d]\big)\right|$$

$$\leq \sum_{\boldsymbol{x}_{\leq d-1}\in[M]^{d-1}} \left|p'_\theta(\boldsymbol{x}_{\leq d-1}|y) - p_\theta(\boldsymbol{x}_{\leq d-1}|y)\right| \left(\sum_{x_d\in[M]} \boldsymbol{P}'_\theta(\boldsymbol{x},y)[x_d,d]\right) + 4\epsilon_{\mathcal{H}}e^{4\epsilon_{\mathcal{H}}} \sum_{\boldsymbol{x}_{\leq d}\in[M]^d} \prod_{j=1}^{d} \boldsymbol{P}_\theta(\boldsymbol{x},y)[x_j,j]$$

(Equation (21))

$$= a_{d-1} \sum_{x_d\in[M]} \boldsymbol{P}'_\theta(\boldsymbol{x},y)[x_d,d] + 4\epsilon_{\mathcal{H}}e^{4\epsilon_{\mathcal{H}}} \sum_{\boldsymbol{x}_{\leq d}\in[M]^d} p_\theta(\boldsymbol{x}_{\leq d-1}|y)$$

$$= e^{2\epsilon_{\mathcal{H}}} a_{d-1} + 4\epsilon_{\mathcal{H}}e^{4\epsilon_{\mathcal{H}}} \qquad (\textstyle\sum_{x_d\in[M]} \boldsymbol{P}'_\theta(\boldsymbol{x},y)[x_d,d] = e^{2\epsilon_{\mathcal{H}}} \text{ and } \sum_{\boldsymbol{x}_{\leq d}\in[M]^d} p_\theta(\boldsymbol{x}_{\leq d}|y) = 1)$$

According to this recursive relation, and

$$a_1 = \sum_{x_1\in[M]} \left|p'_\theta(x_1|y) - p_\theta(x_1|y)\right| = \sum_{x_1\in[M]} \left|\boldsymbol{P}'_\theta(\boldsymbol{x},y)[x_1,1] - \boldsymbol{P}_\theta(\boldsymbol{x},y)[x_1,1]\right|$$

$$\leq \sum_{x_1\in[M]} 4\epsilon_{\mathcal{H}}e^{4\epsilon_{\mathcal{H}}} \boldsymbol{P}_\theta(\boldsymbol{x},y)[x_d,d] = 4\epsilon_{\mathcal{H}}e^{4\epsilon_{\mathcal{H}}}, \qquad \text{(Equation (21))}$$

we have $a_d \leq 4\epsilon_{\mathcal{H}}e^{4\epsilon_{\mathcal{H}}} \frac{e^{2d\epsilon_{\mathcal{H}}}-1}{e^{2\epsilon_{\mathcal{H}}}-1}$. Therefore,

$$\sum_{\boldsymbol{x}\in[M]^D} \left|p'_\theta(\boldsymbol{x}|y) - p_\theta(\boldsymbol{x}|y)\right| = \sum_{\boldsymbol{x}_{\leq D}\in[M]^D} \left|p'_\theta(\boldsymbol{x}_{\leq D}|y) - p_\theta(\boldsymbol{x}_{\leq D}|y)\right|$$

$$= a_d \leq 4\epsilon_{\mathcal{H}}e^{4\epsilon_{\mathcal{H}}} \frac{e^{2d\epsilon_{\mathcal{H}}}-1}{e^{2\epsilon_{\mathcal{H}}}-1} \leq 4\epsilon_{\mathcal{H}}e^{4\epsilon_{\mathcal{H}}} \frac{e^{2d\epsilon_{\mathcal{H}}}-1}{2\epsilon_{\mathcal{H}}} = 2e^{4\epsilon_{\mathcal{H}}}\left(e^{2d\epsilon_{\mathcal{H}}}-1\right).$$

Suppose that $\epsilon_{\mathcal{H}} \in (0,\frac{1}{4d})$, we have $e^{4\epsilon_{\mathcal{H}}} \leq e^{\frac{1}{d}} \leq e$ and $e^{2d\epsilon_{\mathcal{H}}}-1 \leq 4d\epsilon_{\mathcal{H}}$ as $e^x \leq 1+2x$ for $x\in[0,1]$, and thus $2e^{4\epsilon_{\mathcal{H}}}\left(e^{2d\epsilon_{\mathcal{H}}}-1\right) \leq 8ed\epsilon$. Therefore, given any $y\in\mathcal{Y}$, the $L^1(\mathcal{X})$ distance between $p'_\theta(\cdot|y)$ and $p_\theta(\cdot|y)$ can be bounded as

$$\|p'_\theta(\cdot|y) - p_\theta(\cdot|y)\|_{L^1(\mathcal{X})} = \sum_{\boldsymbol{x}\in[M]^D} \left|p'_\theta(\boldsymbol{x}|y) - p_\theta(\boldsymbol{x}|y)\right| \leq 2e^{4\epsilon_{\mathcal{H}}}\left(e^{2d\epsilon_{\mathcal{H}}}-1\right) = 8ed\epsilon_{\mathcal{H}}.$$

Therefore, $\mathcal{B}_{\mathcal{P}} := \left\{p'_\theta(\boldsymbol{x}|y) = \prod_{d=1}^{D}\prod_{m=1}^{M}\left(\frac{e^{\boldsymbol{H}'_\theta(\boldsymbol{x},y)[m,d]+\epsilon_{\mathcal{H}}}}{\sum_{i=1}^{M} e^{\boldsymbol{H}'_\theta(\boldsymbol{x},y)[i,d]-\epsilon_{\mathcal{H}}}}\right)^{\mathbb{I}(x_d=m)} : \boldsymbol{H}'_\theta\in\mathcal{C}_{\mathcal{H}}\right\}$ is an $8ed\epsilon_{\mathcal{H}}$-upper bracket w.r.t. $L^1(\mathcal{X})$ of $\mathcal{P}_{X|Y}$, and we have

$$\mathcal{N}_{[]}\left(8ed\epsilon_{\mathcal{H}}; \mathcal{P}_{X|Y}, L^1(\mathcal{X})\right) \leq |\mathcal{B}_{\mathcal{P}}| \leq |\mathcal{C}_{\mathcal{H}}| = \mathcal{N}\left(\epsilon_{\mathcal{H}}; \mathcal{H}_\theta, \|\cdot\|_{\infty,L^\infty(\mathcal{X}\times\mathcal{Y})}\right).$$

Letting $8ed\epsilon_{\mathcal{H}} = \epsilon \in (0,1]$, we have $\epsilon_{\mathcal{H}} \leq \frac{1}{8ed} < \frac{1}{4d}$, and thus

$$\mathcal{N}_{[]}\left(\epsilon; \mathcal{P}_{X|Y}, L^1(\mathcal{X})\right) \leq \mathcal{N}\left(\frac{\epsilon}{8ed}; \mathcal{H}_\theta, \|\cdot\|_{\infty,L^\infty(\mathcal{X}\times\mathcal{Y})}\right).$$

$\square$

## C.4. Proof of Theorem 4.3

Based on the relation between the bracketing number of conditional distribution space $\mathcal{P}_{X|Y}$ and the covering number of output logit space of the neural network $\mathcal{H}_\theta$ derived in previous lemmas, we obtain the final result.

*Proof of Theorem 4.3.* With conditional distributions as defined in Equation (6), we have

$$\mathcal{P}_{X|Y}^{\text{multi}} = \left\{ p_\theta(\boldsymbol{x}|y) = p\big(x_1; \boldsymbol{\rho}_\theta(y)\big) \cdots p\big(x_D; \boldsymbol{\rho}_\theta(\boldsymbol{x}_{<D}, y)\big) : \boldsymbol{H}_\theta \in \mathcal{H}_\theta \right\}$$

where $\boldsymbol{H}_\theta(\boldsymbol{x}, y) = \begin{bmatrix} \boldsymbol{h}_{\theta,1}(\boldsymbol{x}, y) & \cdots & \boldsymbol{h}_{\theta,D}(\boldsymbol{x}, y) \end{bmatrix} : \mathcal{X} \times \mathcal{Y} \to \mathbb{R}^{M \times D}$ with $\boldsymbol{h}_{\theta,d}(\boldsymbol{x}, y) = \boldsymbol{f}_\omega\Big(\boldsymbol{v}_{\boldsymbol{A}_0, \boldsymbol{b}_0}^{\backslash \boldsymbol{0}_{D-d}}\big(\boldsymbol{E}_{\boldsymbol{V}_Y, \boldsymbol{V}_X}(\boldsymbol{x}, y)\big)\Big) :$ $\mathcal{X} \times \mathcal{Y} \to \mathbb{R}^M$ as defined in Section 4.2.

According to Lemma C.7 and Lemma C.6,

$$\mathcal{N}_{[]}\left(\frac{1}{n}; \mathcal{P}_{X|Y}, L^1(\mathcal{X})\right) \leq \mathcal{N}\left(\frac{1}{8edn}; \mathcal{H}_\theta, \|\cdot\|_{\infty, L^\infty(\mathcal{X} \times \mathcal{Y})}\right) \leq \Big(24edn(L+3)(B \vee 1)^{L+2}(W+1)^L\Big)^{S+D+(D+M+K)d_e}.$$

According to Theorem 3.2, we arrive at the conclusion that

$$\mathcal{R}_{\overline{\text{TV}}}(\hat{p}_{X|Y}^{\text{multi}}) \leq 3\sqrt{\frac{1}{n}\left(\log \mathcal{N}_{[]}\left(\frac{1}{n}; \mathcal{P}_{X|Y}^{\text{multi}}, L^1(\mathcal{X})\right) + \log\frac{1}{\delta}\right)}$$

$$\leq 3\sqrt{\frac{1}{n}\left(\big(S + D + (D+M+K)d_e\big)\log\big(24edn(L+3)(B \vee 1)^{L+2}(W+1)^L\big) + \log\frac{1}{\delta}\right)}$$

Omitting constants about $n, K, d_e, L, W, S, B$, and the logarithm term we have $\mathcal{R}_{\overline{\text{TV}}}(\hat{p}_{X|Y}^{\text{multi}})$ $=$ $\tilde{\mathcal{O}}\left(\sqrt{\frac{L\big(S + D + (D+M+K)d_e\big)}{n}}\right)$. $\qquad\qquad \square$

## C.5. Average TV error bound under single-source training

**Theorem C.8** (Average TV error bound for ARMs under single-source training). *Let $\hat{p}_{X|Y}^{\text{single}}$ be the likelihood maximizer defined in Equation (3) given $\mathcal{P}_{X|Y}^{\text{single}}$ with conditional distributions as in Equation (6). Suppose that $\Phi = [0,1]^{d_e}$ and $\Psi = \mathcal{W}(L, W, S, B)$ and assume $\phi_k^* \in \Phi$, $\psi^* \in \Psi$. Then, for any $0 < \delta \leq 1/2$, it holds with probability at least $1 - \delta$ that*

$$\mathcal{R}_{\overline{\text{TV}}}(\hat{p}_{X|Y}^{\text{single}}) = \tilde{\mathcal{O}}\left(\sqrt{\frac{KL\big(S + D + (D+M+1)d_e\big)}{n}}\right).$$

*Proof.* As formulated in Section 2 and with conditional distributions as in Equation (6), we have

$$\mathcal{P}_{X|Y}^{\text{single}} = \left\{ \prod_{k=1}^{K} \big(p_{\theta_k}(\boldsymbol{x}|y)\big)^{\mathbb{I}(y=k)} : p_{\theta_k}(\boldsymbol{x}|y) = p\big(x_1; \boldsymbol{\rho}_{\theta_k}(y)\big) \cdots p\big(x_D; \boldsymbol{\rho}_{\theta_k}(\boldsymbol{x}_{<D}, y)\big) : \boldsymbol{H}_{\theta_k} \in \mathcal{H}_{\theta_k} \right\},$$

where $\boldsymbol{H}_{\theta_k}(\boldsymbol{x}, y) = \begin{bmatrix} \boldsymbol{h}_{\theta_k,1}(\boldsymbol{x}, y) \cdots \boldsymbol{h}_{\theta_k,D}(\boldsymbol{x}, y) \end{bmatrix} : \mathcal{X} \times \mathcal{Y} \to \mathbb{R}^{M \times D}$ with $\boldsymbol{h}_{\theta_k,d}(\boldsymbol{x}, y) = \boldsymbol{f}_{\omega_k}\Big(\boldsymbol{v}_{\boldsymbol{A}_{0k}, \boldsymbol{b}_{0k}}^{\backslash \boldsymbol{0}_{D-d}}\big(\boldsymbol{E}_{\boldsymbol{V}_Y[k,:], \boldsymbol{V}_{Xk}}(\boldsymbol{x}, y)\big)\Big) : \mathcal{X} \times \mathcal{Y} \to \mathbb{R}^M$ as defined in Section 4.2. where

$$\mathcal{U}_{\theta_k} = \left\{ u_{\theta_k}(\boldsymbol{x}|y) = f_{\omega_k} \circ \boldsymbol{e}_{\boldsymbol{V}[k,:]}(\boldsymbol{x}, y) : \omega_k \in \mathcal{W}(L, W, S, B), \boldsymbol{V}[k,:] \in [0,1]^{d_e} \right\}.$$

For all $k \in [K]$, let $\mathcal{B}_{\mathcal{P}_k}$ be an $\frac{1}{n}$-upper bracket of $\mathcal{P}_{X|Y,k} = \left\{ p_{\theta_k}(\boldsymbol{x}|y) = p(x_1; \boldsymbol{\rho}_{\theta_k}(y)) \cdots p(x_D; \boldsymbol{\rho}_{\theta_k}(\boldsymbol{x}_{<D}, y)) : \boldsymbol{H}_{\theta_k} \in \mathcal{H}_{\theta_k} \right\}$ w.r.t. $L^1(\mathcal{X})$ such that $|\mathcal{B}_{\mathcal{P}_k}| = \mathcal{N}_{[]}\left(\frac{1}{n}; \mathcal{P}_{X|Y,k}, L^1(\mathcal{X})\right)$. According to Lemma C.6 and Lemma C.7, we know that

$$|\mathcal{B}_{\mathcal{P}_k}| \leq \mathcal{N}\left( \frac{1}{8edn}; \mathcal{H}_\theta, \|\cdot\|_{\infty, L^\infty(\mathcal{X} \times \mathcal{Y})} \right) \leq \left( 24edn(L+3)(B \vee 1)^{L+2}(W+1)^L \right)^{S+D+(D+M+1)d_e}.$$

For any $p(\boldsymbol{x}|y) = \prod_{k=1}^K \left( p_{\theta_k}(\boldsymbol{x}|y) \right)^{\mathbb{I}(y=k)} \in \mathcal{P}_{X|Y}^{\text{single}}$, there exists $p'_{\theta_1} \in \mathcal{B}_{\mathcal{P}_1}, \ldots, p'_{\theta_K} \in \mathcal{B}_{\mathcal{P}_K}$ such that for all $k \in [K]$, we have: Given any $y \in \mathcal{Y}$, it holds that $\forall \boldsymbol{x} \in \mathcal{X}, p'_{\theta_k}(\boldsymbol{x}|y) \geq p_{\theta_k}(\boldsymbol{x}|y)$, and $\|p'_{\theta_k}(\cdot|y) - p_{\theta_k}(\cdot|y)\|_{L^1(\mathcal{X})} \leq \frac{1}{n}$.

Let $p'(\boldsymbol{x}|y) = \prod_{k=1}^K \left( p'_{\theta_k}(\boldsymbol{x}|y) \right)^{\mathbb{I}(y=k)}$, then we have that given any $y \in \mathcal{Y}$,

$$\forall \boldsymbol{x} \in \mathcal{X}, p'(\boldsymbol{x}|y) = \prod_{k=1}^K \left( p'_{\theta_k}(\boldsymbol{x}|y) \right)^{\mathbb{I}(y=k)} \geq \prod_{k=1}^K \left( p_{\theta_k}(\boldsymbol{x}|y) \right)^{\mathbb{I}(y=k)} = p(\boldsymbol{x}|y),$$

and

$$\|p'(\cdot|y) - p(\cdot|y)\|_{L^1(\mathcal{X})} \leq \sup_{k \in [K]} \|p'_{\theta_k}(\cdot|y) - p_{\theta_k}(\cdot|y)\|_{L^1(\mathcal{X})} \leq \frac{1}{n}.$$

Therefore, $\mathcal{B}_{\mathcal{P}} := \left\{ p'(\boldsymbol{x}|y) = \prod_{k=1}^K \left( p'_{\theta_k}(\boldsymbol{x}|y) \right)^{\mathbb{I}(y=k)} : p'_{\theta_k} \in \mathcal{B}_{\mathcal{P}_k} \right\}$ is an $\frac{1}{n}$-upper bracket of $\mathcal{P}_{X|Y}^{\text{single}}$ w.r.t. $L^1(\mathcal{X})$. Thus we have

$$\mathcal{N}_{[]}\left( \frac{1}{n}; \mathcal{P}_{X|Y}^{\text{single}}, L^1(\mathcal{X}) \right) \leq |\mathcal{B}_{\mathcal{P}}| = \left| \bigcup_{k \in [K]} \mathcal{B}_{\mathcal{P}_k} \right| \leq \prod_{k \in [K]} |\mathcal{B}_{\mathcal{P}_k}|$$

$$= \prod_{k \in [K]} \left( 24edn(L+3)(B \vee 1)^{L+2}(W+1)^L \right)^{S+D+(D+M+1)d_e}$$

$$= \left( 24edn(L+3)(B \vee 1)^{L+2}(W+1)^L \right)^{K(S+D+(D+M+1)d_e)}.$$

According to Theorem 3.2, we arrive at the conclusion that

$$\mathcal{R}_{\overline{\text{TV}}}(\hat{p}_{X|Y}^{\text{single}}) \leq 3 \sqrt{ \frac{1}{n} \left( \log \mathcal{N}_{[]}\left( \frac{1}{n}; \mathcal{P}_{X|Y}^{\text{single}}, L^1(\mathcal{X}) \right) + \log \frac{1}{\delta} \right) }$$

$$\leq 3 \sqrt{ \frac{1}{n} \left( K(S + D + (D+M+1)d_e) \log\left( 24edn(L+3)(B \vee 1)^{L+2}(W+1)^L \right) + \log \frac{1}{\delta} \right) }.$$

Omitting constants about $n, K, d_e, L, W, S, B$, and the logarithm term we have $\mathcal{R}_{\overline{\text{TV}}}(\hat{p}_{X|Y}^{\text{single}}) = \tilde{\mathcal{O}}\left( \sqrt{\frac{KL(S+D+(D+M+1)d_e)}{n}} \right)$.

$\square$

# D. Proofs for Section 4.3

## D.1. Covering number of the energy function class

**Lemma D.1** (Covering number of the energy function class). *Given* $\mathcal{U}_\theta = \{u_\theta(\boldsymbol{x}|y) = f_\omega \circ \boldsymbol{e_V}(\boldsymbol{x}, y)\}$ *with* $u_\theta(\boldsymbol{x}|y) :$ $\mathcal{X} \times \mathcal{Y} \to \mathbb{R}$ *as defined in Section 4.3. Suppose* $\boldsymbol{V}[k,:] \in [0,1]^{d_e}, \omega \in \mathcal{W}(L, W, S, B)$ *with constants* $L, W, S, B > 0$, *and* $\mathcal{X} = [0,1]^D, \mathcal{Y} = [K]$. *Then, given any* $\epsilon > 0$, *we have*

$$\mathcal{N}\left( \epsilon; \mathcal{U}_\theta, \|\cdot\|_{\infty, L^\infty(\mathcal{X} \times \mathcal{Y})} \right) \leq \left( \frac{3(L+1)(B \vee 1)^{L+1}(W+1)^L}{\epsilon} \right)^{S+Kd_e}.$$

*Proof.* As defined, $\mathcal{U}_\theta$ can be written as

$$\mathcal{U}_\theta = \big\{ u_\theta(\boldsymbol{x}|y) = f_\omega \circ \boldsymbol{e}_{\boldsymbol{V}}(\boldsymbol{x}, y) : f_\omega \in \mathcal{F}(L, W, B, S), \boldsymbol{e}_{\boldsymbol{V}} \in \mathcal{E}_{\boldsymbol{V}} \big\} = \mathcal{F}(L, W, B, S) \circ \mathcal{E}_{\boldsymbol{V}},$$

where $\mathcal{E}_{\boldsymbol{V}} = \left\{ \boldsymbol{e}_{\boldsymbol{V}}(\boldsymbol{x}, y) = \begin{bmatrix} \boldsymbol{x} \\ \boldsymbol{V}[y, :] \end{bmatrix} : \boldsymbol{V} \in [0, 1]^{K \times d_e} \right\}$. Denote by $\mathcal{X}_1 := \mathcal{X} \times \mathcal{Y} = [0, 1]^D \times [K]$ and $\mathcal{X}_2 := [0, 1]^{d_e + D}$. Given any $\delta > 0$, we first evaluate the $\delta$-covering number of $\mathcal{E}_{\boldsymbol{V}}$ w.r.t. $\|\cdot\|_{\infty, L^\infty(\mathcal{X}_1)}$.

**Covering number of the embedding layer.** Let $\mathcal{S}_{\text{entry}}(\boldsymbol{V})$ denote the value space of all entries in $\boldsymbol{V}$. We first discretize the value spaces of $\boldsymbol{V}$ into $\delta$-width grids to get a finite embedding function class:

$$\mathcal{E}_{\boldsymbol{V}, \delta\mathbb{Z}} := \{ \boldsymbol{e}_{\boldsymbol{V}} \in \mathcal{E}_{\boldsymbol{V}} : \mathcal{S}_{\text{entry}}(\boldsymbol{V})) = [0, 1] \cap \delta\mathbb{Z} \}.$$

For any $\boldsymbol{e}_{\boldsymbol{V}} \in \mathcal{E}_{\boldsymbol{V}}$, there exists $\boldsymbol{e}_{\boldsymbol{V}'} \in \mathcal{E}_{\boldsymbol{V}, \delta\mathbb{Z}}$ such that $\|\boldsymbol{V} - \boldsymbol{V}'\|_\infty \leq \delta$. Then we have for any $\boldsymbol{x}, y \in \mathcal{X}_1$,

$$\|\boldsymbol{e}_{\boldsymbol{V}}(\boldsymbol{x}, y) - \boldsymbol{e}_{\boldsymbol{V}'}(\boldsymbol{x}, y)\|_\infty = \left\| \begin{bmatrix} \boldsymbol{x} \\ \boldsymbol{V}[y, :] \end{bmatrix} - \begin{bmatrix} \boldsymbol{x} \\ \boldsymbol{V}'[y, :] \end{bmatrix} \right\|_\infty = \left\| \begin{bmatrix} \boldsymbol{0} \\ \boldsymbol{V}[y, :] - \boldsymbol{V}'[y, :] \end{bmatrix} \right\|_\infty \leq \|\boldsymbol{V} - \boldsymbol{V}'\|_\infty \leq \delta. \tag{22}$$

Therefore, $\mathcal{E}_{\boldsymbol{V}, \delta\mathbb{Z}}$ is an $\delta$-cover of $\mathcal{E}_{\boldsymbol{V}}$ w.r.t. $\|\cdot\|_{\infty, L^\infty(\mathcal{X}_1)}$ and thus we have

$$\mathcal{N}\Big(\delta; \mathcal{E}_{\boldsymbol{V}}, \|\cdot\|_{\infty, L^\infty(\mathcal{X}_1)}\Big) \leq |\mathcal{E}_{\boldsymbol{V}, \delta\mathbb{Z}}| = \big|\boldsymbol{V} : \mathcal{S}_{\text{entry}}(\boldsymbol{V})) = [0, 1] \cap \delta\mathbb{Z}\big| \leq \left( \frac{1}{\delta} + 1 \right)^{K d_e}.$$

**Composite energy function.** According to Lemma C.3, given any $\epsilon_{\mathcal{F}}, \epsilon_{\mathcal{E}} > 0$, the covering number of $\mathcal{U}_\theta$ is bounded by

$$\mathcal{N}\Big(\epsilon_{\mathcal{F}} + \kappa_{\mathcal{F}} \epsilon_{\mathcal{E}}; \mathcal{U}_\theta, \|\cdot\|_{\infty, L^\infty(\mathcal{X}_1)}\Big) \leq \mathcal{N}\Big(\epsilon_{\mathcal{F}}; \mathcal{F}(L, W, B, S), \|\cdot\|_{\infty, L^\infty(\mathcal{X}_2)}\Big) \mathcal{N}\Big(\epsilon_{\mathcal{E}}; \mathcal{E}_{\boldsymbol{V}}, \|\cdot\|_{\infty, L^\infty(\mathcal{X}_1)}\Big).$$

According to Lemma C.5, $\kappa_{\mathcal{F}} = B^L W^L$. Further taking $\epsilon_{\mathcal{F}} = L(B \vee 1)^{L-1}(W + 1)^L \delta$ and $\epsilon_{\mathcal{E}} = \delta$, we have

$$\epsilon_{\mathcal{F}} + \kappa_{\mathcal{F}} \epsilon_{\mathcal{E}} = L(B \vee 1)^{L-1}(W + 1)^L \delta + B^L W^L \delta = \Big( L(B \vee 1)^{L-1}(W + 1)^L + B^L W^L \Big) \delta.$$

According to Lemma C.4 and Equation (22), we have

$$\mathcal{N}\Big(\epsilon_{\mathcal{F}}; \mathcal{F}(L, W, B, S), \|\cdot\|_{\infty, L^\infty(\mathcal{X}_2)}\Big) \leq \left( \frac{2B}{\delta} + 1 \right)^S, \text{ and } \mathcal{N}\Big(\epsilon_{\mathcal{E}}; \mathcal{E}_{\boldsymbol{V}}, \|\cdot\|_{\infty, L^\infty(\mathcal{X}_1)}\Big) \leq \left( \frac{1}{\delta} + 1 \right)^{K d_e}.$$

Therefore,

$$\begin{aligned} \mathcal{N}\Big( \big( L(B \vee 1)^{L-1}(W + 1)^L + B^L W^L \big) \delta; \mathcal{U}_\theta, \|\cdot\|_{\infty, L^\infty(\mathcal{X}_1)} \Big) &\leq \left( \frac{2B}{\delta} + 1 \right)^S \left( \frac{1}{\delta} + 1 \right)^{K d_e} \\ &\leq \left( \frac{(2B \vee 1)}{\delta} + 1 \right)^{S + K d_e} \\ &\leq \left( \frac{2(B \vee 1)}{\delta} + 1 \right)^{S + K d_e} \\ &\leq \left( \frac{3(B \vee 1)}{\delta} \right)^{S + K d_e}. \qquad (\tfrac{(B \vee 1)}{\delta} \geq 1) \end{aligned}$$

Taking $\epsilon = \big( L(B \vee 1)^{L-1}(W + 1)^L + B^L W^L \big) \delta$, we have

$$\begin{aligned} \mathcal{N}\Big(\epsilon; \mathcal{U}_\theta, \|\cdot\|_{\infty, L^\infty(\mathcal{X}_1)}\Big) &\leq \left( \frac{3(B \vee 1)\big( L(B \vee 1)^{L-1}(W + 1)^L + B^L W^L \big)}{\epsilon} \right)^{S + K d_e} \\ &\leq \left( \frac{3(B \vee 1)\big( L(B \vee 1)^L(W + 1)^L + (B \vee 1)^L(W + 1)^L \big)}{\epsilon} \right)^{S + K d_e} \\ &= \left( \frac{3(L + 1)(B \vee 1)^{L+1}(W + 1)^L}{\epsilon} \right)^{S + K d_e}, \end{aligned}$$

which completes the proof. $\qquad\square$

## D.2. Bracketing number of the conditional distribution via the energy function

**Lemma D.2** (Bracketing number of the conditional distribution via the energy function). *Given a class of energy-based conditional distributions that* $\mathcal{P}_{X|Y} = \left\{ p_\theta(\boldsymbol{x}|y) = \frac{e^{-u_\theta(\boldsymbol{x}|y)}}{\int_{\mathcal{X}} e^{-u_\theta(\boldsymbol{x}|y)} d\boldsymbol{x}} : u_\theta \in \mathcal{U}_\theta \right\}$, *for any* $0 < \epsilon \leq 1$, *it holds that*

$$\mathcal{N}_{[]}\left(\epsilon; \mathcal{P}_{X|Y}, L^1(\mathcal{X})\right) \leq \mathcal{N}\left(\frac{\epsilon}{4e}; \mathcal{U}_\theta, \|\cdot\|_{\infty, L^\infty(\mathcal{X} \times \mathcal{Y})}\right).$$

*Proof.* Let $\mathcal{C}_{\mathcal{U}}$ be an $\epsilon_{\mathcal{U}}$-cover of $\mathcal{U}_\theta$ w.r.t. $\|\cdot\|_{\infty, L^\infty(\mathcal{X} \times \mathcal{Y})}$ such that $|\mathcal{C}_{\mathcal{U}}| = \mathcal{N}\left(\epsilon_{\mathcal{U}}; \mathcal{U}_\theta, \|\cdot\|_{\infty, L^\infty(\mathcal{X} \times \mathcal{Y})}\right)$. For any $p_\theta(\boldsymbol{x}|y) = \frac{e^{-u_\theta(\boldsymbol{x}|y)}}{\int_{\mathcal{X}} e^{-u_\theta(\boldsymbol{x}|y)} d\boldsymbol{x}} \in \mathcal{P}_{X|Y}$, there exists $u'_\theta \in \mathcal{C}_{\mathcal{U}}$ such that for all $\boldsymbol{x} \in \mathcal{X}$ and $y \in \mathcal{Y}$, $\|u_\theta(\boldsymbol{x}|y) - u'_\theta(\boldsymbol{x}|y)\|_\infty = |u_\theta(\boldsymbol{x}|y) - u'_\theta(\boldsymbol{x}|y)| \leq \epsilon_{\mathcal{U}}$, which equals $u'_\theta(\boldsymbol{x}|y) - \epsilon_{\mathcal{U}} \leq u_\theta(\boldsymbol{x}|y) \leq u'_\theta(\boldsymbol{x}|y) + \epsilon_{\mathcal{U}}$.

Let $p'_\theta(\boldsymbol{x}|y) = \frac{e^{-u'_\theta(\boldsymbol{x}|y) + 2\epsilon_{\mathcal{U}}}}{\int_{\mathcal{X}} e^{-u'_\theta(\boldsymbol{x}|y)} d\boldsymbol{x}}$. Then we immediately obtain that: given any $y \in \mathcal{Y}$,

$$\forall x \in \mathcal{X}, p'_\theta(\boldsymbol{x}|y) = \frac{e^{-u'_\theta(\boldsymbol{x}|y) + \epsilon_{\mathcal{U}}}}{\int_{\mathcal{X}} e^{-u'_\theta(\boldsymbol{x}|y) - \epsilon_{\mathcal{U}}} d\boldsymbol{x}} \geq \frac{e^{-u_\theta(\boldsymbol{x}|y)}}{\int_{\mathcal{X}} e^{-u_\theta(\boldsymbol{x}|y)} d\boldsymbol{x}} = p_\theta(\boldsymbol{x}|y), \tag{23}$$

since for all $\boldsymbol{x} \in \mathcal{X}, y \in \mathcal{Y}$, $e^{-u'_\theta(\boldsymbol{x}|y) + \epsilon_{\mathcal{U}}} \geq e^{-u_\theta(\boldsymbol{x}|y)}$ and $\int_{\mathcal{X}} e^{-u'_\theta(\boldsymbol{x}|y) - \epsilon_{\mathcal{U}}} d\boldsymbol{x} \leq \int_{\mathcal{X}} e^{-u_\theta(\boldsymbol{x}|y)} d\boldsymbol{x}$.

Moreover, we can bound the $L^1(\mathcal{X})$ distance between $p'_\theta(\cdot|y)$ and $p_\theta(\cdot|y)$ as

$$\|p'_\theta(\cdot|y) - p_\theta(\cdot|y)\|_{L^1(\mathcal{X})}$$

$$= \int_{\mathcal{X}} |p'_\theta(\boldsymbol{x}|y) - p_\theta(\boldsymbol{x}|y)| d\boldsymbol{x}$$

$$= \int_{\mathcal{X}} \left| \frac{e^{-u'_\theta(\boldsymbol{x}|y) + 2\epsilon_{\mathcal{U}}}}{\int_{\mathcal{X}} e^{-u'_\theta(\boldsymbol{s}|y)} d\boldsymbol{s}} - \frac{e^{-u_\theta(\boldsymbol{x}|y)}}{\int_{\mathcal{X}} e^{-u_\theta(\boldsymbol{s}|y)} d\boldsymbol{s}} \right| d\boldsymbol{x}$$

$$= \int_{\mathcal{X}} \left| \frac{e^{-u'_\theta(\boldsymbol{x}|y) + 2\epsilon_{\mathcal{U}}} \int_{\mathcal{X}} e^{-u_\theta(\boldsymbol{s}|y)} d\boldsymbol{s} - e^{-u_\theta(\boldsymbol{x}|y)} \int_{\mathcal{X}} e^{-u'_\theta(\boldsymbol{s}|y)} d\boldsymbol{s}}{\int_{\mathcal{X}} e^{-u'_\theta(\boldsymbol{s}|y)} d\boldsymbol{s} \int_{\mathcal{X}} e^{-u_\theta(\boldsymbol{s}|y)} d\boldsymbol{s}} \right| d\boldsymbol{x}$$

$$\leq \int_{\mathcal{X}} \left| \frac{\left(e^{-u'_\theta(\boldsymbol{x}|y) + 2\epsilon_{\mathcal{U}}} - e^{-u_\theta(\boldsymbol{x}|y)}\right) \int_{\mathcal{X}} e^{-u_\theta(\boldsymbol{s}|y)} d\boldsymbol{s} + e^{-u_\theta(\boldsymbol{x}|y)} \left(\int_{\mathcal{X}} e^{-u_\theta(\boldsymbol{s}|y)} d\boldsymbol{s} - e^{-u'_\theta(\boldsymbol{s}|y)} d\boldsymbol{s}\right)}{\int_{\mathcal{X}} e^{-u'_\theta(\boldsymbol{s}|y)} d\boldsymbol{s} \int_{\mathcal{X}} e^{-u_\theta(\boldsymbol{s}|y)} d\boldsymbol{s}} \right| d\boldsymbol{x}$$

$$\leq \int_{\mathcal{X}} \frac{\left| e^{-u'_\theta(\boldsymbol{x}|y) + 2\epsilon_{\mathcal{U}}} - e^{-u_\theta(\boldsymbol{x}|y)} \right| \int_{\mathcal{X}} e^{-u_\theta(\boldsymbol{s}|y)} d\boldsymbol{s} + e^{-u_\theta(\boldsymbol{x}|y)} \int_{\mathcal{X}} \left| e^{-u_\theta(\boldsymbol{s}|y)} - e^{-u'_\theta(\boldsymbol{s}|y)} \right| d\boldsymbol{s}}{\int_{\mathcal{X}} e^{-u'_\theta(\boldsymbol{s}|y)} d\boldsymbol{s} \int_{\mathcal{X}} e^{-u_\theta(\boldsymbol{s}|y)} d\boldsymbol{s}} d\boldsymbol{x}$$

$$\leq \int_{\mathcal{X}} \frac{e^{-u'_\theta(\boldsymbol{x}|y) + 2\epsilon_{\mathcal{U}}} \left| u_\theta(\boldsymbol{x}|y) - u'_\theta(\boldsymbol{x}|y) + 2\epsilon_{\mathcal{U}} \right| \int_{\mathcal{X}} e^{-u_\theta(\boldsymbol{s}|y)} d\boldsymbol{s} + e^{-u_\theta(\boldsymbol{x}|y)} \int_{\mathcal{X}} e^{\left(-u_\theta(\boldsymbol{s}|y)\right) \vee \left(-u'_\theta(\boldsymbol{s}|y)\right)} \left| u_\theta(\boldsymbol{s}|y) - u'_\theta(\boldsymbol{s}|y) \right| d\boldsymbol{s}}{\int_{\mathcal{X}} e^{-u'_\theta(\boldsymbol{s}|y)} d\boldsymbol{s} \int_{\mathcal{X}} e^{-u_\theta(\boldsymbol{s}|y)} d\boldsymbol{s}} d\boldsymbol{x}$$

$$\left( |e^a - e^b| = \left| \int_b^a e^x dx \right| \leq \left| \int_b^a e^{a \vee b} dx \right| = e^{a \vee b} |a - b| \right)$$

$$\leq \int_{\mathcal{X}} \frac{e^{-u'_\theta(\boldsymbol{x}|y) + 2\epsilon_{\mathcal{U}}} 3\epsilon_{\mathcal{U}} \int_{\mathcal{X}} e^{-u_\theta(\boldsymbol{s}|y)} d\boldsymbol{s} + e^{-u_\theta(\boldsymbol{x}|y)} \int_{\mathcal{X}} e^{-u'_\theta(\boldsymbol{s}|y) + \epsilon_{\mathcal{U}}} \epsilon_{\mathcal{U}} d\boldsymbol{s}}{\int_{\mathcal{X}} e^{-u'_\theta(\boldsymbol{s}|y)} d\boldsymbol{s} \int_{\mathcal{X}} e^{-u_\theta(\boldsymbol{s}|y)} d\boldsymbol{s}} d\boldsymbol{x}$$

$$\left( \forall x \in \mathcal{X}, y \in \mathcal{Y}, \left| u_\theta(\boldsymbol{x}|y) - u'_\theta(\boldsymbol{x}|y) \right| \leq \epsilon_{\mathcal{U}} \right)$$

$$\leq \int_{\mathcal{X}} \frac{3\epsilon_{\mathcal{U}} e^{-u_\theta(\boldsymbol{x}|y) + 3\epsilon_{\mathcal{U}}} \int_{\mathcal{X}} e^{-u'_\theta(\boldsymbol{s}|y) + \epsilon_{\mathcal{U}}} d\boldsymbol{s} + \epsilon_{\mathcal{U}} e^{-u_\theta(\boldsymbol{x}|y)} \int_{\mathcal{X}} e^{-u'_\theta(\boldsymbol{s}|y) + \epsilon_{\mathcal{U}}} d\boldsymbol{s}}{\int_{\mathcal{X}} e^{-u'_\theta(\boldsymbol{s}|y)} d\boldsymbol{s} \int_{\mathcal{X}} e^{-u_\theta(\boldsymbol{s}|y)} d\boldsymbol{s}} d\boldsymbol{x}$$

$$\left( \forall x \in \mathcal{X}, y \in \mathcal{Y}, \left| u_\theta(\boldsymbol{x}|y) - u'_\theta(\boldsymbol{x}|y) \right| \leq \epsilon_{\mathcal{U}} \right)$$

$$= \int_{\mathcal{X}} \frac{3\epsilon_{\mathcal{U}} e^{4\epsilon_{\mathcal{U}}} e^{-u_\theta(\boldsymbol{x}|y)} \int_{\mathcal{X}} e^{-u'_\theta(\boldsymbol{s}|y)} d\boldsymbol{s} + \epsilon_{\mathcal{U}} e^{\epsilon_{\mathcal{U}}} e^{-u_\theta(\boldsymbol{x}|y)} \int_{\mathcal{X}} e^{-u'_\theta(\boldsymbol{s}|y)} d\boldsymbol{s}}{\int_{\mathcal{X}} e^{-u'_\theta(\boldsymbol{s}|y)} d\boldsymbol{s} \int_{\mathcal{X}} e^{-u_\theta(\boldsymbol{s}|y)} d\boldsymbol{s}} d\boldsymbol{x}$$

$$= \int_{\mathcal{X}} \frac{\left(3\epsilon_{\mathcal{U}} e^{4\epsilon_{\mathcal{U}}} + \epsilon_{\mathcal{U}} e^{\epsilon_{\mathcal{U}}}\right) e^{-u_\theta(\boldsymbol{x}|y)}}{\int_{\mathcal{X}} e^{-u_\theta(\boldsymbol{s}|y)} d\boldsymbol{s}} d\boldsymbol{x} = 3\epsilon_{\mathcal{U}} e^{4\epsilon_{\mathcal{U}}} + \epsilon_{\mathcal{U}} e^{\epsilon_{\mathcal{U}}} \leq 4\epsilon_{\mathcal{U}} e^{(4\epsilon_{\mathcal{U}}) \vee 1}. \tag{24}$$

Therefore, $\mathcal{B}_{\mathcal{P}} := \left\{ p'_\theta(\boldsymbol{x}|y) = \frac{e^{-u'_\theta(\boldsymbol{x}|y)+\epsilon_{\mathcal{U}}}}{\int_{\mathcal{X}} e^{-u'_\theta(\boldsymbol{x}|y)-\epsilon_{\mathcal{U}}} d\boldsymbol{x}} : -u'_\theta(\boldsymbol{x}|y) \in \mathcal{C}_{\mathcal{U}} \right\}$ is an $4\epsilon_{\mathcal{U}} e^{4\epsilon_{\mathcal{U}}}$-upper bracket of $\mathcal{P}_{X|Y}$ w.r.t. $L^1(\mathcal{X})$. Thus we have

$$\mathcal{N}_{[]}\left(4\epsilon_{\mathcal{U}} e^{(4\epsilon_{\mathcal{U}})\vee 1}; \mathcal{P}_{X|Y}, L^1(\mathcal{X})\right) \leq |\mathcal{B}_{\mathcal{P}}| = |\mathcal{C}_{\mathcal{U}}| = \mathcal{N}\left(\epsilon_{\mathcal{U}}; \mathcal{U}_\theta, \|\cdot\|_{\infty, L^\infty(\mathcal{X}\times\mathcal{Y})}\right).$$

Let $4\epsilon_{\mathcal{U}} e^{(4\epsilon_{\mathcal{U}})\vee 1} = \epsilon$, we have $4\epsilon_{\mathcal{U}} = \frac{\epsilon}{e^{(4\epsilon_{\mathcal{U}})\vee 1}} \leq 1$ and thus $4\epsilon_{\mathcal{U}} e^{(4\epsilon_{\mathcal{U}})\vee 1} = 4e\epsilon_{\mathcal{U}}$, so that we get $\epsilon_{\mathcal{U}} = \frac{\epsilon}{4e}$. Therefore, we have for any $0 < \epsilon \leq 1$,

$$\mathcal{N}_{[]}\left(\epsilon; \mathcal{P}_{X|Y}, L^1(\mathcal{X})\right) \leq \mathcal{N}\left(\frac{\epsilon}{4e}; \mathcal{U}_\theta, \|\cdot\|_{\infty, L^\infty(\mathcal{X}\times\mathcal{Y})}\right)$$

$\square$

## D.3. Proof of Theorem 4.4

Based on the relation between the bracketing number of conditional distribution space $\mathcal{P}_{X|Y}$ and the covering number of energy function space $\mathcal{U}_\theta$ derived in previous lemmas, we obtain the final result.

*Proof of Theorem 4.4.* With conditional distributions as defined in Equation (7), we have

$$\mathcal{P}_{X|Y}^{\text{multi}} = \left\{ p_\theta(\boldsymbol{x}|y) = \frac{e^{-u_\theta(\boldsymbol{x}|y)}}{\int_{\mathcal{X}} e^{-u_\theta(\boldsymbol{x}|y)} d\boldsymbol{x}} : u_\theta \in \mathcal{U}_\theta \right\},$$

where

$$\mathcal{U}_\theta = \left\{ u_\theta(\boldsymbol{x}|y) = f_\omega \circ \boldsymbol{e_V}(\boldsymbol{x}, y) : \omega \in \mathcal{W}(L, W, S, B), \boldsymbol{V}[k, :] \in [0, 1]^{d_e} \right\}.$$

Let $\epsilon_{\mathcal{U}}$ be an constant that $\epsilon_{\mathcal{U}} > 0$, according to Lemma D.1, we have $\mathcal{N}\left(\epsilon_{\mathcal{U}}; \mathcal{U}_\theta, \|\cdot\|_{\infty, L^\infty(\mathcal{X}\times\mathcal{Y})}\right) \leq \left(\frac{3(L+1)(B\vee 1)^{L+1}(W+1)^L}{\epsilon_{\mathcal{U}}}\right)^{S+Kd_e}$. Then with Lemma D.2, we further obtain that

$$\mathcal{N}_{[]}\left(\frac{1}{n}; \mathcal{P}_{X|Y}^{\text{multi}}, L^1(\mathcal{X})\right) \leq \mathcal{N}\left(\frac{1}{4en}; \mathcal{U}_\theta, \|\cdot\|_{\infty, L^\infty(\mathcal{X}\times\mathcal{Y})}\right) \leq \left(12e(L+1)(B\vee 1)^{L+1}(W+1)^L\right)^{S+Kd_e}.$$

According to Theorem 3.2, we arrive at the conclusion that

$$\mathcal{R}_{\overline{\text{TV}}}(\hat{p}_{X|Y}^{\text{multi}}) \leq 3\sqrt{\frac{1}{n}\left(\log \mathcal{N}_{[]}\left(\frac{1}{n}; \mathcal{P}_{X|Y}^{\text{multi}}, L^1(\mathcal{X})\right) + \log\frac{1}{\delta}\right)}$$

$$\leq 3\sqrt{\frac{1}{n}\left((S + Kd_e)\log\left(12en(L+1)(B\vee 1)^{L+1}(W+1)^L\right) + \log\frac{1}{\delta}\right)}$$

$$= 3\sqrt{\frac{1}{n}\left(L(S + Kd_e)\log\left(12en(L+1)^{\frac{1}{L}}(B\vee 1)^{1+\frac{1}{L}}(W+1)\right) + \log\frac{1}{\delta}\right)}$$

Omitting constants about $n, K, d_e, L, W, S, B$, and the logarithm term we have $\mathcal{R}_{\overline{\text{TV}}}(\hat{p}_{X|Y}^{\text{multi}}) = \tilde{\mathcal{O}}\left(\sqrt{\frac{L(S+Kd_e)}{n}}\right)$. $\square$

## D.4. Average TV error bound under single-source training

**Theorem D.3** (average TV error bound for EBMs under single-source training). *Let $\hat{p}_{X|Y}^{\text{single}}$ be the likelihood maximizer defined in Equation (3) given $\mathcal{P}_{X|Y}^{\text{single}}$ with conditional distributions as in Equation (7), suppose that $\Phi = [0, 1]^{d_e}$ and $\Psi = \mathcal{W}(L, W, S, B)$ and assume $\phi_k^* \in \Phi, \psi^* \in \Psi$. Then, for any $0 < \delta \leq 1/2$, it holds with probability at least $1 - \delta$ that*

$$\mathcal{R}_{\overline{\text{TV}}}(\hat{p}_{X|Y}^{\text{single}}) = \tilde{\mathcal{O}}\left(\sqrt{\frac{LK(S+d_e)}{n}}\right).$$

*Proof.* As formulated in Section 2 and with conditional distributions as in Equation (7), we have

$$\mathcal{P}_{X|Y}^{\text{single}} = \left\{ \prod_{k=1}^{K} \big(p_{\theta_k}(\boldsymbol{x}|y)\big)^{\mathbb{I}(y=k)} : p_{\theta_k}(\boldsymbol{x}|y) = \frac{e^{-u_{\theta_k}(\boldsymbol{x}|y)}}{\int_{\mathcal{X}} e^{-u_{\theta_k}(\boldsymbol{x}|y)} d\boldsymbol{x}} : u_{\theta_k} \in \mathcal{U}_{\theta_k} \right\},$$

where

$$\mathcal{U}_{\theta_k} = \left\{ u_{\theta_k}(\boldsymbol{x}|y) = f_{\omega_k} \circ \boldsymbol{e}_{\boldsymbol{V}[k,:]}(\boldsymbol{x}, y) : \omega_k \in \mathcal{W}(L, W, S, B), \boldsymbol{V}[k,:] \in [0,1]^{d_{\text{e}}} \right\}.$$

For all $k \in [K]$, let $\mathcal{B}_{\mathcal{P}_k}$ be an $\frac{1}{n}$-upper bracket of $\mathcal{P}_{X|Y,k} = \left\{ p_{\theta_k}(\boldsymbol{x}|y) = \frac{e^{-u_{\theta_k}(\boldsymbol{x}|y)}}{\int_{\mathcal{X}} e^{-u_{\theta_k}(\boldsymbol{x}|y)} d\boldsymbol{x}} : u_{\theta_k} \in \mathcal{U}_{\theta_k} \right\}$ w.r.t. $L^1(\mathcal{X})$ such that $|\mathcal{B}_{\mathcal{P}_k}| = \mathcal{N}_{[]}\left(\frac{1}{n}; \mathcal{P}_{X|Y,k}, L^1(\mathcal{X})\right)$. According to Lemma D.1 and Lemma D.2, we know that

$$|\mathcal{B}_{\mathcal{P}_k}| \leq \mathcal{N}\left(\frac{1}{4en}; \mathcal{U}_{\theta_k}, \|\cdot\|_{\infty, L^\infty(\mathcal{X} \times \mathcal{Y})}\right) \leq \left(12en(L+1)(B \vee 1)^{L+1}(W+1)^L\right)^{S+d_{\text{e}}}.$$

For any $p(\boldsymbol{x}|y) = \prod_{k=1}^{K} \big(p_{\theta_k}(\boldsymbol{x}|y)\big)^{\mathbb{I}(y=k)} \in \mathcal{P}_{X|Y}^{\text{single}}$, there exists $p'_{\theta_1} \in \mathcal{B}_{\mathcal{P}_1}, \ldots, p'_{\theta_K} \in \mathcal{B}_{\mathcal{P}_K}$ such that for all $k \in [K]$, we have: Given any $y \in \mathcal{Y}$, it holds that $\forall \boldsymbol{x} \in \mathcal{X}, p'_{\theta_k}(\boldsymbol{x}|y) \geq p_{\theta_k}(\boldsymbol{x}|y)$, and $\|p'_{\theta_k}(\cdot|y) - p_{\theta_k}(\cdot|y)\|_{L^1(\mathcal{X})} \leq \frac{1}{n}$.

Let $p'(\boldsymbol{x}|y) = \prod_{k=1}^{K} \big(p'_{\theta_k}(\boldsymbol{x}|y)\big)^{\mathbb{I}(y=k)}$, then we have that given any $y \in \mathcal{Y}$,

$$\forall \boldsymbol{x} \in \mathcal{X}, p'(\boldsymbol{x}|y) = \prod_{k=1}^{K} \big(p'_{\theta_k}(\boldsymbol{x}|y)\big)^{\mathbb{I}(y=k)} \geq \prod_{k=1}^{K} \big(p_{\theta_k}(\boldsymbol{x}|y)\big)^{\mathbb{I}(y=k)} = p(\boldsymbol{x}|y),$$

and

$$\|p'(\cdot|y) - p(\cdot|y)\|_{L^1(\mathcal{X})} \leq \sup_{k \in [K]} \|p'_{\theta_k}(\cdot|y) - p_{\theta_k}(\cdot|y)\|_{L^1(\mathcal{X})} \leq \frac{1}{n}.$$

Therefore, $\mathcal{B}_{\mathcal{P}} := \left\{ p'(\boldsymbol{x}|y) = \prod_{k=1}^{K} \big(p'_{\theta_k}(\boldsymbol{x}|y)\big)^{\mathbb{I}(y=k)} : p'_{\theta_k} \in \mathcal{B}_{\mathcal{P}_k} \right\}$ is an $\frac{1}{n}$-upper bracket of $\mathcal{P}_{X|Y}^{\text{single}}$ w.r.t. $L^1(\mathcal{X})$. Thus we have

$$\mathcal{N}_{[]}\left(\frac{1}{n}; \mathcal{P}_{X|Y}^{\text{single}}, L^1(\mathcal{X})\right) \leq |\mathcal{B}_{\mathcal{P}}| = \left| \bigcup_{k \in [K]} \mathcal{B}_{\mathcal{P}_k} \right| \leq \prod_{k \in [K]} |\mathcal{B}_{\mathcal{P}_k}|$$

$$= \prod_{k \in [K]} \left(12en(L+1)(B \vee 1)^{L+1}(W+1)^L\right)^{S+d_{\text{e}}}$$

$$= \left(12en(L+1)(B \vee 1)^{L+1}(W+1)^L\right)^{K(S+d_{\text{e}})}.$$

According to Theorem 3.2, we arrive at the conclusion that

$$\mathcal{R}_{\overline{\text{TV}}}(\hat{p}_{X|Y}^{\text{single}}) \leq 3\sqrt{\frac{1}{n}\left(\log \mathcal{N}_{[]}\left(\frac{1}{n}; \mathcal{P}_{X|Y}^{\text{single}}, L^1(\mathcal{X})\right) + \log \frac{1}{\delta}\right)}$$

$$\leq 3\sqrt{\frac{1}{n}\left(LK(S+d_{\text{e}})\log\left((12enL+1)^{\frac{1}{L}}(B \vee 1)^{1+\frac{1}{L}}(W+1)\right) + \log \frac{1}{\delta}\right)}$$

Omitting constants about $n, K, d_{\text{e}}, L, W, S, B$, and the logarithm term we have $\mathcal{R}_{\overline{\text{TV}}}(\hat{p}_{X|Y}^{\text{single}}) = \tilde{\mathcal{O}}\left(\sqrt{\frac{LK(S+d_{\text{e}})}{n}}\right)$.

$\square$

*Table 2.* Hyparameters of our experiments. '1c' denotes training from single-source, and others denote training from multi-source which contains 3,5, and 10 classes.

| Setup | Iterations (kimg) | Learning rate | Decay (kimg) |
|-------|-------------------|---------------|--------------|
| 1c    | 184549            | 0.005         | 2500         |
| 3c    | 268435            | 0.006         | 4000         |
| 10c   | 1610612           | 0.012         | 6000         |

*Table 3.* Standard deviations of FID scores over five times of sampling.

| $N$ | Sim | $K$ | Std Dev (Single) | Std Dev (Multi) |
|-----|-----|-----|------------------|-----------------|
| 500 | 1 | 3 | 0.0086 | 0.0057 |
|     |   | 10 | 0.0018 | 0.0336 |
|     | 2 | 3 | 0.0160 | 0.0158 |
|     |   | 10 | 0.0056 | 0.0035 |
| 1000 | 1 | 3 | 0.0034 | 0.0064 |
|     |   | 10 | 0.0028 | 0.0250 |
|     | 2 | 3 | 0.0047 | 0.0051 |
|     |   | 10 | 0.0013 | 0.0084 |

# E. Supplementary for experiments

## E.1. Additional detail of real-world experiments

**Implementation.** Following EDM2, we use the Latent Diffusion Model (LDM) (Rombach et al., 2022) to down-sample each image $x \in \mathbb{R}^{3 \times 256 \times 256}$ to a corresponding latent $z \in \mathbb{R}^{4 \times 32 \times 32}$ for training a diffusion models.

All experiments are trained and sampled on $8 \times$ NVIDIA A800 80GB, $8 \times$ NVIDIA GeForce RTX 4090, and $8 \times$ NVIDIA GeForce RTX 3090 on the Linux Ubuntu-22.04 platform.

For a fair comparison, we set different hyperparameters for experiments with different numbers of sources as shown in Table 2, but these parameters are the same within each similarity.

Based on these trained models, we perform multiple samplings using five different random seeds to estimate the randomness in calculating the FID scores. The standard deviations of FID scores over multiple samplings are reported in Table 3, corresponding to Table 1 in the main paper.

**The selection of sample sizes and the number of classes.** In the real-world experiments, we set the number of classes $K$ in 3 and 10, and the sample size per class $N$ in 500 and 1000. We would like to clarify that this selection is influenced by several inherent characteristics of the ILSVRC2012 dataset: (1) Sample sizes: The maximum number of images per class in ILSVRC2012 is 1300, so we selected sample sizes of 1000 and 500 images per class, which are common choices. (2) Number of sources: Given that distribution similarity levels were manually defined, it was difficult to establish a large number of structured subdivisions. To be specific, to ensure reasonable similarity levels for the controlled experiment, we designed a two-level tree structure for the dataset, as shown in Figure 2. Overall, we divided the whole ILSVRC2012 into 10 high-level categories (mammal, amphibian, bird, fish, reptile, vehicle, furniture, musical instrument, geological formation, and utensil). Each category was further subdivided into 10 subsets (e.g., for mammals, we have Italian greyhound, Border terrier, standard schnauzer, etc.). Defining such semantically meaningful and mutually exclusive divisions is not trivial. As a result, the number of classes within each similarity level in our experiments is limited to 10.

While our experiments are not on large-scale datasets, there are existing studies that provide valuable empirical observations for large-scale multi-source training, including: cross-lingual model transfer for similar languages (Pires et al., 2019), pretraining with additionla high-quality images to improve overall aesthetics in image generation (Chen et al., 2024), and knowledge augmentation on subsets of data to enhance model performance on other subsets (Allen-Zhu & Li, 2024a). They have offered relevant findings that inform our work.

**Connection between FID and the theoretical guarantees.** Our theory provides guarantees for the average TV distance

*Table 4.* TV errors with the number of sources $K$ in simulations on ARMs.

| $K \uparrow$ | 1 | 3 | 5 | 7 | 10 |
|---|---|---|---|---|---|
| Single-source | 0.0763 | 0.1212 | 0.1519 | 0.1787 | 0.2127 |
| Multi-source | 0.0763 | 0.1145 | 0.1318 | 0.1364 | 0.1369 |

*Table 5.* TV errors with the sample size $n$ in simulations on ARMs.

| $n \downarrow$ | 1000 | 3000 | 5000 | 10000 | 30000 |
|---|---|---|---|---|---|
| Single-source | 0.5680 | 0.3516 | 0.2882 | 0.2036 | 0.1212 |
| Multi-source | 0.5491 | 0.3467 | 0.2747 | 0.1922 | 0.1145 |

*Table 6.* TV errors with the sequence length $D$ in simulations on ARMs.

| $D \uparrow$ | 10 | 12 | 14 | 16 | 18 |
|---|---|---|---|---|---|
| Single-source | 0.2036 | 0.3785 | 0.5932 | 0.7242 | 0.7505 |
| Multi-source | 0.1922 | 0.3530 | 0.5068 | 0.5747 | 0.6289 |

(Equation (4)), which quantifies distribution estimation quality but is incomputable without access to the true conditional distributions. Therefore, in real-world experiments, we use FID as a practical alternative. FID measures the similarity between generated and real data distributions by comparing their feature representations in a pretrained neural network. It is widely used to evaluate image generation quality and serves as the best available metric for our setting.

**Connection with the theoretical analysis of EBMs.** Additionally, we would like to discuss the connection between our real-world diffusion model experiments and the theoretical analysis of EBMs. As mentioned in Section 1, EBMs are a general and flexible class of generative models closely connected to diffusion models. To be specific, first, the training and sampling methods in (Song & Ermon, 2019; Song et al., 2021) are directly inspired by EBMs. The distinction is that EBMs parameterize the energy function, while diffusion models parameterize its gradient (the score function). Second, Salimans & Ho (2021) shows that under a specific energy function formulation (Equation (5) in their paper), EBMs are equivalent to constrained diffusion models. Their experimental results (Table 1, Rows A and B) indicate that the constraint has a minor impact on generative performance. Thus, our diffusion model experiments provide insight into EBMs' behavior in real-world settings to some extent.

### E.2. Supplementary Simulations on ARMs

We conduct additional simulations on autoregressive models (ARMs) to examine how empirical total variation (TV) errors align with the theoretical predictions. The empirical results are summarized in Tables 4, 5, and 6.

Each ground truth source distribution is defined as a discrete categorical distribution supported on the set $[M]^D$, where $M$ is the vocabulary size and $D$ is the sequence length. The variable $Y$ is drawn uniformly from $\{1, 2, \ldots, K\}$. A multi-source dataset of size $n$ is sampled from the joint distribution of $(X, Y)$ by first drawing $n$ samples of $Y$, followed by sampling $X \mid Y$ conditionally.

The network architecture exactly follows the setup in Section 4.2. It consists of two embedding matrices to encode $Y$ and the first $D - 1$ dimensions of $X$ into $d_e$-dimensional vectors. These embeddings are processed by a single encoding layer, followed by a multi-layer perceptron (MLP) with width $W$, depth $L$, and a softmax output. The conditional distribution parameters are computed autoregressively using a masked input vector.

For multi-source training, a single model is trained on the full dataset. In contrast, single-source training involves training $K$ separate models, each using data from its corresponding source. In all experiments, we fix $M = 2$ and use network configurations with $d_e = W = 64$, $L = 5$, and batch size $B = 1$. We vary the number of sources $K \in \{1, 3, 5, 7, 10\}$, the total number of samples $n \in \{1000, 3000, 5000, 10000, 30000\}$, and the sequence length $D \in \{10, 12, 14, 16, 18\}$ to assess the alignment between empirical total variation (TV) error and the theoretical bounds. For each configuration, the batch size and learning rate are selected from $\{100, 300, 500\}$ and $\{10^{-5}, 10^{-4}, 10^{-3}\}$, respectively, for maximum likelihood.

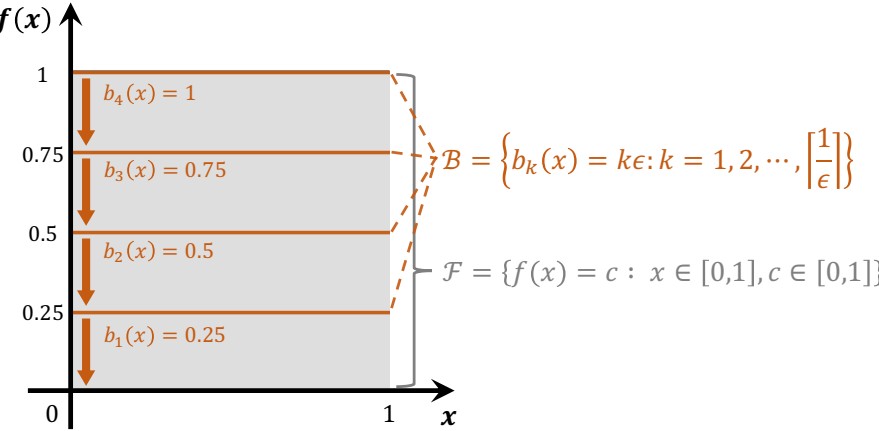

*Figure 4.* Illustration of $\epsilon$-upper brackets for the constant function class with $\epsilon = 0.25$. Each bracket function (horizontal line) lies above the function it covers, with a difference at most $\epsilon$.

## F. Additional discussions on the notion of $\beta_{sim}$

The notion of $\beta_{sim}$ in Section 4 is defined *by induction* based on our three specific model instantiations. It directly measures the *model parameter sharing across different sources*, and thus reflects the *source distribution similarity* under our theoretical formulation in Section 2.2. As such, its exact formulation varies depending on the model instantiation.

Specifically, in the Gaussian model (Section 4.1), $\beta_{sim} = \frac{d-d_1}{d}$ measures the proportion of shared mean vector dimensions, which seems to correspond to the property of the ground truth distribution. While for ARMs or EBMs (Section 4.2 and Section 4.3), $\beta_{sim} = \frac{S}{S+d_e}$ is based on shared model parameters, which do not explicitly represent the data distribution itself. Despite this difference, in both cases, $\beta_{sim}$ fundamentally represents the extent of parameter sharing across sources. The distinction arises from the modeling paradigm: the Gaussian case assumes a parametric form for distributions, where model parameters (e.g., mean vectors) explicitly encode data properties, whereas EBMs use neural networks as a function approximator to fit probability densities without an explicit distributional form, making no explicit connection between parameters and data.

As a result, $\beta_{sim}$ cannot be directly computed from general datasets without model-specific assumptions. We remark that rigorously quantifying dataset similarity in practice is still a direction under exploration. Possible approaches might include: (1) From a practical perspective, a small proxy model can be used to estimate source distributions' interaction (Xie et al., 2023). (2) From a theoretical perspective, several existing notions in multi-task learning and meta-learning could be adapted for this purpose, such as transformation equivalence (Ben-David & Borbely, 2008), parameter distance (Balcan et al., 2019), and distribution divergence (Jose & Simeone, 2021).

## G. Intuitive illustration of the upper bracketing number

The $\epsilon$-upper bracketing number (Definition 3.1) is a way to quantify the complexity of an infinite set of functions. The key idea is to construct a collection of "brackets" that enclose every function in the set within a small margin.

To illustrate this, consider a simple example. Suppose we have the function set $\mathcal{F} = \{f(x) = c : x \in [0,1], c \in [0,1]\}$, which consists of all constant functions taking values in the interval $[0,1]$. We can construct an $\epsilon$-upper bracket for $\mathcal{F}$ by defining the finite set $\mathcal{B} = \{b(x) = k\epsilon : k = 1, \ldots, \lceil 1/\epsilon \rceil\}$. Then, for any function $f \in \mathcal{F}$, there exists a bracket function $b \in \mathcal{B}$ such that: (1) For all $x \in [0,1]$, the bracket function is always an upper bound: $b(x) \geq f(x)$. (2) The total "gap" between $b$ and $f$, measured by the integral $\int_0^1 b(x) - f(x)dx$, is at most $\epsilon$. Intuitively, this means we can cover the entire function class using a small number of simple approximating functions that "overestimate" each function just slightly. This concept is visualized in Figure 4, where we show such brackets for $\epsilon = 0.25$.

In our paper, we extend this idea to conditional probability spaces. There, each condition defines its own function set, and we construct corresponding upper brackets that ensure every conditional distribution is approximated with a small error uniformly across conditions.

