# OpenReview forum: "A Theory for Conditional Generative Modeling on Multiple Data Sources"
_ICML.cc/2025/Conference — ICML 2025 poster_

### Official Review · Reviewer_8TSP · 2025-03-04

**Overall Recommendation:** 4

**Summary:**

This work analyzes the effect of training with multiple data sources on conditional generative models. The authors establish a bound on the total variation distance between true and model distributions in terms of the bracketing number.

## update after rebuttal

The other reviews and the rebuttal have increased my confidence in my initial estimate of the quality of this paper, so I have increased my score by 1 accordingly.

**Claims And Evidence:**

The experimental data aligns well with the theoretical bounds for the Gaussian case. For ARMs and EBMs, the connection to the experiment with diffusion models should be more clearly delineated.

**Essential References Not Discussed:**

\-

**Experimental Designs Or Analyses:**

There is no explicit experimental design. The standard deviations on the results seem reasonably small based on the smoothness of the curves but it would be great to report an error estimate for the FID scores.

**Methods And Evaluation Criteria:**

Yes, the combination of a simple setup with Gaussians and an empirical evaluation based on diffusion models for image data fits well.

**Other Comments Or Suggestions:**

- I believe in line 61 there should be parentheses around $K - 1$.

**Other Strengths And Weaknesses:**

I don't see any specific weaknesses but I am not knowledgeable enough about this field to judge the importance of these results to the wider community. My impression is that this paper examines a practically relevant setting theoretically in a sound way and and validates the main results empirically.

**Questions For Authors:**

1. How exactly does the diffusion model experiment relate to your bounds for ARMs and EBMs?
2. What are the obstacles for applying these bounds to practical generative models?
3. Is there a direct connection between the FID and your theoretical guarantees?

**Relation To Broader Scientific Literature:**

\-

**Theoretical Claims:**

I skimmed the appendix but did not check the correctness of the proofs in detail.

---

> ### Author Rebuttal · Authors · 2025-03-31
>
> # Experimental suggestion: Error estimate for FID scores
> We thank the reviewer for the valuable suggestion regarding error estimation.
>
> We would like to clarify that the real-world experiments in Section 5.2 were run only once due to the long training time. Based on these trained models, we additionally performed multiple samplings using five different random seeds to estimate the randomness in calculating FID scores following the reviewer's suggestion. The mean values and standard deviations of FID scores over multiple samplings are reported in the table below (corresponding to Table 1 of our submission).
>
> | $N$  | $\mathrm{Sim}$ | $K$ | Avg. FID (Single) | Std Dev (Single) | Avg. FID (Multi) | Std Dev (Multi) |
> | --- | --- | --- | --- | --- | ---| ---|
> | 500  | 1  | 3   | 30.03  | 0.0086  | 29.94 | 0.0057 |
> | |   | 10  | 30.18  | 0.0018 | 29.28 | 0.0336 |
> | | 2 | 3 | 32.69  | 0.0160 | 30.69 | 0.0158 |
> | | | 10  | 30.54 | 0.0056 | 28.75 | 0.0035 |
> | 1000 | 1 | 3   | 28.01| 0.0034  | 26.41| 0.0064 |
> | | | 10  | 27.49 | 0.0028 | 25.84 | 0.0250 |
> | | 2  | 3   | 30.58 | 0.0047 | 29.35 | 0.0051 |
> | | | 10  | 29.01 | 0.0013 | 27.81 | 0.0084 |
>
> We will include these results in the revised version.
>
> # Typo: Missing parentheses
> Thank you for pointing this out. We will correct this in the revised version.
>
> # Q1: Experiments on diffusion models and the theory for ARMs & EBMs
>
> We thank the reviewer for the insightful question.
>
> EBMs, as mentioned in lines 51-55 in our submission, are a general and flexible class of generative models closely connected to diffusion models. To be specific, first, the training and sampling methods in [1,2] are directly inspired by EBMs. The distinction is that EBMs parameterize the energy function, while diffusion models parameterize the score function, which is the energy function's gradient. Second, [3] shows that under a specific energy function formulation (Equation (5) in their paper), EBMs are equivalent to constrained diffusion models. Their experimental results (Table 1, Rows A and B in their paper) indicate that the constraint has a minor impact on generative performance. Thus, our diffusion model experiments provide insight into EBMs' behavior in real-world settings to some extent.
>
> Additionally, we have added supplementary simulations for ARMs according to the formulation in Section 4.2. The empirical TV errors exhibit similar trends as theoretical bounds in Theorem 4.3 regarding several key factors—the number of sources $K$, sample size $n$, and data length $D$. Due to space constraints in the rebuttal, please refer to our response to Reviewer 7LUU (Q1) for detailed experimental settings and results.
>
> We will add the above discussions for EBMs, and simulation results along with implementation details for ARMs in the revised version of our paper.
>
> # Q2: Obstacles for practical application
>
> As discussed in Section 7 in the submission (lines 422-432, right column), our theoretical formulation of multi-source training through conditional generative modeling abstracts real-world scenarios to some extent. In practice, conditions may not be explicitly given (e.g., in language models) or may involve multiple source labels (e.g., large-scale image generation).
>
> Our analysis provides a first step toward understanding multi-source training under a simplified yet reasonable setting. Extending it to more complex, fine-grained multi-source interaction scenarios is a valuable direction for future work. Possible approaches might include: characterizing distribution similarity without explicit conditions [1,2] or investigating the multiple-label case by compositional generative modeling [3, 4].
>
> [1] Ben-David, S., & Borbely, R. S. (2008). A notion of task relatedness yielding provable multiple-task learning guarantees.
>
> [2] Jose, S. T., & Simeone, O. (2021). An information-theoretic analysis of the impact of task similarity on meta-learning.
>
> [3] Okawa, M., Lubana, E. S., Dick, R., & Tanaka, H. (2023). Compositional abilities emerge multiplicatively: Exploring diffusion models on a synthetic task.
>
> [4] Lake, B. M., & Baroni, M. (2023). Human-like systematic generalization through a meta-learning neural network.
>
> # Q3: Connection between FID and theoretical guarantees
>
> Our theory provides guarantees for the average TV distance (lines 142-155, left column), which quantifies distribution estimation quality but is incomputable without access to the true conditional distributions.
>
> Therefore, in real-world experiments (Section 5.2), we use FID as a practical alternative. FID measures the similarity between generated and real data distributions by comparing their feature representations in a pretrained neural network. It is widely used to evaluate image generation quality and serves as the best available metric for our setting.
>
> We will add the above discussion to clarify the choice of FID in the revised version.

---

### Official Review · Reviewer_MrF2 · 2025-03-12

**Overall Recommendation:** 4

**Summary:**

This paper takes the first step toward a rigorous analysis of multi-source training in conditional generative modeling, where each condition represents a distinct data source. Specifically, the article establishes a general distribution estimation error bound in average total variation distance for conditional maximum likelihood estimation based on the bracketing number.
Result shows that when source distributions share certain similarities and the model is expressive enough, multi-source training guarantees a sharper bound than single-source training.
They further instantiate the general theory on conditional Gaussian estimation and deep generative models including autoregressive and flexible energy-based models, by characterizing their bracketing numbers. Simulations and real-world experiments validate this theory.

**Claims And Evidence:**

Yes.

**Essential References Not Discussed:**

No.

**Experimental Designs Or Analyses:**

Yes, I have checked them.

**Methods And Evaluation Criteria:**

It makes much sense to the problem at hand.

**Other Comments Or Suggestions:**

No.

**Other Strengths And Weaknesses:**

## Strength
This article is well organized and well written, using mathematical notations clearly and making it easy to understand. The new theory establishes a theoretical bridge between the general error bound of single-source training and multi-source training in conditional generative modelling. This will guide the researchers in choosing the data sources and models empirically and theoretically.

## Weakness

This article has a small issue. Providing intuitive explanations for some theoretical concepts would be beneficial. Additionally, the assumptions in this work deviate slightly from real environments.

**Questions For Authors:**

1. Can you provide an intuitive explanation of the ε-upper bracketing number? An example would be helpful.

2. I initially felt some counter intuitiveness about the error bound of a single source. I believe one model training with one dataset has no relation to the whole number of datasets K, but I realize that this work computes the accumulated error of all models, so the error bound will be related to K. I don't know if I understand you correctly. I would like to discuss this with the author.

3.  In real-world training, the model with large parameters for multi-source training does not always achieve better results than small expert models with single-source training. For example, when K=10 for Model=10, counting all errors of these 10 models corresponding to one dataset will be higher than a model for all these 10 datasets. However, the error of each model corresponding to one dataset will not always be higher than the model with 10 datasets. What is your opinion of this example, and does this example match your theory?

4.  I would like to know if the N and K in Table 1 are set to larger values, such as N=1500 or 2000 and K =15 or 20, the results will have a similar tendency to those in Table 1.


5. The theoretical analysis for the error bound is based on EBM and ARM; however, the article does not provide numerical results for these models but with diffusion models. I think the author should provide some experiment results on ARM or EBM, which would be better.

6. Is the model for multi-source training the same as the single-source training in theoretical analysis and empirical training?

**Relation To Broader Scientific Literature:**

The author has listed these findings in the related work and preliminary section.

**Theoretical Claims:**

Yes, I have checked them.

---

> ### Author Rebuttal · Authors · 2025-03-31
>
> # Q1: Intuitive explanation for upper bracketing number
>
> The $\epsilon$-upper bracketing number is a notion to quantify the complexity of an infinite set of functions. The key idea is to construct a finite collection of "brackets" that enclose every function in the set within a small margin.
>
> To illustrate this, consider a simple example. Suppose we have the infinite function set $\mathcal{F} = \lbrace f(x) = c:x\in [0,1];c \in [0,1] \rbrace,$ which consists of all constant functions taking values in the interval $[0, 1]$. We can construct an $\epsilon$-upper bracket for $\mathcal{F}$ by defining a finite set $\mathcal{B} = \lbrace b(x)=k\epsilon : k=0,1,\dots,\lceil 1/\epsilon \rceil \rbrace,$ which contains $\lceil 1/\epsilon \rceil+1$ functions. Then, for any function $f \in \mathcal{F}$, there exists a bracket function $b \in \mathcal{B}$ such that:
> (1) For all $x \in [0,1]$, the bracket function is always an upper bound: $b(x) \ge f(x)$.
> (2) The total "gap" between $b$ and $f$, measured by the integral $\int_0^1 b(x) - f(x) dx$, is at most $\epsilon$. Therefore, the $\epsilon$-upper bracketing number of $\mathcal{F}$ is at most $\lceil 1/\epsilon \rceil+1$.
>
> In our paper, we extend this idea to conditional probability spaces. There, each condition defines its own function set, and we construct corresponding upper brackets that ensure every conditional distribution is approximated with a small error uniformly across conditions.
>
> We will include additional intuitive explanations and diagrams to make this idea more accessible in the revised version.
>
> # Q2: Definition of the estimation error
>
> Your interpretation is essentially correct. In our paper, we define the error in terms of the average TV distance (see Equation 4 on line 147, left column). This metric evaluates the accuracy of conditional distribution estimates across all $K$ sources by averaging the error over each source. Therefore, even for single-source training, the error bound is related to $K$ because it aggregates the errors from the $K$ separate models.
>
> # Q3: Guarantee on one specific source
> You have correctly captured the main idea. Our theory demonstrates that, in terms of the average distribution error, multi-source training has a better guarantee than single-source training. However, this does not necessarily imply that for every individual source, the corresponding multi-source model will yield lower error than a dedicated single-source model. Thus, your example is consistent with our theoretical findings.
>
> # Q4: Real-world experiments with larger $N$ and $K$
>
> We would like to clarify that the selection of sample sizes and the number of classes in the experiments in Section 5.2 was influenced by several inherent characteristics of ILSVRC2012 dataset:
>
> - Sample Sizes: The maximum number of images per class in ILSVRC2012 is 1300, so we selected sample sizes of 1000 and 500 images per class, which are common choices.
>
> - Number of Sources: Given that distribution similarity levels were manually defined, it was difficult to establish a large number of structured subdivisions. To be specific, to ensure reasonable similarity levels for the controlled experiment, we designed two-level tree structure for the dataset, as shown in Figure 3 on Page 35 of our submission. Overall, we divided the whole ILSVRC2012 into 10 high-level categories (mammal, amphibian, bird, fish, reptile, vehicle, furniture, musical instrument, geological formation, and utensil). Each category was further subdivided into 10 subsets (e.g., for mammals, we have Italian greyhound, Border terrier, standard schnauzer, etc.). Defining such semantically meaningful and mutually exclusive divisions is not trivial. As a result, the number of classes within each similarity level in our experiments is limited to 10.
>
> Additionally, for the 10-dimensional Gaussian example in Section 5.1, we used a maximum sample size of $N = 5000$ and $K = 15$ (see Figure 1(a) and (b)), which we believe are sufficiently large to verify the theoretical predictions in that case.
>
> We will add the above explanations for the experimental settings in our revised version.
>
> # Q5: Experiments for ARMs or EBMs
>
> Following the reviewer's suggestion, we have added supplementary simulations for ARMs and further illustrations for EBMs. Generally speaking, for ARMs, the empirical TV errors exhibit similar trends as theoretical bounds in Section 4.2 regarding several key factors—the number of sources $K$, sample size $n$, and data length $D$. For EBMs, we clarify their connection with the diffusion model experiments. Due to space constraints in the rebuttal, please refer to our response to Reviewer 7LUU (Q1) for details.
>
> # Q6: Consistency of models used for multi/single
> Yes, for both theoretical analysis and empirical experiments, the models used for multi-source and single-source training are exactly the same across all settings, such as model architecture, number of parameters, initialization, and optimizers.

---

> > ### Comment · Reviewer_MrF2 · 2025-04-08
> >
> > Thanks for the clarification. I have read the author's rebuttal for all reviewers, and it has solved my concerns. So, my evaluation remains unchanged.

---

> > > ### Author Response · Authors · 2025-04-08
> > >
> > > We thank Reviewer MrF2 for acknowledging our contributions and constructive feedback.

---

### Official Review · Reviewer_QALB · 2025-03-21

**Overall Recommendation:** 4

**Summary:**

This paper investigates conditional generative models with multiple data sources. It establishes a general upper bound on the MLE error. The theoretical result is then specialized to conditional Gaussian distributions, autoregressive models, and energy-based models. Finally, the theoretical findings are validated through both simulation studies and real-world experiments.

**Claims And Evidence:**

I think the advantage of multi-source training is not very convincing, and the characterization is somewhat confusing.

Intuitively, multi-source training should only be beneficial when there are similarities among different classes. For example, in the Gaussian distribution setting in Section 4.1, when $d_1 = d$, i.e., there are no shared features across all sources, single-source training should perform just as well as multi-source training. Therefore, a reasonable characterization of the advantage should involve conditions on the ground truth data distribution. However, in Section 4.3, the advantage of multi-source learning is quantified by $S$ and $d_e$, both of which are parameters of the distribution family rather than direct information about the underlying data distribution.

**Essential References Not Discussed:**

No.

**Experimental Designs Or Analyses:**

The experimental designs are reasonable.

**Methods And Evaluation Criteria:**

Yes

**Other Comments Or Suggestions:**

None.

**Other Strengths And Weaknesses:**

This paper is overall well-written and presents a solid theoretical framework for multi-source learning.

**Questions For Authors:**

None.

**Relation To Broader Scientific Literature:**

This paper provides theoretical guarantees for conditional generative models, with results applicable to both large language models and diffusion models.

**Theoretical Claims:**

No

---

> ### Author Rebuttal · Authors · 2025-03-31
>
> # Q1: Characterization of multi-source advantage
>
> We thank the reviewer for the insightful comment.
>
> We would like to clarify that the advantage of multi-source training is indeed measured by the model parameter sharing, while the degree of the model parameter sharing reflects the source distribution similarity under our theoretical formulation (lines 94-108, right column).
>
> We understand the reviewer’s concern. In the Gaussian model (Section 4.1), $\beta_{sim} = \frac{d - d_1}{d}$ measures the proportion of shared mean vector dimensions, which seems to correspond to the property of the ground truth distribution. While for EBMs (Section 4.3), $\beta_{sim} = \frac{S}{S+d_e}$ is based on model parameters, which does not explicitly represent the data distribution itself.
>
> Despite this difference, in both cases, $\beta_{sim}$ is fundamentally defined by the extent of parameter sharing across sources. The distinction arises from the modeling paradigm: the Gaussian case assumes a parametric form for distributions, where model parameters (e.g., mean vectors) explicitly encode data properties, whereas EBMs use neural networks as a function approximator to fit probability densities without a predefined distributional form, making no explicit connection between parameters and data.
>
> We will add detailed clarification on the relationship between parameter sharing, distribution similarity, and the advantages of multi-source training in the revised version.

---

> > ### Comment · Reviewer_QALB · 2025-04-03
> >
> > Thanks for the clarification. My evaluation remains unchanged.

---

> > > ### Author Response · Authors · 2025-04-08
> > >
> > > We thank Reviewer QALB for acknowledging our contributions and constructive feedback.

---

### Official Review · Reviewer_uwo5 · 2025-03-21

**Overall Recommendation:** 3

**Summary:**

This paper provides a theoretical framework proving that training conditional generative models on multiple data sources outperforms single-source training when sources share similarities. The authors instantiate their theory across Gaussian distributions, autoregressive models, and energy-based models, demonstrating that both the number of sources and their similarity improves multi-source training benefits. Simulations and experiments with diffusion models validate the theoretical findings, explaining why large generative models trained on diverse but related data often perform better than specialized models.

**Claims And Evidence:**

The paper's claims are supported by theoretical proofs and empirical evidence that appear convincing.

**Essential References Not Discussed:**

N/A

**Experimental Designs Or Analyses:**

I noticed that Figure 1 shows a very close alignment between theoretical and empirical results. This perfect alignment is somewhat suspicious. From my understanding, the theoretical bounds are derived using worst-case analysis and typically contain constants that are not optimized, making perfect alignment with empirical results unusual. In most papers comparing theory and practice, you'd expect to see similar trends but with some gap between theoretical bounds and empirical measurements. Can the authors explain more about that?

The theoretical results cover three model types (Gaussian, ARM, EBM), but real-world experiments focus only on diffusion models, with no validation for autoregressive models. This should be a more important experiment.

The theory addresses large-scale generative modeling, but experiments use relatively small datasets (500-1000 images per class) and only up to 10 classes, raising questions about how well the findings generalize to truly large-scale settings.

**Methods And Evaluation Criteria:**

Yes.

This paper uses bracketing numbers as a theoretical tool to measure distribution space complexity, which is well-suited for analyzing generative model estimation errors.

And three representative model types (Gaussian, autoregressive, energy-based) are selected that cover key generative modeling approaches.

The evaluation framework connects TV error to FID, and systematically varies K and βsim identified in the theory, making the approach well-aligned with the problem being studied.

**Other Comments Or Suggestions:**

See the sections above.

**Other Strengths And Weaknesses:**

I really like the topic of this paper and believe this is very important to the field.

Even though the theoretical proof seems solid, the experiment part is weak. The whole section focuses on simple cases, and hard to evaluate the generality of the theory. There are more factors that are not involved, such as ARM.

**Questions For Authors:**

See the sections above.

**Relation To Broader Scientific Literature:**

N/A

**Theoretical Claims:**

I'm not an expert in theoretical deep learning, but I feel the structure of the proof is sound and clear. Although I haven't checked the details of the proof, it seems solid to me.

---

> ### Author Rebuttal · Authors · 2025-03-31
>
> # Q1: Close alignment in Figure 1
>
> We appreciate the reviewer’s careful examination of Figure 1.
>
> As detailed in lines 339-340 of our submission (the caption of Figure 1), the empirical and theoretical values are plotted on **separate vertical axes**: empirical values correspond to the left axis, while theoretical values correspond to the right axis. This visualization normalizes differences in constants between empirical results and theoretical bounds, emphasizing the comparison of trends rather than absolute values.
>
> We will highlight the dual-axis annotation in Figure 1 in the revised version to avoid any potential confusion.
>
> # Q2: Experiments for ARMs
>
> We thank the reviewer for the valuable comment.
>
> Following the reviewer's suggestion, we have conducted supplementary simulations for ARMs according to the formulation in Section 4.2. Experimental settings and results are presented below. Generally speaking, the empirical TV errors exhibit similar trends as theoretical bounds in Theorem 4.3 regarding several key factors—the number of sources $K$, sample size $n$, and data length $D$.
>
> In all experiments, we define a ground-truth sequential discrete distribution, enabling exact computation of the practical TV error. We fix the vocabulary size $M=2$, neural network configurations with $d_e=W=64$, $L=5$, $B=1$. Then we vary $K$ in $[1, 3, 5, 7, 10]$, $n$ in $[1000, 3000, 5000, 10000, 30000]$, and $D$ in $[10, 12, 14, 16, 18]$ to examine alignment of practical TV error with theoretical bounds. For each setting, the batch size and learning rate are selected from $\lbrace 100, 300, 500 \rbrace$ and $\lbrace 10^{-5}, 10^{-4}, 10^{-3} \rbrace$ for the minimum likelihood. Empirical TV errors are presented in the following tables:
>
> | $K \uparrow$ | 1      | 3      | 5      | 7      | 10     |
> | --- | --- | --- | --- | --- | --- |
> | single       | 0.0763 | 0.1212 | 0.1519 | 0.1787 | 0.2127 |
> | multi        | 0.0763 | 0.1145 | 0.1318 | 0.1364 | 0.1369 |
>
> | $n \downarrow$    | 1000   | 3000   | 5000   | 10000  | 30000  |
> | --- | --- | --- | --- | --- | --- |
> | single | 0.5680 | 0.3516 | 0.2882 | 0.2036 | 0.1212 |
> | multi  | 0.5491 | 0.3467 | 0.2747 | 0.1922 | 0.1145 |
>
> | $D \uparrow$ | 10  | 12  | 14  | 16  | 18  |
> | --- | --- | --- | --- | --- | --- |
> | single       | 0.2036 | 0.3785 | 0.5932 | 0.7242 | 0.7505 |
> | multi        | 0.1922 | 0.3530 | 0.5068 | 0.5747 | 0.6289 |
>
> The results show consistent trends between practical TV error and theoretical bounds with respect to $n$, $K$, and $D$, i.e., the TV error decreases as $n$ and increases with $K$ and $D$, and multi-source training generally outperforms single-source training.
>
> We will add the simulation results and implementation details for ARMs in the revised version.
>
> # Q3: Real-world experiments are on small datasets
>
> We would like to clarify that the selection of sample sizes and the number of classes in the experiments in Section 5.2 was influenced by several inherent characteristics of ILSVRC2012 dataset:
>
> - Sample Sizes: The maximum number of images per class in ILSVRC2012 is 1300, so we selected sample sizes of 1000 and 500 images per class, which are common choices.
>
> - Number of Sources: Given that distribution similarity levels were manually defined, it was difficult to establish a large number of structured subdivisions. To be specific, to ensure reasonable similarity levels for the controlled experiment, we designed two-level tree structure for the dataset, as shown in Figure 3 on Page 35 of our submission. Overall, we divided the whole ILSVRC2012 into 10 high-level categories (mammal, amphibian, bird, fish, reptile, vehicle, furniture, musical instrument, geological formation, and utensil). Each category was further subdivided into 10 subsets (e.g., for mammals, we have Italian greyhound, Border terrier, standard schnauzer, etc.). Defining such semantically meaningful and mutually exclusive divisions is not trivial. As a result, the number of classes within each similarity level in our experiments is limited to 10.
>
> While our experiments are not on large-scale datasets, there are existing studies that provide valuable empirical observations for large-scale multi-source training as mentioned in our Introduction section (lines 8-13, right column), including: cross-lingual model transfer for similar languages [Pires et al., 2019], pretraining with additionla high-quality images to improve overall aesthetics in image generation [Chen et al., 2024], and knowledge augmentation on subsets of data to enhance model performance on other subsets [Allen-Zhu & Li, 2024a]. They have offered relevant findings that inform our work.
>
> We will provide a more detailed explanation of our experimental settings in the revised version.
>
> To summarize, we sincerely thank the reviewer for the constructive comments regarding our experiments, which we believe can improve the quality of this paper.

---

### Official Review · Reviewer_7LUU · 2025-03-24

**Overall Recommendation:** 3

**Summary:**

The paper establishes a distribution estimation error bound in average total variation distance for conditional maximum likelihood estimation. The main result is based on the bracketing number; it shows that when source distributions share certain similarities and the model is expressive enough, multi-source training guarantees a sharper bound than single-source training.

**Claims And Evidence:**

The paper focuses mostly on the theoretical part of the claims, which is backed by detailed proofs. Simulations and real-world experiments on diffusion models partly validate the results.

**Essential References Not Discussed:**

None.

**Experimental Designs Or Analyses:**

I am not entirely sure if I missed something. But it seems that the paper characterizes the bracketing numbers for conditional Gaussian
 estimation, autoregressive models and energy-based models, but the real-world experiments focus largely on a particular diffusion model, i.e., EDM2. There seems to be a discrepancy between the proposed theory and its numerical validation.

**Methods And Evaluation Criteria:**

The evaluation criteria seems reasonable.

**Other Comments Or Suggestions:**

I would like to see more experimental results regarding autoregressive models or EBMs to validate the theoretical results.

**Other Strengths And Weaknesses:**

Please see the ``Experimental Designs Or Analyses`` section.

**Questions For Authors:**

Sorry if I missed this part, but can the authors advise on how to quantify the source similarity $\beta_{\rm sim}$ for general datasets?

**Relation To Broader Scientific Literature:**

The paper establishes a new framework for the analysis of multi-source training in conditional generative modeling, where each condition represents a distinct data source. This could be potentially helpful for general multimodal data learning.

**Theoretical Claims:**

I am not very familiar with this particular problem. Due to time limit, I did not carefully check the proofs.

---

> ### Author Rebuttal · Authors · 2025-03-31
>
> # Q1: Experiments for ARMs or EBMs
>
> We thank the reviewer for the valuable comment.
>
> Following the reviewer's suggestion, we have conducted supplementary simulations for ARMs according to the formulation in Section 4.2. Experimental settings and results are presented below. Generally speaking, the empirical TV errors exhibit similar trends as theoretical bounds in Theorem 4.3 regarding several key factors—the number of sources $K$, sample size $n$, and data length $D$.
>
> In all experiments, we define a ground-truth sequential discrete distribution, enabling exact computation of the practical TV error. We fix the vocabulary size $M=2$, neural network configurations with $d_e=W=64$, $L=5$, $B=1$. Then we vary $K$ in $[1, 3, 5, 7, 10]$, $n$ in $[1000, 3000, 5000, 10000, 30000]$, and $D$ in $[10, 12, 14, 16, 18]$ to examine alignment of practical TV error with theoretical bounds. For each setting, the batch size and learning rate are selected from $\lbrace 100, 300, 500 \rbrace$ and $\lbrace 10^{-5}, 10^{-4}, 10^{-3} \rbrace$ for the minimum likelihood. Empirical TV errors are presented in the following tables:
>
> | $K \uparrow$ | 1      | 3      | 5      | 7      | 10     |
> | --- | --- | --- | --- | --- | --- |
> | single       | 0.0763 | 0.1212 | 0.1519 | 0.1787 | 0.2127 |
> | multi        | 0.0763 | 0.1145 | 0.1318 | 0.1364 | 0.1369 |
>
> | $n \downarrow$    | 1000   | 3000   | 5000   | 10000  | 30000  |
> | --- | --- | --- | --- | --- | --- |
> | single | 0.5680 | 0.3516 | 0.2882 | 0.2036 | 0.1212 |
> | multi  | 0.5491 | 0.3467 | 0.2747 | 0.1922 | 0.1145 |
>
> | $D \uparrow$ | 10  | 12  | 14  | 16  | 18  |
> | --- | --- | --- | --- | --- | --- |
> | single       | 0.2036 | 0.3785 | 0.5932 | 0.7242 | 0.7505 |
> | multi        | 0.1922 | 0.3530 | 0.5068 | 0.5747 | 0.6289 |
>
> The results show consistent trends between empirical and theoretical values with respect to $n$, $K$, and $D$, i.e., the TV error decreases as $n$ and increases with $K$ and $D$, and multi-source training generally outperforms single-source training.
>
> Additionally, we would like to clarify the connection between our diffusion model experiments and the theoretical analysis of EBMs. As mentioned in lines 51-55 in our submission, EBMs are a general and flexible class of generative models closely connected to diffusion models. To be specific, first, the training and sampling methods in [1,2] are directly inspired by EBMs. The distinction is that EBMs parameterize the energy function, while diffusion models parameterize its gradient (the score function). Second, [3] shows that under a specific energy function formulation (Equation (5) in their paper), EBMs are equivalent to constrained diffusion models. Their experimental results (Table 1, Rows A and B) indicate that the constraint has minor impact on generative performance. Thus, our diffusion model experiments provide insight into EBMs' behavior in real-world settings to some extent.
>
> We will provide the implementation details and simulation results for ARMs in the revised version of our paper, along with the above discussions for EBMs.
>
> [1] Song, Y., & Ermon, S. (2019). Generative modeling by estimating gradients of the data distribution.
>
> [2] Song, Y., Sohl-Dickstein, J., Kingma, D. P., Kumar, A., Ermon, S., & Poole, B. (2020). Score-based generative modeling through stochastic differential equations.
>
> [3] Salimans, T., & Ho, J. (2021). Should EBMs model the energy or the score?
>
> # Q2: Quantifying similarity for general datasets
>
> We thank the reviewer for raising this insightful question.
>
> In our paper, $\beta_{sim}$ is defined by induction based on our three specific model instantiations in Section 4. It is not an inherent, directly measurable property of the source distributions themselves, meaning it cannot be directly computed given general datasets.
>
> A fundamental question underlying the reviewer's inquiry might be: *How can we quantify dataset similarity in practice with theoretical guarantees?* We acknowledge that there is no single method currently that provides a solution to this problem, and we are still exploring ways towards this goal.
>
> Possible approaches might include: (1) From a practical perspective, a small proxy model can be used to estimate source distributions' interaction [4]. (2) From a theoretical perspective, several existing notions in multi-task learning and meta-learning could be adapted for this purpose, such as transformation equivalence [5], parameter distance [6], and distribution divergence [7].
>
> [4] Xie, S. M., Pham, H., Dong, X., et al (2023). Doremi: Optimizing data mixtures speeds up language model pretraining.
>
> [5] Ben-David, S., & Borbely, R. S. (2008). A notion of task relatedness yielding provable multiple-task learning guarantees.
>
> [6] Balcan, M. F., Khodak, M., & Talwalkar, A. (2019). Provable guarantees for gradient-based meta-learning.
>
> [7] Jose, S. T., & Simeone, O. (2021). An information-theoretic analysis of the impact of task similarity on meta-learning.

---

### Decision · Program_Chairs · 2025-05-01

**Decision:**

Accept (poster)

**Comment:**

This paper introduces a theoretical framework for analyzing multi-source training in conditional generative modeling, establishing distribution estimation error bounds that demonstrate when and why multi-source training outperforms single-source approaches. The authors provide an analysis through the lens of bracketing numbers, instantiating their theory across conditional Gaussian estimation, autoregressive models, and energy-based models. Reviewers acknowledged the paper's theoretical soundness and appreciated the authors' thorough responses to concerns about experimental validation, particularly their additional ARM simulations and clarification of the connection between EBMs and diffusion models. While some reviewers initially questioned the characterization of source similarity and the scale of experiments, the authors addressed these points in their rebuttal, providing additional results and explaining practical constraints. Given the paper's theoretical contribution to understanding an important phenomenon in generative modeling and the authors' willingness to enhance their experimental validation, I recommend acceptance of this paper to ICML 2025.